# Leveraging Explanation to Improve Generalization of Meta Reinforcement Learning

**Shicheng Liu & Minghui Zhu**
Department of Electrical Engineering
Pennsylvania State University
University Park, PA 16802, USA
{sfl5539,muz16}@psu.edu

## Abstract

A common and effective human strategy to improve a poor outcome is to first identify prior experiences most relevant to the outcome and then focus on learning from those experiences. This paper investigates whether this human strategy can improve generalization of meta-reinforcement learning (MRL). MRL learns a meta-prior from a set of training tasks such that the meta-prior can adapt to new tasks in a distribution. However, the meta-prior usually has imbalanced generalization, i.e., it adapts well to some tasks but adapts poorly to others. We propose a two-stage approach to improve generalization. The first stage identifies "critical" training tasks that are most relevant to achieve good performance on the poorly adapted tasks. The second stage improves generalization by encouraging the meta-prior to pay more attention to the critical tasks. We use conditional mutual information to mathematically formalize the notion of "paying more attention". We formulate a bilevel optimization problem to maximize the conditional mutual information by augmenting the critical tasks and propose an algorithm to solve the bilevel optimization problem. We theoretically guarantee that (1) the algorithm converges at the rate of $O(1/\sqrt{K})$ and (2) the generalization improves after the task augmentation. We use two real-world experiments, two MuJoCo experiments, and a Meta-World experiment to validate the algorithm.

## 1 Introduction

Meta-reinforcement learning (MRL) learns a meta-prior from a set of training tasks where each training task is an RL problem and is drawn from an implicit task distribution. The predominant approach in existing works (Beck et al., 2023) is to learn a meta-policy as the meta-prior. In this paper, we follow this standard setting and denote the learned meta-policy by $\pi_0$. The goal is for $\pi_0$ to generalize effectively across the task distribution. However, both prior works (Dhillon et al., 2019; Nguyen et al., 2021; Yu et al., 2020) and our empirical findings (see Appendix M.9) indicate that $\pi_0$ usually adapts well to some tasks but poorly to others. This paper proposes a method to improve generalization of the meta-policy $\pi_0$. Our method is inspired by an effective strategy that humans commonly use in daily life to improve a poor outcome, where humans first identify prior experiences most relevant to the poor outcome and then focus on learning from these experiences to improve the outcome. For example, if a student fails some problems in an exam, a common improvement strategy for the student is to find similar problems from previous homework and focus more on these problems in future study. Our approach consists of two stages. The first stage identifies "critical" training tasks most important to the poorly adapted tasks. The second stage encourages $\pi_0$ to focus more on the critical tasks to improve generalization of $\pi_0$. Note that our approach operates in a post hoc setting, i.e., after the MRL algorithm has already produced a meta-policy $\pi_0$.

The first stage proposes an example-based explanation method to identify the training tasks most relevant to the poorly adapted tasks. Example-based explanation is widely used in explainable machine learning (Caruana et al., 1999; Sun et al., 2024) to explain a model's decision through relevant examples. It is inspired by the observation that humans usually use relevant experiences to interpret a new thing (Crabbé et al., 2021). In our case, we formulate a bilevel optimization problem where the upper level learns a weight vector to weight each training task such that the corresponding weighted

meta-policy performs best on the poorly adapted tasks and the lower level learns this weighted meta-policy. The training tasks with highest weights are the most important/relevant tasks to achieve good performance on the poorly adapted tasks. We refer to these training tasks as "critical" tasks.

The second stage improves generalization by encouraging the meta-policy to pay more attention to the critical tasks. We mathematically formalize the notion of "attention" through an information-theoretic lens. Specifically, we use the mutual information between the meta-policy and the critical tasks to quantify the task information of the critical tasks stored in the meta-policy (Yin et al., 2019; Yao et al., 2021). An increase in mutual information indicates that the meta-policy stores more information of the critical tasks and thus pays more attention to the critical tasks. To increase this mutual information, we propose to augment the critical tasks by generating augmented data. The augmented data enhances data diversity of the critical tasks and contains additional information. Therefore, it is expected that the meta-policy trained on the augmented data stores more information of the critical task. Data augmentation has been applied to RL (Wang et al., 2020; Laskin et al., 2020) and meta-learning (Yao et al., 2021; Rajendran et al., 2020), but these methods use predefined rules to augment the data. While the predefined augmentation rules can increase the task information of the critical tasks stored in the meta-policy, they do not maximally increase such stored information. Motivated by (Yin et al., 2019) which improves generalization by maximizing mutual information between the task data and meta-parameter, we formulate a bilevel optimization problem. In the upper level, we learn how to augment the critical tasks to maximally increase the stored information. To achieve this, we use conditional mutual information (CMI) to quantify the additional information of the critical tasks stored in the meta-policy after the task augmentation, and learn an augmentation method to maximize CMI. The difficulty of the upper-level optimization is to compute a distribution of the meta-policy. Therefore, the lower level formulates a distributional optimization problem where a meta-policy distribution corresponding to the current augmentation is learned.

We include related works in Appendix A and summarize our contributions as follows.

**Contribution statement**. This paper proposes to leverage explanation to improve generalization of the specific meta-policy $\pi_0$. Our contributions are threefold:

First, we propose an example-based explanation method to identify the critical training tasks that are most important/relevant to the poorly adapted tasks as an explanation.

Second, we introduce an information-theoretic framework and formalize the problem of leveraging the explanation to improve generalization as a bilevel optimization problem. The upper level learns how to augment the critical tasks to maximize the conditional mutual information, and the lower level computes the meta-policy distribution corresponding to the current augmentation. We propose an algorithm to solve the bilevel optimization problem.

Third, we theoretically guarantee that (i) our algorithm converges at the rate of $O(1/\sqrt{K})$ and (ii) the generalization improves after the task augmentation. We use two real-world experiments, two MuJoCo experiments, and a Meta-World experiment to empirically validate that our algorithm can improve the generalization of the meta-policy $\pi_0$.

## 2 PRELIMINARIES

**Reinforcement learning**. An RL task $\mathcal{T}_i$ is based on a Markov decision process (MDP) $\mathcal{M}_i = (\mathcal{S}, \mathcal{A}, \gamma, P_i, \nu_i, r_i)$ which includes a state set $\mathcal{S}$, an action set $\mathcal{A}$, a discount factor $\gamma \in (0, 1)$, a state transition function $P_i(\cdot|\cdot, \cdot)$, an initial state distribution $\nu_i(\cdot)$, and a reward function $r_i(\cdot, \cdot)$. RL learns a policy $\pi_\varphi$ to maximize the cumulative reward $\max_\varphi E^{\pi_\varphi}[\sum_{t=0}^\infty \gamma^t r_i(s_t, a_t)|s_0 \sim \nu_i]$. The policy gradient is $E_{(s,a)\sim\rho^{\pi_\varphi}}[\nabla_\varphi \log \pi_\varphi(a|s) A_i^{\pi_\varphi}(s, a)]$ where $A_i^\pi$ is the advantage function under the reward $r_i$ and policy $\pi$, $\rho^\pi(s, a) \triangleq E^\pi[\sum_{t=0}^\infty \gamma^t \mathbb{1}\{s_t = s, a_t = a\}|s_0 \sim \nu_i]$ is the stationary state-action distribution of the policy $\pi$, and $\mathbb{1}\{\cdot\}$ is the indicator function. Based on the policy gradient, we can formulate a surrogate objective for RL (Wang et al., 2020): $J_i(\pi) \triangleq E_{(s,a)\sim\rho^\pi}[\log \pi(a|s) A_i^\pi(s, a)]$. For brevity, we omit the explicit notation of the policy parameter $\varphi$.

**Meta-reinforcement learning**. MRL aims to efficiently solve multiple RL tasks by learning a meta-policy. The meta-policy is learned from a group of $N^{\text{tr}}$ training tasks $\{\mathcal{T}_i^{\text{tr}}\}_{i=1}^{N^{\text{tr}}}$ sampled from an implicit task distribution $P(\mathcal{T})$. It is typically assumed (Beck et al., 2023) that different tasks

share $(\mathcal{S}, \mathcal{A}, \gamma)$ but may have different $(P_i^{\mathrm{tr}}, \nu_i^{\mathrm{tr}}, r_i^{\mathrm{tr}})$. Here, the superscript "tr" means that these components belong to training tasks. Later on, we will use different superscripts to represent different kinds of tasks. Current mainstream MRL works (Beck et al., 2023; Finn et al., 2017; Fallah et al., 2021; Xu et al., 2018; Liu et al., 2019) have the following bilevel structure:

$$\max_{\theta} \ L(\theta, \{\mathcal{T}_i^{\mathrm{tr}}\}_{i=1}^{N^{\mathrm{tr}}}) = \frac{1}{N^{\mathrm{tr}}} \sum_{i=1}^{N^{\mathrm{tr}}} J_i^{\mathrm{tr}}(\pi_i^{\mathrm{tr}}(\theta)), \quad \text{s.t. } \pi_i^{\mathrm{tr}}(\theta) = Alg(\pi_\theta, \mathcal{T}_i^{\mathrm{tr}}), \tag{1}$$

where the upper level learns a meta-policy $\pi_\theta$ such that the corresponding task-specific adaptation $\pi_i^{\mathrm{tr}}(\theta)$ can maximize the cumulative reward $J_i^{\mathrm{tr}}(\pi_i^{\mathrm{tr}}(\theta))$ on each training task $\mathcal{T}_i^{\mathrm{tr}}$, and the lower level computes the task-specific adaptation $\pi_i^{\mathrm{tr}}(\theta)$ induced from $\pi_\theta$. Different meta-learning methods use different algorithms to compute the task-specific adaptation $\pi_i^{\mathrm{tr}}(\theta)$. Here, we use $Alg(\pi_\theta, \mathcal{T}_i^{\mathrm{tr}})$ to generally represent an algorithm that computes the task-specific adaptation.

We denote the meta-policy learned from (1) by $\pi_0$, and evaluate its generalization by sampling a set of new tasks from $P(\mathcal{T})$. For each sampled task, we perform task-specific adaptation and measure performance using the resulting cumulative reward. However, as noted in the prior work (Yu et al., 2020) and confirmed by our experiment (see Appendix M.9), only a subset of the adapted policies achieve high cumulative reward, while others perform poorly. Therefore, we can find $N^{\mathrm{poor}}$ poorly adapted tasks and denote them by the set $\{\mathcal{T}_i^{\mathrm{poor}}\}_{i=1}^{N^{\mathrm{poor}}}$. We include the details of how to find poorly adapted tasks in Appendix M.3.

## 3 EXAMPLE-BASED EXPLANATION

This section proposes an example-based explanation method. The proposed method is motivated by the recent advances in example-based explanation for RL. For example, (Liu & Zhu, 2025) identifies the state-action pairs that are most important to suboptimal performance as an explanation and (Liu et al., 2025b) identifies the preference data that is most important to unsatisfactory responses as an explanation. Inspired by these approaches, we extend the idea to the MRL setting and aim to identify the training tasks that are most important for the meta-policy to achieve high cumulative reward on the poorly adapted tasks $\{\mathcal{T}_i^{\mathrm{poor}}\}_{i=1}^{N^{\mathrm{poor}}}$ (after adaptation). We refer to these training tasks as "critical tasks" and aim to identify the top $N^{\mathrm{cri}}$ critical training tasks as an explanation. For this purpose, we propose to learn an importance vector $\omega \in \mathbb{R}^{N^{\mathrm{tr}}}$ where each dimension $\omega_i$ captures the importance of the corresponding training task $\mathcal{T}_i^{\mathrm{tr}}$ for improving cumulative reward on $\{\mathcal{T}_i^{\mathrm{poor}}\}_{i=1}^{N^{\mathrm{poor}}}$. The problem is formulated as the following bilevel optimization problem:

$$\max_{\omega} \ L(\theta^*(\omega), \{\mathcal{T}_i^{\mathrm{poor}}\}_{i=1}^{N^{\mathrm{poor}}}), \quad \text{s.t. } \theta^*(\omega) = \arg\max_{\theta} \sum_{i=1}^{N^{\mathrm{tr}}} \omega_i J_i^{\mathrm{tr}}(\pi_i^{\mathrm{tr}}(\theta)), \tag{2}$$

where the upper level learns how to weight each training task such that the corresponding weighted meta-policy $\pi_{\theta^*(\omega)}$ can adapt to $\{\mathcal{T}_i^{\mathrm{poor}}\}_{i=1}^{N^{\mathrm{poor}}}$ with maximum cumulative reward, and the lower level computes the weighted meta-policy $\pi_{\theta^*(\omega)}$ corresponding to the current weight $\omega$. We include the algorithm to solve the problem (2) in Appendix B. The algorithm is single-loop where at each iteration, we start from the parameters in the previous iteration and only partially solve both the upper-level and lower-level problems via one-step gradient ascent.

We denote by $\omega^*$ an optimal solution of the problem (2). A larger value of $\omega_i^*$ means that the weighted meta-policy $\pi_{\theta^*(\omega^*)}$ assigns more importance to the training task $\mathcal{T}_i^{\mathrm{tr}}$, indicating the high relevance of this task to achieve high cumulative reward on $\{\mathcal{T}_i^{\mathrm{poor}}\}_{i=1}^{N^{\mathrm{poor}}}$. Accordingly, we define the top $N^{\mathrm{cri}}$ training tasks with the highest weight values as the critical tasks, denoted by $\{\mathcal{T}_i^{\mathrm{cri}}\}_{i=1}^{N^{\mathrm{cri}}}$.

## 4 GENERALIZATION IMPROVEMENT VIA TASK AUGMENTATION

This section uses the explanation (i.e., the critical tasks $\{\mathcal{T}_i^{\mathrm{cri}}\}_{i=1}^{N^{\mathrm{cri}}}$) in Section 3 to improve generalization by encouraging the meta-policy $\pi_0$ to pay more attention to the critical tasks. *One may be concerned that paying attention to the critical tasks can degrade performance on other tasks, we include an evaluation in the experiment to demonstrate that our method only degrades very few (less*

*than 5%) tasks but the average performance on all the tasks always improves.* The key challenge to encourage the meta-policy to pay more attention to the critical tasks lies in how to mathematically formalize the notion of "paying more attention".

A natural approach is to assign higher weights to the critical tasks. However, existing task weighting methods (Yao et al., 2021; Cai et al., 2020) typically require an additional target task set to guide how to assign the weights. As a result, these methods aim to generalize specifically to the given target task set. In contrast, we do not have a target task set and our goal is not to generalize well to a specific target task set. While solving the problem (2) yields a weighted meta-policy $\pi_{\theta^*(\omega^*)}$ that improves generalization on $\{\mathcal{T}_i\}_{i=1}^{N^{\text{poor}}}$, the generalization over the task distribution may not improve. In Section 5, we empirically demonstrate that our method outperforms the task weighting method.

We study the notion of "attention" from an information-theoretic perspective. If the task information of the critical tasks stored in the meta-policy increases, it means that the meta-policy pays more attention to the critical tasks (Rajendran et al., 2020). To achieve this, we propose to augment the critical tasks by generating augmented data. The augmented data contains additional information and diversifies the original training data, therefore, training on the augmented critical tasks can store more information of the critical tasks in the meta-policy (formally proved in Theorem 2).

Inspired by the empirical success of mixup augmentation in improving generalization in supervised learning (Zhang et al., 2018), meta-learning (Yao et al., 2021), and RL (Wang et al., 2020), we adopt mixup to augment the critical tasks. Recall from Section 2 that the surrogate RL objective for a critical task $\mathcal{T}_i^{\text{cri}}$ is: $J_i^{\text{cri}}(\pi) = E_{(s,a)\sim\rho^\pi}[\log \pi(a|s)A_i^\pi(s,a)]$, where $\rho^\pi(s,a)$ is the stationary state-action distribution of the policy $\pi$. We define the stationary state distribution of $\pi$ as $\rho^\pi(s) \triangleq \int_{a\in\mathcal{A}} \rho^\pi(s,a)da$. To train a policy, we collect transition tuples $(s,a,r,s_{\text{next}})$ where $s \sim \rho^\pi(\cdot)$, the policy $\pi$ selects action $a$, and the environment returns reward $r$ and next state $s_{\text{next}}$. Given two sampled states $s, s' \sim \rho^\pi(\cdot)$, mixup generates an augmented state $\bar{s} = \lambda_i s + (1 - \lambda_i)s'$ where the mixing coefficient $\lambda_i \in [0, 1]$ of the critical task $\mathcal{T}_i^{\text{cri}}$ is sampled from a distribution $P(\lambda)$. In this paper, we use $\Lambda_i$ to denote a random variable that follows the distribution $P(\lambda)$ and use $\lambda_i$ to denote a specific value sampled from $P(\lambda)$. Note that our augmentation method assumes that the augmented state $\bar{s}$ is always feasible. In practice, the augmented state is always feasible in our experiments (explained in Appendix M.2). The policy $\pi$ then selects an action $\bar{a}$ at the augmented state $\bar{s}$, and executes this action to collect the augmented tuple $(\bar{s}, \bar{a}, \bar{r}, \bar{s}_{\text{next}})$. Since the augmented state $\bar{s}$ is different from the original two states $(s, s')$, the augmented tuple $(\bar{s}, \bar{a}, \bar{r}, \bar{s}')$ contains different information from the original two tuples $((s, a, r, s_{\text{next}})$ and $(s', a', r', s'_{\text{next}}))$. Therefore, adding augmented tuples will diversify the original training tuples and contain additional information.

For a specific $\lambda_i$, the mixup augmentation induces an augmented stationary state-action distribution $\bar{\rho}^{\pi,\lambda_i}(\cdot, \cdot)$ whose expression is in Appendix C. This gives rise to an augmented task $\bar{\mathcal{T}}_i^{\text{cri}}(\lambda_i)$, with the augmented surrogate objective defined as $\bar{J}_i^{\text{cri}}(\pi, \lambda_i) \triangleq E_{(\bar{s},\bar{a})\sim\bar{\rho}^{\pi,\lambda_i}}[\log \pi(\bar{a}|\bar{s})A_i^\pi(\bar{s},\bar{a})]$. Given the augmented critical tasks, the meta-objective (i.e., the upper-level objective) in (1) becomes:

$$L(\theta, \{\bar{\mathcal{T}}_i^{\text{cri}}(\lambda_i)\}_{i=1}^{N^{\text{cri}}}, \{\mathcal{T}_i^{\text{tr}}\}_{i=1}^{N^{\text{tr}}-N^{\text{cri}}}) \triangleq \frac{1}{N^{\text{tr}}}\Big[\sum_{i=1}^{N^{\text{cri}}} \bar{J}_i^{\text{cri}}(\pi_i^{\text{cri}}(\theta), \lambda_i) + \sum_{i=1}^{N^{\text{tr}}-N^{\text{cri}}} J_i^{\text{tr}}(\pi_i^{\text{tr}}(\theta))\Big]. \quad (3)$$

In contrast to the original meta-objective in (1), the new objective (3) replaces the original critical tasks $\{\mathcal{T}_i^{\text{cri}}\}_{i=1}^{N^{\text{cri}}}$ with the augmented critical tasks $\{\bar{\mathcal{T}}_i^{\text{cri}}(\lambda_i)\}_{i=1}^{N^{\text{cri}}}$. Since $\Lambda_i$ is a random variable drawn from $P(\lambda)$, the corresponding augmented task $\bar{\mathcal{T}}_i^{\text{cri}}(\Lambda_i)$ is also a random variable. In the following context, we use the notation $\bar{\mathcal{T}}_i^{\text{cri}}(\Lambda_i \sim P(\lambda))$ to highlight this stochasticity.

**Remark 1 (Augmentation is different from simply sampling more data).** *Augmentation is different from using the policy $\pi$ to sample more data. The reason is that the additional data sampled by $\pi$ will always follow the original state-action distribution $\rho^\pi$ and thus the RL objective remains as the original objective $J_i^{cri}(\pi)$. In contrast, augmentation changes the state-action distribution to $\bar{\rho}^{\pi,\lambda_i}$ and thus leads to a different optimization objective $\bar{J}_i^{cri}(\pi, \lambda_i)$.*

In this section, we study the problem of "paying more attention" from the perspective of "storing more information". We have intuitively explained that the task augmentation is expected to store more information of the critical tasks in the meta-policy because the augmented data diversifies the original training data and contains additional information. In the following context, we formalize this intuition and mathematically study the problem from an information-theoretic perspective. This

section has two parts. The first part formulates a bilevel optimization problem to learn how to augment the critical tasks to maximally increase the task information of the critical tasks stored in the meta-policy. The second part proposes an algorithm to solve the bilevel optimization problem and theoretically proves that the generalization improves after the task augmentation.

## 4.1 PROBLEM FORMULATION

In this part, we formulate a bilevel optimization problem that explicitly captures the goal of "paying more attention" to the critical tasks by storing more task information in the meta-policy. As previously discussed, augmenting the critical tasks is expected to increase the amount of task information stored in the meta-policy. We now formalize this intuition by introducing the following information-theoretic definition:

**Definition 1.** *The additional information stored in the meta-policy $\pi_\theta$ after the task augmentation can be quantified by $I(\theta; \{\bar{\mathcal{T}}_i^{\mathrm{cri}}(\Lambda_i \sim P(\lambda))\}_{i=1}^{N^{\mathrm{cri}}} | \{\mathcal{T}_i^{\mathrm{cri}}\}_{i=1}^{N^{\mathrm{cri}}})$ which is the conditional mutual information between the meta-parameter $\theta$ and the augmented critical tasks $\{\bar{\mathcal{T}}_i^{\mathrm{cri}}(\Lambda_i \sim P(\lambda))\}_{i=1}^{N^{\mathrm{cri}}}$, given the original critical tasks $\{\mathcal{T}_i^{\mathrm{cri}}\}_{i=1}^{N^{\mathrm{cri}}}$.*

In information theory (Wyner, 1978), the conditional mutual information quantifies the difference between the information that $\theta$ and $\{\bar{\mathcal{T}}_i^{\mathrm{cri}}(\Lambda_i \sim P(\lambda))\}_{i=1}^{N^{\mathrm{cri}}}$ share and the information that $\theta$ and $\{\mathcal{T}_i^{\mathrm{cri}}\}_{i=1}^{N^{\mathrm{cri}}}$ share. In other words, it quantifies the amount of additional information stored in $\theta$ by additionally knowing $\{\bar{\mathcal{T}}_i^{\mathrm{cri}}(\Lambda_i \sim P(\lambda))\}_{i=1}^{N^{\mathrm{cri}}}$ given that $\{\mathcal{T}_i^{\mathrm{cri}}\}_{i=1}^{N^{\mathrm{cri}}}$ is already known. Therefore, $I(\theta; \{\bar{\mathcal{T}}_i^{\mathrm{cri}}(\Lambda_i \sim P(\lambda))\}_{i=1}^{N^{\mathrm{cri}}} | \{\mathcal{T}_i^{\mathrm{cri}}\}_{i=1}^{N^{\mathrm{cri}}}) > 0$ means that the information of the critical tasks stored in the meta-parameter $\theta$ increases after we augment $\{\mathcal{T}_i^{\mathrm{cri}}\}_{i=1}^{N^{\mathrm{cri}}}$ to $\{\bar{\mathcal{T}}_i^{\mathrm{cri}}(\Lambda_i \sim P(\lambda))\}_{i=1}^{N^{\mathrm{cri}}}$.

Our objective is to augment the critical tasks to store more information, ensuring that $I(\theta; \{\bar{\mathcal{T}}_i^{\mathrm{cri}}(\Lambda_i \sim P(\lambda))\}_{i=1}^{N^{\mathrm{cri}}} | \{\mathcal{T}_i^{\mathrm{cri}}\}_{i=1}^{N^{\mathrm{cri}}}) > 0$. While the current mixup methods (Yao et al., 2021; Wang et al., 2020) use a predetermined distribution $P(\lambda)$ of $\lambda_i$ to mix the data, these methods do not guarantee that the resulting augmentation maximally increases the stored task information. Motivated by (Yin et al., 2019) that improves generalization by maximizing the mutual information between the task data and meta-parameter, we propose to learn an optimal augmentation by optimizing the distribution $P(\lambda)$ to maximize the conditional mutual information. To this end, we model the distribution of $\lambda$ as a parameterized distribution $P_{\phi_\lambda}(\lambda)$ with parameter $\phi_\lambda$. Our objective is to optimize the distribution parameter $\phi_\lambda$ to maximize the following conditional mutual information:

$$
\begin{aligned}
& I(\theta; \{\bar{\mathcal{T}}_i^{\mathrm{cri}}(\Lambda_i \sim P_{\phi_\lambda}(\lambda))\}_{i=1}^{N^{\mathrm{cri}}} | \{\mathcal{T}_i^{\mathrm{cri}}\}_{i=1}^{N^{\mathrm{cri}}}) \\
& = E_{\Lambda_i \in [0,1], \Lambda_i \sim P_{\phi_\lambda}(\lambda), \theta \sim P^*(\cdot | \{\bar{\mathcal{T}}_i^{\mathrm{cri}}(\Lambda_i)\}_{i=1}^{N^{\mathrm{cri}}})} \left[ \log P^*(\theta | \{\bar{\mathcal{T}}_i^{\mathrm{cri}}(\Lambda_i)\}_{i=1}^{N^{\mathrm{cri}}}) - \log P^*(\theta | \{\mathcal{T}_i^{\mathrm{cri}}\}_{i=1}^{N^{\mathrm{cri}}}) \right]
\end{aligned}
\tag{4}
$$

where the derivation is in Appendix D. Here, $P^*(\cdot | \{\bar{\mathcal{T}}_i^{\mathrm{cri}}(\Lambda_i)\}_{i=1}^{N^{\mathrm{cri}}})$ is the posterior distribution of the meta-parameter $\theta$ given the augmented critical tasks $\{\bar{\mathcal{T}}_i^{\mathrm{cri}}(\Lambda_i)\}_{i=1}^{N^{\mathrm{cri}}}$, while $P^*(\cdot | \{\mathcal{T}_i^{\mathrm{cri}}\}_{i=1}^{N^{\mathrm{cri}}})$ is the posterior distribution of $\theta$ if only the original critical tasks $\{\mathcal{T}_i^{\mathrm{cri}}\}_{i=1}^{N^{\mathrm{cri}}}$ are known. The difference $\log P^*(\theta | \{\bar{\mathcal{T}}_i^{\mathrm{cri}}(\Lambda_i)\}_{i=1}^{N^{\mathrm{cri}}}) - \log P^*(\theta | \{\mathcal{T}_i^{\mathrm{cri}}\}_{i=1}^{N^{\mathrm{cri}}})$ measures how much observing the augmented critical tasks changes our belief about the meta-parameter $\theta$ compared to observing only the original tasks. If the augmented tasks are informative, the posterior $P^*(\theta | \{\bar{\mathcal{T}}_i^{\mathrm{cri}}(\Lambda_i)\}_{i=1}^{N^{\mathrm{cri}}})$ becomes more concentrated, yielding a larger log-likelihood for plausible values of $\theta$; if the augmentation provides no additional information, the two log-posteriors coincide and the difference becomes zero. For simplicity, we omit the dependence on the non-critical training tasks (i.e., other training tasks that are not critical tasks), as they remain unchanged during augmentation.

To maximize the conditional mutual information (4), we need to compute the posterior distributions $P^*(\theta | \{\mathcal{T}_i^{\mathrm{cri}}\}_{i=1}^{N^{\mathrm{cri}}})$ and $P^*(\theta | \{\bar{\mathcal{T}}_i^{\mathrm{cri}}(\Lambda_i)\}_{i=1}^{N^{\mathrm{cri}}})$. Analogous to (Yin et al., 2019), we treat $\theta$ as a random variable where the randomness comes from the training stochasticity. Mathematically, the posterior distributions are (the derivation is in Appendix E):

$$
P^*(\cdot | \{\bar{\mathcal{T}}_i^{\mathrm{cri}}(\Lambda_i)\}_{i=1}^{N^{\mathrm{cri}}}) = \arg\max_\phi E_{p_\phi(\theta)} \left[ L(\theta, \{\bar{\mathcal{T}}_i^{\mathrm{cri}}(\Lambda_i)\}_{i=1}^{N^{\mathrm{cri}}}, \{\mathcal{T}_i^{\mathrm{tr}}\}_{i=1}^{N^{\mathrm{tr}} - N^{\mathrm{cri}}}) \right],
$$

$$P^*(\cdot|\{\mathcal{T}_i^{\mathrm{cri}}\}_{i=1}^{N^{\mathrm{cri}}}) = E_{\Lambda_i \in [0,1], \Lambda_i \sim P_{\phi_\lambda}(\lambda)}\left[P^*(\cdot|\{\bar{\mathcal{T}}_i^{\mathrm{cri}}(\Lambda_i)\}_{i=1}^{N^{\mathrm{cri}}})\right], \tag{5}$$

where $P_\phi(\theta)$ is a distribution of $\theta$ parameterized by $\phi$. The posterior $P^*(\cdot|\{\bar{\mathcal{T}}_i^{\mathrm{cri}}(\Lambda_i)\}_{i=1}^{N^{\mathrm{cri}}})$ can be straightforwardly obtained by optimizing the meta-objective (3) because the meta-parameter $\theta$ is directly trained over the augmented tasks. However, the posterior $P^*(\cdot|\{\mathcal{T}_i^{\mathrm{cri}}\}_{i=1}^{N^{\mathrm{cri}}})$ cannot be directly computed because the meta-parameter $\theta$ is no longer trained on the vanilla training tasks (without augmentation). To obtain the posterior $P^*(\cdot|\{\mathcal{T}_i^{\mathrm{cri}}\}_{i=1}^{N^{\mathrm{cri}}})$, we need to marginalize over all the possible augmentations and thus it is an expectation of $P^*(\cdot|\{\bar{\mathcal{T}}_i^{\mathrm{cri}}(\Lambda_i)\}_{i=1}^{N^{\mathrm{cri}}})$ over all possible mixing coefficient $\{\Lambda_i\}_{i=1}^{N^{\mathrm{cri}}}$. By combining (4) and (5), we reach the bi-level optimization problem:

$$\max_{\phi_\lambda} \ I(\theta; \{\bar{\mathcal{T}}_i^{\mathrm{cri}}(\Lambda_i \sim P_{\phi_\lambda}(\lambda))\}_{i=1}^{N^{\mathrm{cri}}}|\{\mathcal{T}_i^{\mathrm{cri}}\}_{i=1}^{N^{\mathrm{cri}}}), \quad \text{s.t.} \quad \text{Problem (5).} \tag{6}$$

The bi-level optimization in (6) captures the interaction between choosing good augmentations and evaluating how those augmentations influence the meta-parameter. The upper level optimizes the augmentation distribution $P_{\phi_\lambda}(\lambda)$ to find mixing coefficients that make the augmented critical tasks contain as much additional information about the meta-parameter as possible. However, evaluating how informative an augmentation is requires knowing how the belief over $\theta$ changes after observing the augmented tasks. This is handled by the lower-level problem, which computes the posterior distributions over $\theta$ under both the augmented critical tasks and the original tasks. In essence, the lower level tells us what the learner would believe for a given augmentation, and the upper level uses this feedback to adjust the augmentation distribution in order to maximize information gain. Thus, the bi-level structure reflects that we must jointly learn (i) how to mix the tasks and (ii) how these mixed tasks influence the meta-parameter distribution.

## 4.2 Algorithm

In this section, we develop an algorithm to improve the generalization of MRL. The algorithm consists of two major stages: identifying critical tasks and solving the bi-level optimization problem in (6) to improve generalization. We elaborate the algorithm as follows:

---

**Algorithm 1** Explainable meta reinforcement learning to improve generalization (XMRL)

---

**Input**: Initial mixing coefficient distribution $P_{\phi_{\lambda,0}}(\lambda)$ and meta-parameter distribution $P_{\phi_0}(\theta)$, training tasks $\{\mathcal{T}_i^{\mathrm{tr}}\}_{i=1}^{N^{\mathrm{tr}}}$, and poorly adapted tasks $\{\mathcal{T}_i^{\mathrm{poor}}\}_{i=1}^{N^{\mathrm{poor}}}$.
**Output**: Learned mixing coefficient distribution $P_{\phi_{\lambda,K}}(\lambda)$ and meta-parameter distribution $P_{\phi^*(\{\lambda_{i,K}\}_{i=1}^{N^{\mathrm{cri}}})}(\theta)$.

1: Generate the explanation (i.e., the critical tasks $\{\mathcal{T}_i^{\mathrm{cri}}\}_{i=1}^{N^{\mathrm{cri}}}$) using the algorithm in Appendix B.
2: **for** $k = 0, \cdots, K-1$ **do**
3:     **Lower-level optimization**: Sample $N^{\bar{\zeta}}$ sets of coefficients $\{\lambda_{i,k}^{\bar{\zeta}_j}\}_{i=1}^{N^{\mathrm{cri}}}$ from $P_{\phi_{\lambda,k}}(\lambda)$; for each
    set, compute the distribution parameter $\phi^*(\{\lambda_{i,k}^{\bar{\zeta}_j}\}_{i=1}^{N^{\mathrm{cri}}})$ such that $P^*(\theta|\{\bar{\mathcal{T}}_i^{\mathrm{cri}}(\lambda_{i,k}^{\bar{\zeta}_j})\}_{i=1}^{N^{\mathrm{cri}}}) = P_{\phi^*(\{\lambda_{i,k}^{\bar{\zeta}_j}\}_{i=1}^{N^{\mathrm{cri}}})}(\theta)$. Estimate $P^*(\cdot|\{\mathcal{T}_i^{\mathrm{cri}}\}_{i=1}^{N^{\mathrm{cri}}}) = \frac{1}{N^{\bar{\zeta}}}\sum_{j=1}^{N^{\bar{\zeta}}} P^*(\cdot|\{\bar{\mathcal{T}}_i^{\mathrm{cri}}(\lambda_{i,k}^{\bar{\zeta}_j})\}_{i=1}^{N^{\mathrm{cri}}})$.
4:     **Upper-level optimization**: Compute the hyper-gradient $g_{\phi_{\lambda,k}}$ in Lemma 1 and update the
    mixing coefficient distribution parameter $\phi_{\lambda,k+1} = \phi_{\lambda,k} + \beta g_{\phi_{\lambda,k}}$.
5: **end for**

---

**Stage 1: Identify critical tasks (line 1 in Algorithm 1)**. We begin by generating explanations for the original meta-policy $\pi_0$ and extracting the critical tasks $\{\mathcal{T}_i^{\mathrm{cri}}\}_{i=1}^{N^{\mathrm{cri}}}$ using the algorithm in Appendix B. These tasks are the ones to which the meta-policy $\pi_0$ should pay more attention, and therefore they serve as the targets for augmentation.

**Stage 2: Solve the bi-level optimization problem (6) to augment critical tasks**. Once the critical tasks are identified, the algorithm solves the bi-level optimization problem (6) via lower-level and upper-level optimization. We elaborate the lower-level and upper-level optimization as follows:

**Lower-level optimization (line 3).** At iteration $k$, the lower-level step is to compute the posterior distribution over $\theta$ under both the augmented critical tasks and the original critical tasks. To obtain the posterior $P^*(\theta|\{\bar{\mathcal{T}}_i^{\mathrm{cri}}(\lambda_i)\}_{i=1}^{N^{\mathrm{cri}}})$ over the augmented critical tasks, we solve the optimization problem in (5 via gradient ascent. Specifically, we parameterize $P_\phi(\theta)$ as a Gaussian distribution whose parameter $\phi = (\mu, \Sigma)$ includes a mean vector $\mu$ and a covariance matrix $\Sigma = \sigma\sigma^\top$. We reparameterize $\theta$ as $\theta = \mu + \sigma \circ \zeta$ where $\zeta \sim \mathcal{N}(0, I)$ denotes a standard Gaussian distribution and $\circ$ is component-wise multiplication. The gradient of the lower-level optimization problem is $E_{\zeta \sim \mathcal{N}(0,I)}[\nabla_\phi \theta \cdot \nabla_\theta L(\theta, \{\bar{\mathcal{T}}_i^{\mathrm{cri}}(\lambda_i)\}_{i=1}^{N^{\mathrm{cri}}}, \{\mathcal{T}_i^{\mathrm{tr}}\}_{i=1}^{N^{\mathrm{tr}}-N^{\mathrm{cri}}})]$ and we can use $N^\zeta$ samples $\zeta_j \sim \mathcal{N}(0, I)$ to estimate the gradient: $g_\phi = \frac{1}{N^\zeta} \sum_{j=1}^{N^\zeta} \nabla_\phi \theta_j \cdot \nabla_\theta L(\theta_j, \{\bar{\mathcal{T}}_i^{\mathrm{cri}}(\lambda_i)\}_{i=1}^{N^{\mathrm{cri}}}, \{\mathcal{T}_i^{\mathrm{tr}}\}_{i=1}^{N^{\mathrm{tr}}-N^{\mathrm{cri}}})$ where $\theta_j = \mu + \sigma \circ \zeta_j$. We include the detailed expressions of the gradient in Appendix G.1. Recall that the posterior distribution $P^*(\theta|\{\mathcal{T}_i^{\mathrm{cri}}\}_{i=1}^{N^{\mathrm{cri}}})$ is obtained by marginalizing $P^*(\theta|\{\bar{\mathcal{T}}_i^{\mathrm{cri}}(\lambda_i)\}_{i=1}^{N^{\mathrm{cri}}})$ over $\{\lambda_i\}_{i=1}^{N^{\mathrm{cri}}}$. Therefore, we sample $N^{\bar\zeta}$ sets of mixing coefficients $\{\{\lambda_{i,k}^{\bar\zeta_j}\}_{i=1}^{N^{\mathrm{cri}}}\}_{\bar\zeta_j=1}^{N^{\bar\zeta}}$ and compute the posterior distribution $P^*(\cdot|\{\bar{\mathcal{T}}_i^{\mathrm{cri}}(\lambda_{i,k}^{\bar\zeta_j})\}_{i=1}^{N^{\mathrm{cri}}})$ corresponding to each set. The posterior under the original critical tasks can be estimated by $P^*(\theta|\{\mathcal{T}_i^{\mathrm{cri}}\}_{i=1}^{N^{\mathrm{cri}}}) = \frac{1}{N^{\bar\zeta}} \sum_{j=1}^{N^{\bar\zeta}} P^*(\theta|\{\bar{\mathcal{T}}_i^{\mathrm{cri}}(\lambda_i^{\bar\zeta_j})\}_{i=1}^{N^{\mathrm{cri}}})$.

**Upper-level optimization** (line 4). We use gradient ascent to solve the upper-level problem. In order to compute the the hyper-gradient, i.e., the gradient of the conditional mutual information (4) w.r.t. $\phi_\lambda$, we use a Gaussian distribution to parameterize $P_{\phi_\lambda}(\lambda)$ where the distribution parameter $\phi_\lambda = (\mu_\lambda, \sigma_\lambda)$ includes a mean $\mu_\lambda$ and a standard deviation $\sigma_\lambda$. Therefore, we can reparameterize each sample $\lambda_i^{\bar\zeta_j}$ from $P_{\phi_\lambda}(\lambda)$ via $\lambda_i^{\bar\zeta_j} = \mu_\lambda + \sigma_\lambda \bar\zeta_{i,j}$ where $\bar\zeta_{i,j} \sim \mathcal{N}(0, 1)$.

**Lemma 1.** *Suppose we reparameterize the mixing coefficient $\lambda_i^{\bar\zeta_j}$ via $\lambda_i^{\bar\zeta_j} = \mu_\lambda + \sigma_\lambda \bar\zeta_{i,j}$, the hyper-gradient can be estimated by* $g_{\phi_\lambda} = \frac{\sum_{j=1}^{N^{\bar\zeta}} \nabla_{\phi_\lambda} \sigma^*(\{\lambda_i^{\bar\zeta_j}\}_{i=1}^{N^{\mathrm{cri}}})}{|| \sum_{j=1}^{N^{\bar\zeta}} \sigma^*(\{\lambda_i^{\bar\zeta_j}\}_{i=1}^{N^{\mathrm{cri}}})||} - \frac{1}{N^{\bar\zeta}} \sum_{j=1}^{N^{\bar\zeta}} \frac{\nabla_{\phi_\lambda} \sigma^*(\{\lambda_i^{\bar\zeta_j}\}_{i=1}^{N^{\mathrm{cri}}})}{||\sigma^*(\{\lambda_i^{\bar\zeta_j}\}_{i=1}^{N^{\mathrm{cri}}})||}$ *where* $\nabla_{\phi_\lambda} \sigma^*(\{\lambda_i^{\bar\zeta_j}\}_{i=1}^{N^{\mathrm{cri}}}) = -\left[\nabla_{\sigma\sigma}^2 E_{P_{\phi^*}(\theta)}[L(\theta, \{\bar{\mathcal{T}}_i^{\mathrm{cri}}(\lambda_i^{\bar\zeta_j})\}_{i=1}^{N^{\mathrm{cri}}}, \{\mathcal{T}_i^{\mathrm{tr}}\}_{i=1}^{N^{\mathrm{tr}}-N^{\mathrm{cri}}})]\right]^{-1} \cdot \nabla_{\sigma\phi_\lambda}^2 E_{P_{\phi^*}(\theta)}[L(\theta, \{\bar{\mathcal{T}}_i^{\mathrm{cri}}(\lambda_i^{\bar\zeta_j})\}_{i=1}^{N^{\mathrm{cri}}}, \{\mathcal{T}_i^{\mathrm{tr}}\}_{i=1}^{N^{\mathrm{tr}}-N^{\mathrm{cri}}})].$

We include the expression of all the gradients in Appendix G. We solve the upper-level problem in (6) via gradient ascent $\phi_{\lambda,k+1} = \phi_{\lambda,k} + \beta g_{\phi_{\lambda,k}}$ where $\beta$ is the step size.

## 4.3 THEORETICAL ANALYSIS

This part shows that (i) Algorithm 1 converges at the rate of $O(1/\sqrt{K})$, (ii) the learned augmentation increases the task information of the critical tasks stored in the meta-policy, and (iii) the generalization over the task distribution improves after the augmentation. We start with the assumption:

**Assumption 1.** *The parameterized meta-policy $\pi_\theta$ satisfies the following:* $||\nabla_\theta \log \pi_\theta(a|s)|| \leq C_\theta$ *and* $||\nabla_{\theta\theta}^2 \log \pi_\theta(a|s)|| \leq \bar{C}_\theta$ *for any* $(s, a) \in \mathcal{S} \times \mathcal{A}$ *where $C_\theta$ and $\bar{C}_\theta$ are positive constants.*

Assumption 1 assumes that the parameterized log-policy $\log \pi_\theta$ is $C_\theta$-Lipschitz continuous and $\bar{C}_\theta$-smooth w.r.t. the parameter $\theta$, which is a standard assumption in RL (Liu & Zhu, 2022; Kumar et al., 2023; Liu & Zhu, 2023; Agarwal et al., 2021; Liu & Zhu, 2024a;b; Liu et al., 2025a).

**Theorem 1.** *Suppose Assumption 1 holds and we choose the step size $\beta = \frac{2}{\bar{C}_I \sqrt{K}}$ where $\bar{C}_I$ is a positive constant whose derivation is in Appendix H, then Algorithm 1 converges:* $\frac{1}{K} \sum_{k=0}^{K-1} ||\nabla_{\phi_\lambda} I(\theta; \{\bar{\mathcal{T}}_i^{cri}(\lambda_i \sim P_{\phi_\lambda,k}(\lambda))\}_{i=1}^{N^{cri}}|\{\mathcal{T}_i^{cri}\}_{i=1}^{N^{cri}})||^2 \leq O(1/\sqrt{K}).$

Theorem 1 guarantees that Algorithm 1 converges at the rate of $O(1/\sqrt{K})$. We next show that the learned task augmentation stores more information in the meta-parameter:

**Theorem 2.** *Suppose Assumption 1 holds and $\beta < \frac{2}{\bar{C}_I}$, then the output $P_{\phi_\lambda,K}(\lambda)$ of Algorithm 1 satisfies* $I(\theta; \{\bar{\mathcal{T}}_i^{cri}(\lambda_i \sim P_{\phi_\lambda,K}(\lambda))\}_{i=1}^{N^{cri}}|\{\mathcal{T}_i^{cri}\}_{i=1}^{N^{cri}}) > 0.$

Theorem 2 guarantees that the augmented critical tasks store additional information in the meta-parameter. Moreover, Appendix J guarantees that the task information of the non-critical tasks stored in the meta-parameter does not change even if the stored task information of the critical tasks

increases. We next quantify the generalization improvement of the learned augmentation $P_{\phi_{\lambda,K}}(\lambda)$. In particular, we first demonstrate that the learned augmentation imposes a quadratic regularization on the meta-parameter $\theta$ in Lemma 2 and then guarantee that the generalization over the task distribution $P(\mathcal{T})$ improves.

To reason about the generalization, we consider the following softmax parameterized meta-policy $\pi_\theta(a|s) = \frac{e^{\theta^\top f(s,a)}}{\sum_{a' \in \mathcal{A}} e^{\theta^\top f(s,a')}}$ where $f(s,a)$ is a feature vector. This policy parameterization is widely adopted in RL (Sutton et al., 1999; Kakade, 2001; Peters & Schaal, 2008). We consider MAML (Finn et al., 2017; Fallah et al., 2021) as the algorithm to compute the task-specific adaptation $\pi_i^{\mathrm{tr}}(\theta)$, and the task-specific adaptation is also softmax parameterized.

**Lemma 2.** *The second-order approximation of the meta-objective (3) after the task augmentation can be expressed as:* $E_{\lambda_i \sim P_{\phi_{\lambda,K}}}[L(\theta, \{\bar{\mathcal{T}}_i^{\mathrm{cri}}(\lambda_i)\}_{i=1}^{N^{\mathrm{cri}}}, \{\mathcal{T}_i^{\mathrm{tr}}\}_{i=1}^{N^{\mathrm{tr}}-N^{\mathrm{cri}}})] \approx L(\theta, \{\mathcal{T}_i^{\mathrm{tr}}\}_{i=1}^{N^{\mathrm{tr}}}) - \theta^\top(\frac{1}{N^{\mathrm{cri}}} \sum_{i=1}^{N^{\mathrm{cri}}} \bar{H}_i^{\mathrm{cri}})\theta$ *where $\bar{H}_i^{\mathrm{cri}}$ is a positive definite matrix whose expression is in Appendix K.*

Lemma 2 shows that the augmented meta-objective (3) imposes a quadratic regularization on the original meta-objective (1). Since we aim to maximize the meta-objective, this negative quadratic regularization reduces the solution space and thus can lead to better generalization.

To study the generalization property of this regularization, following (Zhang & Deng, 2021; Yao et al., 2021), we consider the following softmax policy class that is closely related to the dual problem of the regularization: $\mathcal{F}_{\bar{\gamma}} = \{\pi_\theta : \theta^\top(E_{i \sim P(\mathcal{T})}[\bar{H}_i])\theta \le \bar{\gamma}\}$. To quantify the improvement of generalization, we denote the generalization gap by $\mathcal{G}(\mathcal{F}_{\bar{\gamma}}) \triangleq L(\theta, \{\mathcal{T}_i^{\mathrm{tr}}\}_{i=1}^{N^{\mathrm{tr}}}) - E_{i \sim P(\mathcal{T})}[L(\theta, \mathcal{T}_i)]$. The following theorem validates improvement of generalization:

**Theorem 3.** *Suppose the policy is softmax parameterized (i.e., $\pi_\theta(a|s) = \frac{e^{\theta^\top f(s,a)}}{\sum_{a' \in \mathcal{A}} e^{\theta^\top f(s,a')}}$) where the feature vector $f(s,a)$ is twice-differentiable and bounded for any $(s,a) \in \mathcal{S} \times \mathcal{A}$, then with probability at least $1 - \delta$, the generalization gap satisfies $|\mathcal{G}(\mathcal{F}_{\bar{\gamma}})| \le O(\sqrt{\frac{\bar{\gamma}}{N^{\mathrm{tr}}}} + \sqrt{\frac{\log(1/\delta)}{N^{\mathrm{tr}}}})$.*

According to Lemma 2, the quadratic regularization (i.e., $\theta^\top(\frac{1}{N^{\mathrm{cri}}} \sum_{i=1}^{N^{\mathrm{cri}}} \bar{H}_i^{\mathrm{cri}})\theta$) imposed by the learned task augmentation encourages a smaller $\bar{\gamma}$. Therefore, according to Theorem 3, the learned task augmentation will lead to a smaller generalization gap and thus improve generalization.

## 5 EXPERIMENT

This section uses two real-world experiments, two MuJoCo experiments, and a Meta-World experiment (Appendix M.9) to validate the effectiveness of Algorithm 1 (XMRL). *Note that XMRL is to improve the performance of existing MRL algorithms, rather than being a standalone MRL algorithm. Accordingly, our primary comparisons are against other MRL **improvement methods**, rather than against MRL algorithms themselves. For completeness, we report comparisons with some MRL algorithms in Appendix M.9.* We use MAML as the base MRL algorithm that our method and the baselines aim to

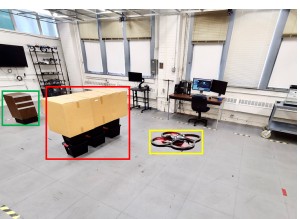

Figure 1: Drone navigation

improve. We introduce three baselines that improve MRL generalization: (1) **Task weighting (TW)** (Cai et al., 2020): This method computes the weighted meta-policy $\pi_{\theta^*(\omega)}$. (2) **Meta augmentation (MA)** (Yao et al., 2021): This method uses a predefined distribution of $\lambda$ to mix the data of each training task to improve generalization. (3) **Meta regularization (MR)** (Wang et al., 2023): This method adds quadratic regularization to the upper level and inverted regularization to the lower level to improve generalization. Combining the base MRL algorithm with each improvement method results in four complete methods: MAML+XMRL, MAML+TW, MAML+MA, and MAML+MR. Note that our method requires additional interactions with the environment to generate augmented samples. For a fair comparison, the baselines use the same amount of samples.

**Experiment I: Drone navigation with obstacles**. We conduct a navigation experiment (Figure 1) on an AR.Drone 2.0 where the drone (yellow bounding box) navigates to the goal (green bounding box) while avoiding the obstacle (red bounding box). We use a motion capture system "Vicon" to

Table 1: Experiment results.

| Method | MAML | MAML+XMRL | MAML+TW | MAML+MA | MAML+MR |
|---|---|---|---|---|---|
| Drone | $0.87 \pm 0.01$ | $0.97 \pm 0.01$ | $0.87 \pm 0.02$ | $0.91 \pm 0.02$ | $0.91 \pm 0.02$ |
| Stock Market | $359.13 \pm 18.63$ | $421.13 \pm 12.11$ | $362.07 \pm 14.21$ | $389.17 \pm 12.66$ | $362.53 \pm 14.27$ |
| HalfCheetah | $-68.89 \pm 4.36$ | $-44.67 \pm 4.35$ | $-65.14 \pm 4.26$ | $-63.49 \pm 4.07$ | $-61.15 \pm 3.82$ |
| Ant | $100.64 \pm 3.63$ | $119.15 \pm 4.02$ | $99.92 \pm 4.56$ | $106.44 \pm 4.55$ | $104.15 \pm 4.74$ |

record the location of the drone. For different navigation tasks, we change the locations of the goal and obstacle. We use success rate (i.e., the rate of successfully reaching the goal and avoiding the obstacle) over randomly generated test tasks as the metric to evaluate generalization performance. We record the mean and standard deviation of success rate in the second row in Table 1.

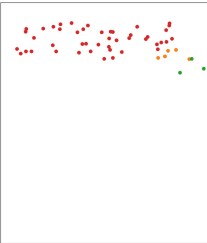

We sample 50 training tasks to learn a meta-policy. We then sample 10 tasks and find the top 3 tasks where the meta-policy adapts with the worst performance as poorly adapted tasks. In Figure 2, the red points and orange points are the goals of the 50 training tasks. The green points are the goals of the top 3 poorly adapted test tasks. The five orange points are the identified critical training tasks. We can see that the green points are far from the red points and thus are poorly adapted. The identified critical tasks are the training tasks whose goals are closest to the goals of the poorly adapted test tasks.

Figure 2: Task visualization

**Experiment II: Stock market**. RL to train a stock trading agent has been widely studied in AI for finance (Deng et al., 2016). We use the real-world data of 30 constituent stocks in Dow Jones Industrial Average from 2021-01-01 to 2022-01-01. We use "FinRL" (Liu et al., 2021) to configure the real-world stock data into an MDP environment. We include the details in Appendix M.5 and the results of cumulative reward in the third row in Table 1.

**Experiment III: MuJoCo**. We consider the target velocity problem (Finn et al., 2017) for two MuJoCo robots: HalfCheetah and Ant. In particular, the robots aim to maintain a target velocity in each task, and the target velocity of different tasks is different. We include the details in Appendix M.6 and the results of cumulative reward in the fourth and fifth rows in Table 1.

Table 1 shows that our proposed method can significantly improve the generalization of MAML and outperform the other three baselines. For example, our method improves MAML by 35% for HalfCheetah while the baselines' improvements are less than 15%.

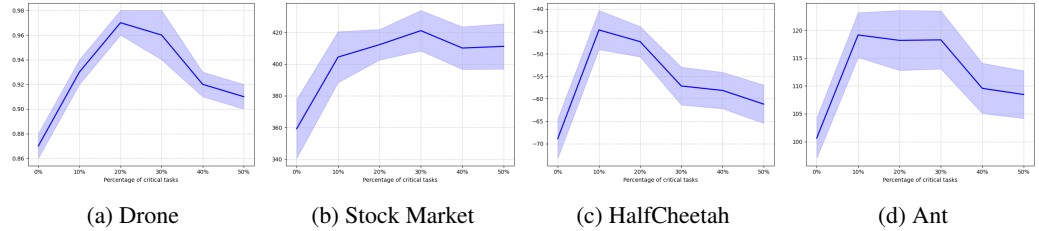

| (a) Drone | (b) Stock Market | (c) HalfCheetah | (d) Ant |
|---|---|---|---|

Figure 3: Ablation study on the number of critical tasks.

**The ablation study on the number of critical tasks** $N^{\text{cri}}$. We evaluate how the number of critical tasks $N^{\text{cri}}$ affects the generalization performance of our method. We vary the number of critical tasks and plot the corresponding generalization performance in Figure 3. The x-axis is the percentage of critical tasks in training tasks. Figure 3 shows that the optimal percentage of critical tasks is 10% for drone, 30% for Stock Market, and 10% for HalfCheetah and Ant. Moreover, the generalization performance becomes worse if the number of critical tasks is too large. The reason is that at this time, the critical tasks can include training tasks that are not helpful to improve poorly adapted tasks. If we still include those non-helpful tasks as critical tasks, those non-helpful tasks may prevent the meta-policy from generalizing well to tasks similar to the poorly adapted tasks. Note that even if augmenting the "false positives" tasks can lead to a decline of generalization performance, the generalization performance is still better than MAML where no tasks are augmented.

**Minor degradation on very few tasks**. We conduct an evaluation to separately report (i) the performance on the original poorly adapted tasks, (ii) the performance on the original non-poorly adapted tasks, (iii) the percentage of tasks with degradation in the original non-poorly adapted tasks, and (iv) performance drop for degraded tasks. For example, if the performance of a non-poorly adapted task drops from 100 to 90, then the performance drop is 10%. We report the average performance drop over all degraded tasks in Table 2. The results demonstrate that (i) the average performance on the poorly adapted tasks significantly improves, (ii) the average performance on the non-poorly adapted tasks does not degrade, (iii) only a very small portion (less or equal to 5%) of non-poorly adapted tasks degrade, (iv) even for degraded tasks, the performance drop is minor (less than 4%).

Table 2: Ablation on performance improvement and performance degradation

|  |  | MAML | Our method (MAML+XMRL) |
|---|---|---|---|
| Drone | poorly adapted tasks | 0.55 | 0.93 |
|  | non-poorly adapted tasks | 0.95 | 0.98 |
|  | percentage of degradation | N/A | 0% |
|  | performance drop for degraded tasks | N/A | N/A |
| Stock market | poorly adapted tasks | 71.05 | 381.33 |
|  | non-poorly adapted tasks | 431.15 | 431.08 |
|  | percentage of degradation | N/A | 5% |
|  | performance drop for degraded tasks | N/A | 3.8% |
| HalfCheetah | poorly adapted tasks | -162.09 | -55.00 |
|  | non-poorly adapted tasks | -45.59 | -42.10 |
|  | percentage of degradation | N/A | 2.5% |
|  | performance drop for degraded tasks | N/A | 2.7% |
| Ant | poorly adapted tasks | 39.68 | 99.67 |
|  | non-poorly adapted tasks | 115.88 | 124.02 |
|  | percentage of degradation | N/A | 5% |
|  | performance drop for degraded tasks | N/A | 2.0% |

**The ablation study on augmentation method**. We aim to learn an optimal augmentation by optimizing this distribution $P(\lambda)$ to maximize the conditional mutual information (4). To show the effectiveness of our method, we include an ablation study in Appendix M.7 where we compare to a method that uses the predefined distribution in (Wang et al., 2020) to augment the critical tasks. The results in Appendix M.7 demonstrate that our method improves generalization better.

**Evaluation of the explanation**. We evaluate the fidelity and usefulness of our explanation. Fidelity is (Guo et al., 2021b; Cheng et al., 2024) to evaluate the correctness of the explanation. The fidelity in our setting means whether the identified critical tasks $\{\mathcal{T}_i^{\mathrm{cri}}\}_{i=1}^{N^{\mathrm{cri}}}$ are indeed the most important training tasks to achieve high cumulative reward on the poorly adapted tasks $\{\mathcal{T}_i^{\mathrm{poor}}\}_{i=1}^{N^{\mathrm{poor}}}$. To evaluate fidelity, we train a meta-policy over the critical tasks and compare its performance on the poorly adapted tasks with a meta-policy trained on $N^{\mathrm{cri}}$ randomly-sampled training tasks. The usefulness means whether our explanation helps improve generalization. To evaluate the usefulness, we randomly pick $N^{\mathrm{cri}}$ training tasks and use our augmentation method to augment these $N^{\mathrm{cri}}$ training tasks to train a meta-policy. We compare the generalization performance of this meta-policy with XMRL. The results in Appendix M.8 show that our explanation has high fidelity and usefulness.

## 6 CONCLUSION

This paper proposes to leverage explanation to improve generalization of MRL. The proposed method has two parts where the first part explains why the learned meta-policy does not adapt well to certain tasks by identifying the critical training tasks that the meta-policy does not pay enough attention to, and the second part formulates a bi-level optimization problem to learn how to augment the critical tasks such that the meta-policy can best pay attention to the critical tasks. We theoretically guarantee that the learned augmentation can improve generalization over the whole task distribution. Experimental results validate that our method improves MRL.

## 7 ACKNOWLEDGEMENTS

This work is partially supported by the National Science Foundation through grants ECCS 1846706 and ECCS 2140175. We would like to thank the reviewers for their insightful and constructive suggestions.

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

## A  RELATED WORKS

**Meta-Reinforcement learning**. MRL has two major categories: optimization-based MRL and black-box (or context-based) MRL. Optimization-based MRL (Finn et al., 2017; Liu et al., 2019; Stadie et al., 2018) usually includes a meta-algorithm and an adaptation algorithm. The meta-algorithm learns a meta-prior (e.g., meta-policy) which is not specialized for each task. The adaptation algorithm uses data of a specific task to specialize the meta-prior to achieve high cumulative reward on this specific task. Black-box MRL (Duan et al., 2016; Rakelly et al., 2019; Zintgraf et al., 2019b) typically learns an end-to-end model that contains specialized knowledge for different tasks in the task distribution. The data of a task is used to indicate the task so that the end-to-end model can directly specialize to this specific task. Note that the method proposed in this paper is not classified as a MRL method. Instead, we aim to develop a method that can improve MRL and this method can be applied to different MRL algorithms.

**Meta-learning generalization improvement**. There are three major ways to improve meta-learning generalization: task weighting, regularization, and meta-augmentation. Task weighting (Nguyen et al., 2023; Yao et al., 2021; Cai et al., 2020) proposes to re-weight the training tasks or reshape the training task distribution to improve generalization. However, (Yao et al., 2021; Cai et al., 2020) require an additional target task set to guide how to weight the training tasks or reshape the training task distribution, and thus the learned meta-prior can be biased towards the target task set and may not adapt well to other tasks that are different from the target task set. Regularization-based methods are also used to improve generalization where (Wang et al., 2023) proposes to add ordinary regularization to the upper level and inverted regularization to the lower level, and (Yin et al., 2019) imposes regularization to prevent memorization overfitting. The most relevant technique to our paper is meta-augmentation which augments the data and train on the augmented data to improve generalization. Specifically, (Rajendran et al., 2020) proposes to add noise to the data and (Yao et al., 2021) proposes to mix data and shuffle the channels in the hidden layers. The augmentation method has also been used in RL (Wang et al., 2020; Laskin et al., 2020) to improve generalization by generating augmented data. These augmentation methods use predefined rules to provide feasible

augmentations. In contrast, our paper formulates a bilevel optimization problem to learn how to best augment the critical tasks.

**Explainable reinforcement learning**. While it lacks research works on explainable MRL, explainable RL (XRL) has been extensively studied to explain the decision making of the RL agents, including learning an interpretable policy (Bastani et al., 2018; Bewley & Lawry, 2021; Verma et al., 2018), pinpointing regions in the observations that are critical for choosing certain actions (Atrey et al., 2019; Guo et al., 2021a; Puri et al., 2019), and reward decomposition (Juozapaitis et al., 2019; Lin et al., 2020; Septon et al., 2023). The most relevant XRL method to our explanation is to identify the critical states that are influential to the cumulative reward as an explanation (Guo et al., 2021b; Cheng et al., 2024; Amir & Amir, 2018) where they respectively use an RNN, masks, and a self-proposed rule to find critical states. In contrast, we formulate a bilevel optimization problem to learn a weight vector that indicates critical tasks.

**Explainable meta-learning**. There are three works on explainable meta-learning where (Woźnica & Biecek, 2021) proposes to learn important features that lead to a specific meta model decision using Friedman's H-statistic (Friedman & Popescu, 2008), and (Shao et al., 2022; 2023) use structural causal model to model the causal relations between the features and the model decision. While these works explain why a decision is made, we explain why certain tasks are poorly adapted.

**Task selection of meta learning**. Task selection has been studied in meta-learning. (Zhan & Anderson, 2024; Chen et al., 2022) focus on task efficiency and propose to select a subset of training tasks that are most representative or informative so that training on fewer tasks (the selected subset) can achieve performance comparable to training on the whole training tasks. In contrast, our method has a different focus, i.e., it does not aim to used fewer training tasks to achieve similar performance but aims to select and focus on critical tasks so as to improve performance. (Luna Gutierrez & Leonetti, 2020; Zhang, 2024) propose to select training tasks such that the meta-prior can perform the best on a certain test task. However, they assume the ability to evaluate on a test set. In contrast, our method does not assume access to a test set and our goal is not to perform well on a specific test set.

**Unsupervised environment design**. Unsupervised environment design (Jiang et al., 2021; Jackson et al., 2023) adaptively and progessively generates more complex tasks for the agent to solve in order to improve the agent's generalization ability. However, our paper does not generate or require new tasks to improve generalization. In contrast, we identify "critical tasks" from the existing training tasks and augment these critical tasks to improve generalization.

## B ALGORITHM TO FIND THE CRITICAL TASKS

Recall from Section 3 that we aim to learn a weight vector $\omega$ by solving the problem (2) where each component $\omega_i$ of the weight vector captures the importance of the corresponding training task $\mathcal{T}_i^{\text{tr}}$. The higher the weight value $\omega_i$ is, the more important the corresponding training task $\mathcal{T}_i^{\text{tr}}$ is. Therefore, the top $N^{\text{cri}}$ training tasks with highest weight values are the $N^{\text{cri}}$ critical tasks we aim to identify. The problem (2) is as follows:

$$\max_{\omega} \ L(\theta^*(\omega), \{\mathcal{T}_i^{\text{poor}}\}_{i=1}^{N^{\text{poor}}}) \quad \text{s.t. } \theta^*(\omega) = \arg\max_{\theta} \sum_{i=1}^{N^{\text{tr}}} \omega_i J_i^{\text{tr}}(\pi_i^{\text{tr}}(\theta)).$$

We use Algorithm 2 to solve this problem where at each iteration $\bar{k}$, we first solve the lower-level problem in (2) to get $\theta^*(\omega)$ and then solve the upper-level problem (2) via gradient ascent.

---

**Algorithm 2** Identifying the critical tasks

---

**Input**: Training tasks $\{\mathcal{T}_i^{\text{tr}}\}_{i=1}^{N^{\text{tr}}}$, poorly adapted tasks $\{\mathcal{T}_i^{\text{poor}}\}_{i=1}^{N^{\text{poor}}}$, and initial weight vector $\omega$.
**Output**: Learned weight vector $\omega_{\bar{K}}$.
1: **for** $\bar{k} = 0, \cdots, \bar{K} - 1$ **do**
2:     Solve the lower-level problem via gradient ascent $\theta_{\bar{k}+1} = \theta_{\bar{k}} + \alpha_{\bar{k}} \sum_{i=1}^{N^{\text{tr}}} \omega_i \nabla_\theta J_i^{\text{tr}}(\pi_i^{\text{tr}}(\theta_{\bar{k}}))$.
3:     Compute the hyper-gradient $g_{\omega_{\bar{k}}}$ in Lemma 3 and update the weight $\omega_{\bar{k}+1} = \omega_{\bar{k}} + \alpha'_{\bar{k}} g_{\omega_{\bar{k}}}$.
4: **end for**

---

**Solve the lower-level problem**. We use gradient ascent to solve the lower-level problem where the gradient is $\sum_{i=1}^{N^{\text{tr}}} \omega_i \nabla_\theta J_i^{\text{tr}}(\pi_i^{\text{tr}}(\theta))$ and the expression of $\nabla_\theta J_i^{\text{tr}}(\pi_i^{\text{tr}}(\theta))$ can be found in Appendix G.1.

**Solve the upper-level problem**. To solve the upper-level problem, we need to compute the hyper-gradient $g_\omega$.

**Lemma 3.** *The hyper-gradient is:*

$$g_\omega =$$
$$- \left[ \nabla_\omega \sum_{i=1}^{N^{\text{tr}}} \omega_i \nabla_\theta J_i^{\text{tr}}(\pi_i^{\text{tr}}(\theta^*(\omega))) \right] \left[ \sum_{i=1}^{N^{\text{tr}}} \omega_i \nabla_{\theta\theta}^2 J_i^{\text{tr}}(\pi_i^{\text{tr}}(\theta^*(\omega))) \right]^{-1} \left[ \sum_{i=1}^{N^{\text{poor}}} \nabla_\theta J_i^{\text{poor}}(\pi_i^{\text{poor}}(\theta^*(\omega))) \right],$$

*where the derivation is in Appendix B.1.*

In practice, we do not have $\theta^*(\omega_{\bar{k}})$ and thus we use the approximation $\theta_{\bar{k}}$ to approximate the hyper-gradient $g_{\omega_{\bar{k}}} \approx - \left[ \nabla_\omega \sum_{i=1}^{N^{\text{tr}}} \omega_i \nabla_\theta J_i^{\text{tr}}(\pi_i^{\text{tr}}(\theta_{\bar{k}})) \right] \left[ \sum_{i=1}^{N^{\text{tr}}} \omega_i \nabla_{\theta\theta}^2 J_i^{\text{tr}}(\pi_i^{\text{tr}}(\theta_{\bar{k}})) \right]^{-1} \left[ \sum_{i=1}^{N^{\text{poor}}} \nabla_\theta J_i^{\text{poor}}(\pi_i^{\text{poor}}(\theta_{\bar{k}})) \right]$.

We use $\bar{K}$-step gradient ascent $\omega_{\bar{k}+1} = \omega_{\bar{k}} + \alpha_{\bar{k}} g_{\omega_{\bar{k}}}$ to solve the problem (2) to get the learned weight $\omega_{\bar{K}}$. Each component $\omega_{\bar{K},i}$ captures the importance of the corresponding training task $\mathcal{T}_i^{\text{tr}}$. We pick the top $N^{\text{cri}}$ training tasks with the highest weight value as the critical tasks.

### B.1  PROOF OF LEMMA 3

Since $\theta^*(\omega) = \arg\max_\theta \sum_{i=1}^{N^{\text{tr}}} \omega_i J_i^{\text{tr}}(\pi_i^{\text{tr}}(\theta))$, then $\nabla_\theta \sum_{i=1}^{N^{\text{tr}}} \omega_i J_i^{\text{tr}}(\pi_i^{\text{tr}}(\theta^*(\omega))) = 0$. Take gradient w.r.t. $\omega$ on both sides, we have that

$$\nabla_{\omega\theta}^2 \sum_{i=1}^{N^{\text{tr}}} \omega_i J_i^{\text{tr}}(\pi_i^{\text{tr}}(\theta^*(\omega))) + \left( \nabla_\omega \theta^*(\omega) \right)^\top \left[ \nabla_{\theta\theta}^2 \sum_{i=1}^{N^{\text{tr}}} \omega_i J_i^{\text{tr}}(\pi_i^{\text{tr}}(\theta^*(\omega))) \right] = 0,$$

$$\Rightarrow \nabla_\omega \theta^*(\omega) = \left[ \nabla_{\theta\theta}^2 \sum_{i=1}^{N^{\text{tr}}} \omega_i J_i^{\text{tr}}(\pi_i^{\text{tr}}(\theta^*(\omega))) \right]^{-1} \left[ \nabla_{\theta\omega}^2 \sum_{i=1}^{N^{\text{tr}}} \omega_i J_i^{\text{tr}}(\pi_i^{\text{tr}}(\theta^*(\omega))) \right]. \quad (7)$$

Therefore, we have that

$$\nabla_\omega L(\theta^*(\omega), \{\mathcal{T}_i^{\text{poor}}\}_{i=1}^{N^{\text{poor}}}) = \left( \nabla_\omega \theta^*(\omega) \right)^\top \nabla_\theta L(\theta^*(\omega), \{\mathcal{T}_i^{\text{poor}}\}_{i=1}^{N^{\text{poor}}}),$$

$$\stackrel{(a)}{=} \left[ \nabla_{\omega\theta}^2 \sum_{i=1}^{N^{\text{tr}}} \omega_i J_i^{\text{tr}}(\pi_i^{\text{tr}}(\theta^*(\omega))) \right] \left[ \nabla_{\theta\theta}^2 \sum_{i=1}^{N^{\text{tr}}} \omega_i J_i^{\text{tr}}(\pi_i^{\text{tr}}(\theta^*(\omega))) \right]^{-1} \nabla_\theta L(\theta^*(\omega), \{\mathcal{T}_i^{\text{poor}}\}_{i=1}^{N^{\text{poor}}}),$$

where $(a)$ follows (7).

## C  EXPRESSION OF THE AUGMENTED STATE-ACTION STATIONARY DISTRIBUTION

This section provides the expression of the augmented stationary state distribution $\bar{\rho}^{\pi,\lambda_i}(\cdot)$ and augmented stationary state-action distribution $\bar{\rho}^{\pi,\lambda_i}(\cdot,\cdot)$. The expression of the augmented stationary state distribution is $\bar{\rho}^{\pi,\lambda_i}(\bar{s}) \triangleq \rho^\pi(\bar{s}) + \int_{s,s'\in\mathcal{S}} \mathbb{1}\{\lambda_i s + (1-\lambda_i)s' = \bar{s}\} \rho^\pi(s)\rho^\pi(s')dsds'$ and the expression of the augmented state-action stationary distribution is $\bar{\rho}^{\pi,\lambda_i}(\bar{s},\bar{a}) = \bar{\rho}^{\pi,\lambda_i}(\bar{s})\pi(\bar{a}|\bar{s})$.

## D  DERIVATION OF THE CONDITIONAL MUTUAL INFORMATION

$$I(\theta; \{\bar{\mathcal{T}}_i^{\text{cri}}(\Lambda_i \sim P(\lambda))\}_{i=1}^{N^{\text{cri}}} | \{\mathcal{T}_i^{\text{cri}}\}_{i=1}^{N^{\text{cri}}}),$$

$$\stackrel{(a)}{=} \int P(\theta, \{\bar{\mathcal{T}}_i^{\mathrm{cri}}(\Lambda_i = \lambda_i)\}_{i=1}^{N^{\mathrm{cri}}}, \{\mathcal{T}_i^{\mathrm{cri}}\}_{i=1}^{N^{\mathrm{cri}}}) \cdot$$

$$\log \frac{P(\theta, \{\bar{\mathcal{T}}_i^{\mathrm{cri}}(\Lambda_i = \lambda_i)\}_{i=1}^{N^{\mathrm{cri}}} | \{\mathcal{T}_i^{\mathrm{cri}}\}_{i=1}^{N^{\mathrm{cri}}})}{P(\theta | \{\mathcal{T}_i^{\mathrm{cri}}\}_{i=1}^{N^{\mathrm{cri}}}) P(\{\bar{\mathcal{T}}_i^{\mathrm{cri}}(\Lambda_i = \lambda_i)\}_{i=1}^{N^{\mathrm{cri}}} | \{\mathcal{T}_i^{\mathrm{cri}}\}_{i=1}^{N^{\mathrm{cri}}})} (d\theta)(d\{\bar{\mathcal{T}}_i^{\mathrm{cri}}(\Lambda_i = \lambda_i)\}_{i=1}^{N^{\mathrm{cri}}})(d\{\mathcal{T}_i^{\mathrm{cri}}\}_{i=1}^{N^{\mathrm{cri}}}),$$

$$\stackrel{(b)}{=} \int P(\theta | \{\bar{\mathcal{T}}_i^{\mathrm{cri}}(\Lambda_i = \lambda_i)\}_{i=1}^{N^{\mathrm{cri}}}, \{\mathcal{T}_i^{\mathrm{cri}}\}_{i=1}^{N^{\mathrm{cri}}}) \cdot P(\{\bar{\mathcal{T}}_i^{\mathrm{cri}}(\Lambda_i = \lambda_i)\}_{i=1}^{N^{\mathrm{cri}}} | \{\mathcal{T}_i^{\mathrm{cri}}\}_{i=1}^{N^{\mathrm{cri}}}) \cdot P(\{\mathcal{T}_i^{\mathrm{cri}}\}_{i=1}^{N^{\mathrm{cri}}}) \cdot$$

$$\log \frac{P(\theta, \{\bar{\mathcal{T}}_i^{\mathrm{cri}}(\Lambda_i = \lambda_i)\}_{i=1}^{N^{\mathrm{cri}}} | \{\mathcal{T}_i^{\mathrm{cri}}\}_{i=1}^{N^{\mathrm{cri}}})}{P(\theta | \{\mathcal{T}_i^{\mathrm{cri}}\}_{i=1}^{N^{\mathrm{cri}}}) P(\{\bar{\mathcal{T}}_i^{\mathrm{cri}}(\Lambda_i = \lambda_i)\}_{i=1}^{N^{\mathrm{cri}}} | \{\mathcal{T}_i^{\mathrm{cri}}\}_{i=1}^{N^{\mathrm{cri}}})} (d\theta)(d\{\bar{\mathcal{T}}_i^{\mathrm{cri}}(\Lambda_i = \lambda_i)\}_{i=1}^{N^{\mathrm{cri}}})(d\{\mathcal{T}_i^{\mathrm{cri}}\}_{i=1}^{N^{\mathrm{cri}}}),$$

$$\stackrel{(c)}{=} \int P(\theta | \{\bar{\mathcal{T}}_i^{\mathrm{cri}}(\Lambda_i = \lambda_i)\}_{i=1}^{N^{\mathrm{cri}}}, \{\mathcal{T}_i^{\mathrm{cri}}\}_{i=1}^{N^{\mathrm{cri}}}) \cdot P(\{\Lambda_i = \lambda_i\}_{i=1}^{N^{\mathrm{cri}}}) \cdot P(\{\mathcal{T}_i^{\mathrm{cri}}\}_{i=1}^{N^{\mathrm{cri}}}) \cdot$$

$$\log \frac{P(\theta, \{\bar{\mathcal{T}}_i^{\mathrm{cri}}(\Lambda_i = \lambda_i)\}_{i=1}^{N^{\mathrm{cri}}} | \{\mathcal{T}_i^{\mathrm{cri}}\}_{i=1}^{N^{\mathrm{cri}}})}{P(\theta | \{\mathcal{T}_i^{\mathrm{cri}}\}_{i=1}^{N^{\mathrm{cri}}}) P(\{\bar{\mathcal{T}}_i^{\mathrm{cri}}(\Lambda_i = \lambda_i)\}_{i=1}^{N^{\mathrm{cri}}} | \{\mathcal{T}_i^{\mathrm{cri}}\}_{i=1}^{N^{\mathrm{cri}}})} (d\theta)(d\{\bar{\mathcal{T}}_i^{\mathrm{cri}}(\Lambda_i = \lambda_i)\}_{i=1}^{N^{\mathrm{cri}}})(d\{\mathcal{T}_i^{\mathrm{cri}}\}_{i=1}^{N^{\mathrm{cri}}}),$$

$$\stackrel{(d)}{=} \int P(\theta | \{\bar{\mathcal{T}}_i^{\mathrm{cri}}(\Lambda_i = \lambda_i)\}_{i=1}^{N^{\mathrm{cri}}}, \{\mathcal{T}_i^{\mathrm{cri}}\}_{i=1}^{N^{\mathrm{cri}}}) \cdot P(\{\Lambda_i = \lambda_i\}_{i=1}^{N^{\mathrm{cri}}}) \cdot P(\{\mathcal{T}_i^{\mathrm{cri}}\}_{i=1}^{N^{\mathrm{cri}}}) \cdot$$

$$\log \frac{P(\theta, \{\bar{\mathcal{T}}_i^{\mathrm{cri}}(\Lambda_i = \lambda_i)\}_{i=1}^{N^{\mathrm{cri}}} | \{\mathcal{T}_i^{\mathrm{cri}}\}_{i=1}^{N^{\mathrm{cri}}})}{P(\theta | \{\mathcal{T}_i^{\mathrm{cri}}\}_{i=1}^{N^{\mathrm{cri}}}) P(\{\bar{\mathcal{T}}_i^{\mathrm{cri}}(\Lambda_i = \lambda_i)\}_{i=1}^{N^{\mathrm{cri}}} | \{\mathcal{T}_i^{\mathrm{cri}}\}_{i=1}^{N^{\mathrm{cri}}})} (d\theta)(d\{\Lambda_i = \lambda_i\}_{i=1}^{N^{\mathrm{cri}}})(d\{\mathcal{T}_i^{\mathrm{cri}}\}_{i=1}^{N^{\mathrm{cri}}}),$$

$$= \int P(\theta | \{\bar{\mathcal{T}}_i^{\mathrm{cri}}(\Lambda_i = \lambda_i)\}_{i=1}^{N^{\mathrm{cri}}}, \{\mathcal{T}_i^{\mathrm{cri}}\}_{i=1}^{N^{\mathrm{cri}}}) P(\{\Lambda_i = \lambda_i\}_{i=1}^{N^{\mathrm{cri}}}) P(\{\mathcal{T}_i^{\mathrm{cri}}\}_{i=1}^{N^{\mathrm{cri}}}) \cdot$$

$$\log \frac{P(\theta | \{\bar{\mathcal{T}}_i^{\mathrm{cri}}(\Lambda_i = \lambda_i)\}_{i=1}^{N^{\mathrm{cri}}}, \{\mathcal{T}_i^{\mathrm{cri}}\}_{i=1}^{N^{\mathrm{cri}}})}{P(\theta | \{\mathcal{T}_i^{\mathrm{cri}}\}_{i=1}^{N^{\mathrm{cri}}})} (d\theta)(d\{\Lambda_i = \lambda_i\}_{i=1}^{N^{\mathrm{cri}}})(d\{\mathcal{T}_i^{\mathrm{cri}}\}_{i=1}^{N^{\mathrm{cri}}}),$$

$$\stackrel{(e)}{=} \int P(\theta | \{\bar{\mathcal{T}}_i^{\mathrm{cri}}(\Lambda_i = \lambda_i)\}_{i=1}^{N^{\mathrm{cri}}}) P(\{\Lambda_i = \lambda_i\}_{i=1}^{N^{\mathrm{cri}}}) P(\{\mathcal{T}_i^{\mathrm{cri}}\}_{i=1}^{N^{\mathrm{cri}}}) \cdot$$

$$\log \frac{P(\theta | \{\bar{\mathcal{T}}_i^{\mathrm{cri}}(\Lambda_i = \lambda_i)\}_{i=1}^{N^{\mathrm{cri}}})}{P(\theta | \{\mathcal{T}_i^{\mathrm{cri}}\}_{i=1}^{N^{\mathrm{cri}}})} (d\theta)(d\{\Lambda_i = \lambda_i\}_{i=1}^{N^{\mathrm{cri}}})(d\{\mathcal{T}_i^{\mathrm{cri}}\}_{i=1}^{N^{\mathrm{cri}}}),$$

$$= E_{\Lambda_i \in [0,1], \Lambda_i \sim P(\lambda), \theta \sim P(\cdot | \{\bar{\mathcal{T}}_i^{\mathrm{cri}}(\Lambda_i)\}_{i=1}^{N^{\mathrm{cri}}})} \left[ \log \frac{P(\theta | \{\bar{\mathcal{T}}_i^{\mathrm{cri}}(\Lambda_i)\}_{i=1}^{N^{\mathrm{cri}}})}{P(\theta | \{\mathcal{T}_i^{\mathrm{cri}}\}_{i=1}^{N^{\mathrm{cri}}})} \right],$$

where $(a)$ follows the definition of conditional mutual information (Wyner, 1978), $(b)$ follows the standard chain rule of probability $P(A, B, C) = P(A|B, C)P(B|C)P(C)$, $(c)$ follows the fact that $P(\{\bar{\mathcal{T}}_i^{\mathrm{cri}}(\Lambda_i = \lambda_i)\}_{i=1}^{N^{\mathrm{cri}}} | \{\mathcal{T}_i^{\mathrm{cri}}\}_{i=1}^{N^{\mathrm{cri}}}) = P(\{\Lambda_i = \lambda_i\}_{i=1}^{N^{\mathrm{cri}}})$ (explained in detail in Appendix E), $(d)$ follows the fact that $\int P(\{\bar{\mathcal{T}}_i(\Lambda_i = \lambda_i)\}_{i=1}^{N^{\mathrm{cri}}}) d(\{\bar{\mathcal{T}}_i(\Lambda_i = \lambda_i)\}_{i=1}^{N^{\mathrm{cri}}}) = \int P(\{\bar{\mathcal{T}}_i(\Lambda_i = \lambda_i)\}_{i=1}^{N^{\mathrm{cri}}}) d(\{\Lambda_i = \lambda_i\}_{i=1}^{N^{\mathrm{cri}}}) d(\{\mathcal{T}_i\}_{i=1}^{N^{\mathrm{cri}}})$ because $\bar{\mathcal{T}}_i^{\mathrm{cri}}(\Lambda_i = \lambda_i)$ is determinisitically determined by $\mathcal{T}_i^{\mathrm{cri}}$ and $\lambda_i$, and $(e)$ follows the fact that $P(\theta | \{\bar{\mathcal{T}}_i^{\mathrm{cri}}(\Lambda_i = \lambda_i)\}_{i=1}^{N^{\mathrm{cri}}}, \{\mathcal{T}_i^{\mathrm{cri}}\}_{i=1}^{N^{\mathrm{cri}}}) = P(\theta | \{\bar{\mathcal{T}}_i^{\mathrm{cri}}(\Lambda_i = \lambda_i)\}_{i=1}^{N^{\mathrm{cri}}})$ because the meta-parameter is trained on the augmented critical tasks $\{\bar{\mathcal{T}}_i^{\mathrm{cri}}(\Lambda_i = \lambda_i)\}_{i=1}^{N^{\mathrm{cri}}}$.

## E  DERIVATION OF THE POSTERIOR DISTRIBUTION (5)

Figure 4: How the distribution of $\theta$ is computed.

Figure 4 shows how the distribution of the meta-parameter $\theta$ is computed. At the beginning, we have the original critical tasks $\{\mathcal{T}_i^{\mathrm{cri}}\}_{i=1}^{N^{\mathrm{cri}}}$, however, the distribution of $\theta$ is not directly trained over the original critical tasks. Instead, we sample $\{\lambda_i\}_{i=1}^{N^{\mathrm{cri}}}$ from $P(\lambda)$ and augment the original critical tasks to the augmented critical tasks $\{\bar{\mathcal{T}}_i^{\mathrm{cri}}(\lambda_i)\}_{i=1}^{N^{\mathrm{cri}}}$ and train the distribution of $\theta$ over the augmented critical tasks. Since $\theta$ is directly trained on the augmented critical tasks $\{\bar{\mathcal{T}}_i^{\mathrm{cri}}(\lambda_i)\}_{i=1}^{N^{\mathrm{cri}}}$, the posterior

distribution of $\theta$ given the augmented critical tasks can be computed by solving the distributional optimization problem: $P^*(\cdot|\{\bar{\mathcal{T}}_i^{\mathrm{cri}}(\lambda_i)\}_{i=1}^{N^{\mathrm{cri}}}) = \arg\max_{\phi} E_{p_\phi(\theta)}\left[L(\theta, \{\bar{\mathcal{T}}_i^{\mathrm{cri}}(\lambda_i)\}_{i=1}^{N^{\mathrm{cri}}}, \{\mathcal{T}_i^{\mathrm{tr}}\}_{i=1}^{N^{\mathrm{tr}}-N^{\mathrm{cri}}})\right]$.

However, we cannot use the same way to formulate a distributional optimization problem to compute the posterior distribution $P^*(\cdot|\{\mathcal{T}_i^{\mathrm{cri}}\}_{i=1}^{N^{\mathrm{cri}}})$ because the meta-parameter $\theta$ is not directly trained over the original critical tasks $\{\mathcal{T}_i^{\mathrm{cri}}\}_{i=1}^{N^{\mathrm{cri}}}$. We can obtain this posterior distribution by marginalizing over $\{\bar{\mathcal{T}}_i^{\mathrm{cri}}(\Lambda_i)\}_{i=1}^{N^{\mathrm{cri}}}$:

$$P^*(\theta|\{\mathcal{T}_i^{\mathrm{cri}}\}_{i=1}^{N^{\mathrm{cri}}}) = \int_{\{\lambda_i\}_{i=1}^{N^{\mathrm{cri}}}} P^*(\theta, \{\bar{\mathcal{T}}_i^{\mathrm{cri}}(\Lambda_i = \lambda_i)\}_{i=1}^{N^{\mathrm{cri}}}|\{\mathcal{T}_i^{\mathrm{cri}}\}_{i=1}^{N^{\mathrm{cri}}}) d(\{\lambda_i\}_{i=1}^{N^{\mathrm{cri}}}),$$

$$= \int_{\{\lambda_i\}_{i=1}^{N^{\mathrm{cri}}}} P^*(\theta|\{\bar{\mathcal{T}}_i^{\mathrm{cri}}(\Lambda_i = \lambda_i)\}_{i=1}^{N^{\mathrm{cri}}}, \{\mathcal{T}_i^{\mathrm{cri}}\}_{i=1}^{N^{\mathrm{cri}}}) P(\{\bar{\mathcal{T}}_i^{\mathrm{cri}}(\Lambda_i = \lambda_i)\}_{i=1}^{N^{\mathrm{cri}}}|\{\mathcal{T}_i^{\mathrm{cri}}\}_{i=1}^{N^{\mathrm{cri}}}) d(\{\lambda_i\}_{i=1}^{N^{\mathrm{cri}}}),$$

$$\overset{(a)}{=} \int_{\{\lambda_i\}_{i=1}^{N^{\mathrm{cri}}}} P^*(\theta|\{\bar{\mathcal{T}}_i^{\mathrm{cri}}(\Lambda_i = \lambda_i)\}_{i=1}^{N^{\mathrm{cri}}}) P(\{\bar{\mathcal{T}}_i^{\mathrm{cri}}(\Lambda_i = \lambda_i)\}_{i=1}^{N^{\mathrm{cri}}}|\{\mathcal{T}_i^{\mathrm{cri}}\}_{i=1}^{N^{\mathrm{cri}}}) d(\{\lambda_i\}_{i=1}^{N^{\mathrm{cri}}}),$$

$$\overset{(b)}{=} \int_{\{\lambda_i\}_{i=1}^{N^{\mathrm{cri}}}} P^*(\theta|\{\bar{\mathcal{T}}_i^{\mathrm{cri}}(\Lambda_i = \lambda_i)\}_{i=1}^{N^{\mathrm{cri}}}) P(\{\Lambda_i = \lambda_i\}_{i=1}^{N^{\mathrm{cri}}}) d(\{\lambda_i\}_{i=1}^{N^{\mathrm{cri}}}),$$

$$= E_{\Lambda_i \in [0,1], \Lambda_i \sim P_{\phi_\lambda}(\lambda)}\left[P^*(\cdot|\{\bar{\mathcal{T}}_i^{\mathrm{cri}}(\Lambda_i)\}_{i=1}^{N^{\mathrm{cri}}})\right],$$

where $(a)$ follows the fact that $P^*(\theta|\{\bar{\mathcal{T}}_i^{\mathrm{cri}}(\Lambda_i = \lambda_i)\}_{i=1}^{N^{\mathrm{cri}}}, \{\mathcal{T}_i^{\mathrm{cri}}\}_{i=1}^{N^{\mathrm{cri}}}) = P^*(\theta|\{\bar{\mathcal{T}}_i^{\mathrm{cri}}(\Lambda_i = \lambda_i)\}_{i=1}^{N^{\mathrm{cri}}})$ because $\theta$ is directly trained on $\{\bar{\mathcal{T}}_i^{\mathrm{cri}}(\Lambda_i = \lambda_i)\}_{i=1}^{N^{\mathrm{cri}}}$, and $(b)$ follows the fact that $P(\{\bar{\mathcal{T}}_i^{\mathrm{cri}}(\Lambda_i = \lambda_i)\}_{i=1}^{N^{\mathrm{cri}}}|\{\mathcal{T}_i^{\mathrm{cri}}\}_{i=1}^{N^{\mathrm{cri}}}) = P(\{\Lambda_i = \lambda_i\}_{i=1}^{N^{\mathrm{cri}}})$ and the reason is provided as follows. For simplicity, we first consider one task, e.g., the first critical task $\mathcal{T}_1^{\mathrm{cri}}$. We use $\Lambda_1$ to denote a random variable that follows the distribution of $P(\lambda)$ and use $\lambda_1$ to denote a specific value sampled from $P(\lambda)$, e.g., $\lambda_1 = 0.5$. Therefore, $\bar{\mathcal{T}}_1^{\mathrm{cri}}(\Lambda_1 = \lambda_1 = 0.5)$ is a specific task because $\lambda_1$ is a specific value. Note that the critical task $\mathcal{T}_1^{\mathrm{cri}}$ is a training task that is already given, therefore, the probability of the (random) augmented task $\bar{\mathcal{T}}_1^{\mathrm{cri}}(\Lambda_1)$ being the specific task $\bar{\mathcal{T}}_1^{\mathrm{cri}}(\Lambda_1 = \lambda_1 = 0.5)$ is the probability of the specific value $\lambda_1 = 0.5$ being sampled from $P(\lambda)$. Therefore, $P(\bar{\mathcal{T}}_1^{\mathrm{cri}}(\Lambda_1 = \lambda_1 = 0.5)|\mathcal{T}_1^{\mathrm{cri}}) = P(\Lambda_1 = \lambda_1 = 0.5)$.

## F  PROOF OF LEMMA 1

Recall from (4) that

$$I(\theta; \{\bar{\mathcal{T}}_i^{\mathrm{cri}}(\Lambda_i \sim P_{\phi_\lambda}(\lambda))\}_{i=1}^{N^{\mathrm{cri}}}|\{\mathcal{T}_i^{\mathrm{cri}}\}_{i=1}^{N^{\mathrm{cri}}}),$$

$$= E_{\Lambda_i \in [0,1], \Lambda_i \sim P_{\phi_\lambda}(\lambda), \theta \sim P^*(\cdot|\{\bar{\mathcal{T}}_i^{\mathrm{cri}}(\Lambda_i)\}_{i=1}^{N^{\mathrm{cri}}})}\left[\log \frac{P^*(\theta|\{\bar{\mathcal{T}}_i^{\mathrm{cri}}(\Lambda_i)\}_{i=1}^{N^{\mathrm{cri}}})}{P^*(\theta|\{\mathcal{T}_i^{\mathrm{cri}}\}_{i=1}^{N^{\mathrm{cri}}})}\right].$$

Since $P_{\phi^*(\{\lambda_i^{\bar{\zeta}_j}\}_{i=1}^{N^{\mathrm{cri}}})}(\theta)$ is Gaussian distribution, we have that

$$I(\theta; \{\bar{\mathcal{T}}_i^{\mathrm{cri}}(\Lambda_i \sim P_{\phi_\lambda}(\lambda))\}_{i=1}^{N^{\mathrm{cri}}}|\{\mathcal{T}_i^{\mathrm{cri}}\}_{i=1}^{N^{\mathrm{cri}}}),$$

$$= E_{\Lambda_i, \theta}\left[\log \frac{\frac{\exp(-\frac{1}{2}(\theta - \mu^*(\{\Lambda_i^{\bar{\zeta}}\}_{i=1}^{N^{\mathrm{cri}}}))^{\top}(\Sigma^*(\{\Lambda_i^{\bar{\zeta}}\}_{i=1}^{N^{\mathrm{cri}}}))^{-1}(\theta - \mu^*(\{\Lambda_i^{\bar{\zeta}}\}_{i=1}^{N^{\mathrm{cri}}})))}{\sqrt{|(\sigma^*(\{\Lambda_i^{\bar{\zeta}}\}_{i=1}^{N^{\mathrm{cri}}}))^{\top}\sigma^*(\{\Lambda_i^{\bar{\zeta}}\}_{i=1}^{N^{\mathrm{cri}}})|}}}{E_{\Lambda_i}\left[\frac{\exp(-\frac{1}{2}(\theta - \mu^*(\{\Lambda_i^{\bar{\zeta}}\}_{i=1}^{N^{\mathrm{cri}}}))^{\top}(\Sigma^*(\{\Lambda_i^{\bar{\zeta}}\}_{i=1}^{N^{\mathrm{cri}}}))^{-1}(\theta - \mu^*(\{\Lambda_i^{\bar{\zeta}}\}_{i=1}^{N^{\mathrm{cri}}})))}{\sqrt{|(\sigma^*(\{\Lambda_i^{\bar{\zeta}}\}_{i=1}^{N^{\mathrm{cri}}}))^{\top}\sigma^*(\{\Lambda_i^{\bar{\zeta}}\}_{i=1}^{N^{\mathrm{cri}}})|}}\right]}\right],$$

$$= E_{\Lambda_i, \theta}\left[\log \frac{\exp(-\frac{1}{2}(\theta - \mu^*(\{\Lambda_i^{\bar{\zeta}}\}_{i=1}^{N^{\mathrm{cri}}}))^{\top}(\Sigma^*(\{\Lambda_i^{\bar{\zeta}}\}_{i=1}^{N^{\mathrm{cri}}}))^{-1}(\theta - \mu^*(\{\Lambda_i^{\bar{\zeta}}\}_{i=1}^{N^{\mathrm{cri}}})))}{\sqrt{|(\sigma^*(\{\Lambda_i^{\bar{\zeta}}\}_{i=1}^{N^{\mathrm{cri}}}))^{\top}\sigma^*(\{\Lambda_i^{\bar{\zeta}}\}_{i=1}^{N^{\mathrm{cri}}})|}}\right]$$

$$- E_{\theta}\left[\log E_{\Lambda_i}\left[\frac{\exp(-\frac{1}{2}(\theta - \mu^*(\{\Lambda_i^{\bar{\zeta}}\}_{i=1}^{N^{\mathrm{cri}}}))^{\top}(\Sigma^*(\{\Lambda_i^{\bar{\zeta}}\}_{i=1}^{N^{\mathrm{cri}}}))^{-1}(\theta - \mu^*(\{\Lambda_i^{\bar{\zeta}}\}_{i=1}^{N^{\mathrm{cri}}})))}{\sqrt{|(\sigma^*(\{\Lambda_i^{\bar{\zeta}}\}_{i=1}^{N^{\mathrm{cri}}}))^{\top}\sigma^*(\{\Lambda_i^{\bar{\zeta}}\}_{i=1}^{N^{\mathrm{cri}}})|}}\right]\right],$$

$$\overset{(a)}{=} E_{\zeta \sim \mathcal{N}(0, I)}\left\{E_{\Lambda_i}\left[\log \frac{\exp(-\frac{1}{2}\zeta^{\top}\zeta)}{\sqrt{|(\sigma^*(\{\Lambda_i^{\bar{\zeta}}\}_{i=1}^{N^{\mathrm{cri}}}))^{\top}\sigma^*(\{\Lambda_i^{\bar{\zeta}}\}_{i=1}^{N^{\mathrm{cri}}})|}}\right]\right.$$

$$- \log E_{\Lambda_i} \Big[ \frac{\exp(-\frac{1}{2}\zeta^\top \zeta)}{\sqrt{|(\sigma^*(\{\Lambda_i^{\bar\zeta}\}_{i=1}^{N^{\mathrm{cri}}}))^\top \sigma^*(\{\Lambda_i^{\bar\zeta_j}\}_{i=1}^{N^{\mathrm{cri}}})|}} \Big] \Big\},$$

$$= E_{\Lambda_i} \Big[ \log \frac{1}{\sqrt{|(\sigma^*(\{\Lambda_i^{\bar\zeta}\}_{i=1}^{N^{\mathrm{cri}}}))^\top \sigma^*(\{\Lambda_i^{\bar\zeta}\}_{i=1}^{N^{\mathrm{cri}}})|}} \Big] - \log E_{\lambda_i} \Big[ \frac{1}{\sqrt{|(\sigma^*(\{\Lambda_i^{\bar\zeta_j}\}_{i=1}^{N^{\mathrm{cri}}}))^\top \sigma^*(\{\Lambda_i^{\bar\zeta}\}_{i=1}^{N^{\mathrm{cri}}})|}} \Big] \tag{8}$$

where $(a)$ follows the fact that $\theta = \mu^*(\{\Lambda_i^{\bar\zeta}\}_{i=1}^{N^{\mathrm{cri}}}) + \sigma^*(\{\Lambda_i^{\bar\zeta}\}_{i=1}^{N^{\mathrm{cri}}}) \circ \zeta$. Since we sample $N^{\bar\zeta}$ sets of mixing coefficients $\{\{\lambda_i^{\bar\zeta}\}_{i=1}^{N^{\mathrm{cri}}}\}_{j=1}^{N^{\bar\zeta}}$ from $P_{\phi_\lambda}(\lambda)$, the conditional mutual information can be estimated by

$$I(\theta; \{\bar{\mathcal{T}}_i^{\mathrm{cri}}(\Lambda_i \sim P_{\phi_\lambda}(\lambda))\}_{i=1}^{N^{\mathrm{cri}}} | \{\mathcal{T}_i^{\mathrm{cri}}\}_{i=1}^{N^{\mathrm{cri}}}),$$

$$= \frac{1}{N^{\bar\zeta}} \sum_{j=1}^{N^{\bar\zeta}} \log \frac{1}{\sqrt{|(\sigma^*(\{\lambda_i^{\bar\zeta_j}\}_{i=1}^{N^{\mathrm{cri}}}))^\top \sigma^*(\{\lambda_i^{\bar\zeta_j}\}_{i=1}^{N^{\mathrm{cri}}})|}} - \log \frac{1}{N^{\bar\zeta}} \sum_{j=1}^{N^{\bar\zeta}} \frac{1}{\sqrt{|(\sigma^*(\{\lambda_i^{\bar\zeta_j}\}_{i=1}^{N^{\mathrm{cri}}}))^\top \sigma^*(\{\lambda_i^{\bar\zeta_j}\}_{i=1}^{N^{\mathrm{cri}}})|}}.$$

Therefore, we can get the gradient:

$$\nabla_{\phi_\lambda} I(\theta; \{\bar{\mathcal{T}}_i^{\mathrm{cri}}(\Lambda_i \sim P_{\phi_\lambda}(\lambda))\}_{i=1}^{N^{\mathrm{cri}}} | \{\mathcal{T}_i^{\mathrm{cri}}\}_{i=1}^{N^{\mathrm{cri}}}),$$

$$= \frac{\sum_{j=1}^{N^{\bar\zeta}} \nabla_{\phi_\lambda} \sigma^*(\{\lambda_i^{\bar\zeta_j}\}_{i=1}^{N^{\mathrm{cri}}})}{\|\sum_{j=1}^{N^{\bar\zeta}} \sigma^*(\{\lambda_i^{\bar\zeta_j}\}_{i=1}^{N^{\mathrm{cri}}})\|} - \frac{1}{N^{\bar\zeta}} \sum_{j=1}^{N^{\bar\zeta}} \frac{\nabla_{\phi_\lambda} \sigma^*(\{\lambda_i^{\bar\zeta_j}\}_{i=1}^{N^{\mathrm{cri}}})}{\|\sigma^*(\{\lambda_i^{\bar\zeta_j}\}_{i=1}^{N^{\mathrm{cri}}})\|}.$$

To get $\nabla_{\phi_\lambda} \sigma^*$, we know that $\phi^* = \arg\max E_{P_\phi(\theta)}[L(\theta, \{\bar{\mathcal{T}}_i^{\mathrm{cri}}(\lambda_i^{\bar\zeta_j})\}_{i=1}^{N^{\mathrm{cri}}}, \{\mathcal{T}_i^{\mathrm{tr}}\}_{i=1}^{N^{\mathrm{tr}}-N^{\mathrm{cri}}})]$, therefore, we have that $\nabla_\sigma E_{P_{\phi^*}(\theta)}[L(\theta, \{\bar{\mathcal{T}}_i^{\mathrm{cri}}(\lambda_i^{\bar\zeta_j})\}_{i=1}^{N^{\mathrm{cri}}}, \{\mathcal{T}_i^{\mathrm{tr}}\}_{i=1}^{N^{\mathrm{tr}}-N^{\mathrm{cri}}})] = 0$. Then we have that

$$\frac{d}{d\phi_\lambda} \nabla_\sigma E_{P_{\phi^*}(\theta)}[L(\theta, \{\bar{\mathcal{T}}_i^{\mathrm{cri}}(\lambda_i^{\bar\zeta_j})\}_{i=1}^{N^{\mathrm{cri}}}, \{\mathcal{T}_i^{\mathrm{tr}}\}_{i=1}^{N^{\mathrm{tr}}-N^{\mathrm{cri}}})],$$

$$= \nabla_{\sigma\phi_\lambda} E_{P_{\phi^*}(\theta)}[L(\theta, \{\bar{\mathcal{T}}_i^{\mathrm{cri}}(\lambda_i^{\bar\zeta_j})\}_{i=1}^{N^{\mathrm{cri}}}, \{\mathcal{T}_i^{\mathrm{tr}}\}_{i=1}^{N^{\mathrm{tr}}-N^{\mathrm{cri}}})]$$

$$+ \nabla_{\sigma\sigma} E_{P_{\phi^*}(\theta)}[L(\theta, \{\bar{\mathcal{T}}_i^{\mathrm{cri}}(\lambda_i^{\bar\zeta_j})\}_{i=1}^{N^{\mathrm{cri}}}, \{\mathcal{T}_i^{\mathrm{tr}}\}_{i=1}^{N^{\mathrm{tr}}-N^{\mathrm{cri}}})] \nabla_{\phi_\lambda} \sigma^* = 0,$$

$$\Rightarrow \nabla_{\phi_\lambda} \sigma^* = -\Big[ \nabla_{\sigma\sigma}^2 E_{P_{\phi^*}(\theta)}[L(\theta, \{\bar{\mathcal{T}}_i^{\mathrm{cri}}(\lambda_i^{\bar\zeta_j})\}_{i=1}^{N^{\mathrm{cri}}}, \{\mathcal{T}_i^{\mathrm{tr}}\}_{i=1}^{N^{\mathrm{tr}}-N^{\mathrm{cri}}})] \Big]^{-1}.$$

$$\nabla_{\sigma\phi_\lambda}^2 E_{P_{\phi^*}(\theta)}[L(\theta, \{\bar{\mathcal{T}}_i^{\mathrm{cri}}(\lambda_i^{\bar\zeta_j})\}_{i=1}^{N^{\mathrm{cri}}}, \{\mathcal{T}_i^{\mathrm{tr}}\}_{i=1}^{N^{\mathrm{tr}}-N^{\mathrm{cri}}})].$$

## G  GRADIENTS

This section provides all the gradients needed in this paper.

### G.1  META-GRADIENTS FOR MAJOR MRL METHODS

Recall the problem formulation (1) of MRL as follows where we omit the superscript for simplicity:

$$\max_\theta L(\theta, \{\mathcal{T}_i\}_{i=1}^N) = \frac{1}{N} \sum_{i=1}^N J_i(\pi_i(\theta)), \quad \text{s.t. } \pi_i(\theta) = Alg(\pi_\theta, \mathcal{T}_i).$$

The meta-gradient is the gradient of the upper-level objective w.r.t. $\theta$, i.e., $\nabla_\theta L(\theta, \{\mathcal{T}_i\}_{i=1}^N)$. The meta-gradient is different for different algorithms because different algorithms use different ways to compute the task specific adaptations $\pi_i(\theta)$. Here, we provide the meta-gradients for several major MRL algorithms, including MAML (Finn et al., 2017; Fallah et al., 2021), iMAML (Rajeswaran et al., 2019), and context-based MRL (e.g., CAVIA (Zintgraf et al., 2019a)).

**Lemma 4.** *The meta-gradients for MAML, iMAML, and CAVIA are respectively:*

$$\nabla_\theta L(\theta, \{\mathcal{T}_i\}_{i=1}^N) = \frac{1}{N} \sum_{i=1}^N [I + \alpha \nabla_{\theta\theta}^2 J_i(\pi_\theta)] \nabla_{\theta_i} J_i(\pi_{\theta_i}), \qquad \text{(MAML)}$$

$$\nabla_\theta L(\theta, \{\mathcal{T}_i\}_{i=1}^N) = \frac{1}{N}\sum_{i=1}^N [1 + \frac{1}{\bar{\lambda}}\nabla_{\psi\psi}^2 J_i(\pi_{\theta_i'})]^{-1}\nabla_{\theta_i} J_i(\pi_{\theta_i}), \qquad (iMAML)$$

$$\nabla_\theta L(\theta, \{\mathcal{T}_i\}_{i=1}^N) = \frac{1}{N}\sum_{i=1}^N \nabla_\theta J_i(\pi_\theta(\cdot|\cdot, \psi_i'')), \qquad (CAVIA)$$

*where $\alpha$ is a step size, $\theta_i = \theta + \alpha\nabla_\theta J_i(\pi_\theta)$, $\nabla_{\theta_i} J_i(\pi_{\theta_i}) = E_{(s,a)\sim\rho^{\pi_{\theta_i}}}[\nabla_{\theta_i}\log\pi_{\theta_i}(a|s)A_i^{\pi_{\theta_i}}(s,a)]$, $\nabla_{\theta\theta}^2 J_i(\pi_\theta) = E_{(s,a)\sim\rho^{\pi_\theta}}\Big[\sum_{t=0}^\infty \gamma^t \nabla_\theta E_{(s,a)\sim\rho^{\pi_\theta}}[\log\pi_\theta(a|s)Q_i^{\pi_\theta}(s,a)](\nabla_\theta\log\pi_\theta(a|s))^\top + \nabla_{\theta\theta}^2 E_{(s,a)\sim\rho^{\pi_\theta}}[\log\pi_\theta(a|s)Q_i^{\pi_\theta}(s,a)]\Big]$, $\bar{\lambda}$ is a hyper-parameter, $\theta_i' = \arg\max_\psi J_i(\pi_\psi) + \frac{\bar{\lambda}}{2}||\psi - \theta||^2$, $\pi_\theta(\cdot|\cdot, \psi_i'')$ is a context-based policy where $\psi_i'' = \psi_0 + \alpha\nabla_\psi J_i(\pi_\theta(\cdot|\cdot, \psi_0))$ is the context.*

*Proof.* MAML computes the task-specific adaptation via one-step gradient ascent. Specifically, suppose the task-specific adaptation is $\pi_{\theta_i} = \pi_i(\theta)$, and thus $\theta_i = \theta + \alpha\nabla_\theta J_i(\pi_\theta)$. Therefore, the meta-gradient is $\nabla_\theta L(\theta, \{\mathcal{T}_i\}_{i=1}^N) = \frac{1}{N}\sum_{i=1}^N \nabla_\theta J_i(\pi_{\theta_i}) = \frac{1}{N}\sum_{i=1}^N (\nabla_\theta\theta_i)^\top \nabla_{\theta_i} J_i(\pi_{\theta_i}) = \frac{1}{N}\sum_{i=1}^N [I + \alpha\nabla_{\theta\theta}^2 J_i(\pi_\theta)]\nabla_{\theta_i} J_i(\pi_{\theta_i})$. From (Fallah et al., 2021), we can get that the policy gradient is $\nabla_{\theta_i} J_i(\pi_{\theta_i}) = E_{(s,a)\sim\rho^{\pi_{\theta_i}}}[\nabla_{\theta_i}\log\pi_{\theta_i}(a|s)A_i^{\pi_{\theta_i}}(s,a)]$ and the Hessian is $\nabla_{\theta\theta}^2 J_i(\pi_\theta) = E_{(s,a)\sim\rho^{\pi_\theta}}\Big[\sum_{t=0}^\infty \gamma^t \nabla_\theta E_{(s,a)\sim\rho^{\pi_\theta}}[\log\pi_\theta(a|s)Q_i^{\pi_\theta}(s,a)](\nabla_\theta\log\pi_\theta(a|s))^\top + \nabla_{\theta\theta}^2 E_{(s,a)\sim\rho^{\pi_\theta}}[\log\pi_\theta(a|s)Q_i^{\pi_\theta}(s,a)]\Big]$.

iMAML solves the optimization problem to get the task-specific adaptation $\pi_{\theta_i'}$ such that $\theta_i' = \arg\max_\psi J_i(\pi_\psi) + \frac{\bar{\lambda}}{2}||\psi - \theta||^2$ where $\bar{\lambda}$ is a hyper-parameter. Since $\theta_i'$ is the optimal parameter of the problem $\max_\psi J_i(\pi_\psi) + \frac{\bar{\lambda}}{2}||\psi - \theta||^2$, we know that $\nabla_\psi J_i(\pi_{\theta_i'}) + \bar{\lambda}(\theta_i' - \theta) = 0$. Take gradient w.r.t. $\theta$ on both sides, we can get that $(\nabla_\theta\theta_i')^\top \nabla_{\psi\psi}^2 J_i(\pi_{\theta_i'}) + \bar{\lambda}(\nabla_\theta\theta_i' - I) = 0 \Rightarrow \nabla_\theta\theta_i' = [1 + \frac{1}{\bar{\lambda}}\nabla_{\psi\psi}^2 J_i(\pi_{\theta_i'})]^{-1}$. Therefore, the meta-gradient is $\nabla_\theta L(\theta, \{\mathcal{T}_i\}_{i=1}^N) = \frac{1}{N}\sum_{i=1}^N \nabla_\theta J_i(\pi_{\theta_i}) = \frac{1}{N}\sum_{i=1}^N (\nabla_\theta\theta_i)^\top \nabla_{\theta_i} J_i(\pi_{\theta_i}) = \frac{1}{N}\sum_{i=1}^N [1 + \frac{1}{\bar{\lambda}}\nabla_{\psi\psi}^2 J_i(\pi_{\theta_i'})]^{-1}\nabla_{\theta_i} J_i(\pi_{\theta_i})$.

CAVIA learns a context-based policy $\pi_\theta(a|s, \psi_i'')$ and uses MAML-like method to update $\psi_i'' = \psi_0 + \alpha\nabla_\psi J_i(\pi_\theta(\cdot|\cdot, \psi_0))$. Therefore, the meta-gradient is $\nabla_\theta L(\theta, \{\mathcal{T}_i\}_{i=1}^N) = \frac{1}{N}\sum_{i=1}^N \nabla_\theta J_i(\pi_\theta(\cdot|\cdot, \psi_i''))$. $\qquad\square$

### G.2 OTHER GRADIENTS

This part provides the expressions of $\nabla_{\sigma\sigma}^2 E_{P_{\phi^*}(\theta)}[L(\theta, \{\bar{\mathcal{T}}_i^{cri}(\lambda_i^{\bar{\zeta}_j})\}_{i=1}^{N^{cri}}, \{\mathcal{T}_i^{tr}\}_{i=1}^{N^{tr}-N^{cri}})]$ and $\nabla_{\sigma\phi_\lambda}^2 E_{P_{\phi^*}(\theta)}[L(\theta, \{\bar{\mathcal{T}}_i^{cri}(\lambda_i^{\bar{\zeta}_j})\}_{i=1}^{N^{cri}}, \{\mathcal{T}_i^{tr}\}_{i=1}^{N^{tr}-N^{cri}})]$ needed in Lemma 1.

**Lemma 5.** *We have the following expressions:*

$$\nabla_{\sigma\sigma}^2 E_{P_{\phi^*}(\theta)}[L(\theta, \{\bar{\mathcal{T}}_i^{cri}(\lambda_i^{\bar{\zeta}_j})\}_{i=1}^{N^{cri}}, \{\mathcal{T}_i^{tr}\}_{i=1}^{N^{tr}-N^{cri}})],$$

$$= E_{\zeta\sim\mathcal{N}(0,I)}\Big[\frac{1}{N^{tr}}[\sum_{i=1}^{N^{cri}} \nabla_{\sigma\sigma}^2 \bar{J}_i^{cri}(\pi_i^{cri}(\mu^* + \sigma^* \circ \zeta), \lambda_i^{\bar{\zeta}_j}) + \sum_{i=1}^{N^{tr}-N^{cri}} \nabla_{\sigma\sigma}^2 J_i^{tr}(\pi_i^{tr}(\mu^* + \sigma^* \circ \zeta))]\Big],$$

$$\nabla_{\sigma\phi_\lambda}^2 E_{P_{\phi^*}(\theta)}[L(\theta, \{\bar{\mathcal{T}}_i^{cri}(\lambda_i^{\bar{\zeta}_j})\}_{i=1}^{N^{cri}}, \{\mathcal{T}_i^{tr}\}_{i=1}^{N^{tr}-N^{cri}})],$$

$$= E_{\zeta\sim\mathcal{N}(0,I)}\Big[\frac{1}{N^{tr}}[\sum_{i=1}^{N^{cri}} \nabla_{\phi_\lambda}\lambda_j \int_{(s_{jj'},a_{jj'})\in\mathcal{S}\times\mathcal{A}} \Big[\bar{\rho}^{\pi_{\theta_i},\lambda_j}(s_{jj'},a_{jj'})\Big(\nabla_{\theta_i s}\log\pi_{\theta_i}(\bar{a}_{jj'}|\bar{s}_{jj'})(s_j - s_{j'})\Big)\cdot$$

$$\bar{A}_{jj'} + \bar{\rho}^{\pi_{\theta_i},\lambda_j}(s_{jj'},a_{jj'})\nabla_{\theta_i}\log\pi_{\theta_i}(\bar{a}_{jj'}|\bar{s}_{jj'})(A_i^{\pi_{\theta_i}}(s_j,a_j) - A_i^{\pi_{\theta_i}}(s_{j'},a_{j'}))\Big]da_{jj'}ds_{jj'}\Big],$$

*where the expression of the second-order term $\nabla_{\sigma\sigma}^2 \bar{J}_i^{cri}(\pi_i^{cri}(\mu^* + \sigma^* \circ \zeta), \lambda_i)$ can be found in Lemma 4.*

*Proof.* Recall that $\phi^* = (\mu^*, \sigma^*)$, $\theta = \mu + \sigma \circ \zeta$, and $\zeta \sim \mathcal{N}(0, I)$. Therefore, we have that

$$\nabla_\sigma E_{P_{\phi^*}(\theta)}[L(\theta, \{\bar{\mathcal{T}}_i^{\text{cri}}(\lambda_i^{\bar{\zeta}_j})\}_{i=1}^{N^{\text{cri}}}, \{\mathcal{T}_i^{\text{tr}}\}_{i=1}^{N^{\text{tr}} - N^{\text{cri}}})],$$

$$= E_{\zeta \sim \mathcal{N}(0,I)}[\nabla_\sigma L(\mu^* + \sigma^* \circ \zeta, \{\bar{\mathcal{T}}_i^{\text{cri}}(\lambda_i^{\bar{\zeta}_j})\}_{i=1}^{N^{\text{cri}}}, \{\mathcal{T}_i^{\text{tr}}\}_{i=1}^{N^{\text{tr}} - N^{\text{cri}}})],$$

$$= E_{\zeta \sim \mathcal{N}(0,I)}\left[\frac{1}{N^{\text{tr}}}[\sum_{i=1}^{N^{\text{cri}}} \nabla_\sigma \bar{J}_i^{\text{cri}}(\pi_i^{\text{cri}}(\mu^* + \sigma^* \circ \zeta), \lambda_i) + \sum_{i=1}^{N^{\text{tr}} - N^{\text{cri}}} \nabla_\sigma J_i^{\text{tr}}(\pi_i^{\text{tr}}(\mu^* + \sigma^* \circ \zeta))]\right]. \quad (9)$$

Therefore, we can get the Hessian:

$$\nabla_{\sigma\sigma}^2 E_{P_{\phi^*}(\theta)}[L(\theta, \{\bar{\mathcal{T}}_i^{\text{cri}}(\lambda_i^{\bar{\zeta}_j})\}_{i=1}^{N^{\text{cri}}}, \{\mathcal{T}_i^{\text{tr}}\}_{i=1}^{N^{\text{tr}} - N^{\text{cri}}})],$$

$$= E_{\zeta \sim \mathcal{N}(0,I)}\left[\frac{1}{N^{\text{tr}}}[\sum_{i=1}^{N^{\text{cri}}} \nabla_{\sigma\sigma}^2 \bar{J}_i^{\text{cri}}(\pi_i^{\text{cri}}(\mu^* + \sigma^* \circ \zeta), \lambda_i^{\bar{\zeta}_j}) + \sum_{i=1}^{N^{\text{tr}} - N^{\text{cri}}} \nabla_{\sigma\sigma}^2 J_i^{\text{tr}}(\pi_i^{\text{tr}}(\mu^* + \sigma^* \circ \zeta))]\right],$$

where the expression of the second-order term $\nabla_{\sigma\sigma}^2 \bar{J}_i^{\text{cri}}(\pi_i^{\text{cri}}(\mu^* + \sigma^* \circ \zeta), \lambda_i)$ can be found in Lemma 4. Similarly, we can get that

$$\nabla_{\sigma\phi_\lambda}^2 E_{P_{\phi^*}(\theta)}[L(\theta, \{\bar{\mathcal{T}}_i^{\text{cri}}(\lambda_i^{\bar{\zeta}_j})\}_{i=1}^{N^{\text{cri}}}, \{\mathcal{T}_i^{\text{tr}}\}_{i=1}^{N^{\text{tr}} - N^{\text{cri}}})],$$

$$= E_{\zeta \sim \mathcal{N}(0,I)}\left[\frac{1}{N^{\text{tr}}}[\sum_{i=1}^{N^{\text{cri}}} \nabla_{\sigma\phi_\lambda}^2 \bar{J}_i^{\text{cri}}(\pi_i^{\text{cri}}(\mu^* + \sigma^* \circ \zeta), \lambda_i^{\bar{\zeta}_j}) + \sum_{i=1}^{N^{\text{tr}} - N^{\text{cri}}} \nabla_{\sigma\phi_\lambda}^2 J_i^{\text{tr}}(\pi_i^{\text{tr}}(\mu^* + \sigma^* \circ \zeta))]\right],$$

$$= E_{\zeta \sim \mathcal{N}(0,I)}\left[\frac{1}{N^{\text{tr}}}[\sum_{i=1}^{N^{\text{cri}}} \nabla_{\sigma\phi_\lambda}^2 \bar{J}_i^{\text{cri}}(\pi_i^{\text{cri}}(\mu^* + \sigma^* \circ \zeta), \mu_\lambda + \sigma_\lambda \bar{\zeta}_j)]\right].$$

Now we need to derive the expression of $\nabla_{\sigma\phi_\lambda}^2 \bar{J}_i^{\text{cri}}(\pi_i^{\text{cri}}(\mu^* + \sigma^* \circ \zeta), \mu_\lambda + \sigma_\lambda \bar{\zeta}_j)$. Suppose we use MAML, and thus the first-order gradient $\nabla_\sigma \bar{J}_i^{\text{cri}}(\pi_i^{\text{cri}}(\mu^* + \sigma^* \circ \zeta), \mu_\lambda + \sigma_\lambda \bar{\zeta}_j) = [I + \alpha\nabla_{\sigma\sigma}^2 \bar{J}_i(\pi_{\mu^*+\sigma^*\circ\zeta}, \mu_\lambda + \sigma_\lambda \bar{\zeta}_j)][\nabla_{\theta_i} \bar{J}_i(\pi_{\theta_i}, \mu_\lambda + \sigma_\lambda \bar{\zeta}_j)]$ where $\theta_i = \mu^* + \sigma^* \circ \zeta + \alpha\nabla_\theta \bar{J}_i(\pi_{\mu^*+\sigma^*\circ\zeta}, \mu_\lambda + \sigma_\lambda \bar{\zeta}_j)$ and $\theta = \mu^* + \sigma^* \circ \zeta$. Following the first-order MAML method in (Fallah et al., 2020), we use the gradient $\nabla_\sigma \bar{J}_i^{\text{cri}}(\pi_i^{\text{cri}}(\mu^* + \sigma^* \circ \zeta), \mu_\lambda + \sigma_\lambda \bar{\zeta}_j) = \nabla_\sigma \bar{J}_i(\pi_{\theta_i}, \mu_\lambda + \sigma_\lambda \bar{\zeta}_j)]$. To get the term $\nabla_{\sigma\phi_\lambda}^2 \bar{J}_i^{\text{cri}}(\pi_i^{\text{cri}}(\mu^* + \sigma^* \circ \zeta), \mu_\lambda + \sigma_\lambda \bar{\zeta}_j)$, we derive $\nabla_{\theta_i\phi_\lambda}^2 \bar{J}_i(\pi_{\theta_i}, \mu_\lambda + \sigma_\lambda \bar{\zeta}_j)$.

$$\nabla_{\phi_\lambda, \theta_i}^2 \bar{J}_i(\pi_{\theta_i}, \mu_\lambda + \sigma_\lambda \bar{\zeta}_j) = \nabla_{\phi_\lambda} E_{(s_{jj'}, a_{jj'}) \sim \bar{\rho}^{\pi_{\theta_i}, \lambda_j}}[\nabla_{\theta_i} \log \pi_{\theta_i}(\bar{a}_{jj'}|\bar{s}_{jj'})\bar{A}_{jj'}],$$

$$= \nabla_{\phi_\lambda} \int_{(s_{jj'}, a_{jj'}) \in \mathcal{S} \times \mathcal{A}} \bar{\rho}^{\pi_{\theta_i}, \lambda_j}(s_{jj'}, a_{jj'})\nabla_{\theta_i} \log \pi_{\theta_i}(\bar{a}_{jj'}|\bar{s}_{jj'})\bar{A}_{jj'} da_{jj'} ds_{jj'},$$

$$= \nabla_{\phi_\lambda} \int_{(s_{jj'}, a_{jj'}) \in \mathcal{S} \times \mathcal{A}} \left[\bar{\rho}^{\pi_{\theta_i}, \lambda_j}(s_{jj'}, a_{jj'})\nabla_{\theta_i} \log \pi_{\theta_i}(\bar{a}_{jj'}|\bar{s}_{jj'})\bar{A}_{jj'}\right] da_{jj'} ds_{jj'},$$

$$= \nabla_{\phi_\lambda}\lambda_j \cdot \int_{(s_{jj'}, a_{jj'}) \in \mathcal{S} \times \mathcal{A}} \nabla_{\lambda_j}\left[\bar{\rho}^{\pi_{\theta_i}, \lambda_j}(s_{jj'}, a_{jj'})\nabla_{\theta_i} \log \pi_{\theta_i}(\bar{a}_{jj'}|\bar{s}_{jj'})\bar{A}_{jj'}\right] da_{jj'} ds_{jj'},$$

$$\overset{(a)}{=} \nabla_{\phi_\lambda}\lambda_j \cdot \int_{(s_{jj'}, a_{jj'}) \in \mathcal{S} \times \mathcal{A}} \left[\bar{\rho}^{\pi_{\theta_i}, \lambda_j}(s_{jj'}, a_{jj'})\left(\nabla_{\theta_i s} \log \pi_{\theta_i}(\bar{a}_{jj'}|\bar{s}_{jj'})(s_j - s_{j'})\right)\bar{A}_{jj'}\right.$$

$$\left. + \bar{\rho}^{\pi_{\theta_i}, \lambda_j}(s_{jj'}, a_{jj'})\nabla_{\theta_i} \log \pi_{\theta_i}(\bar{a}_{jj'}|\bar{s}_{jj'})(A_i^{\pi_{\theta_i}}(s_j, a_j) - A_i^{\pi_{\theta_i}}(s_{j'}, a_{j'}))\right] da_{jj'} ds_{jj'},$$

where $(a)$ follows the fact that $\nabla_{\lambda_i} \bar{\rho}^{\pi_{\theta_i}, \lambda_j}(s_{jj'}, a_{jj'}) = 0$ and $\nabla_{\theta a} \log \pi_{\theta_i}(\bar{a}_{jj'}|\bar{s}_{jj'})$ because they include indicator functions. Therefore, we have that

$$\nabla_{\sigma\phi_\lambda}^2 E_{P_{\phi^*}(\theta)}[L(\theta, \{\bar{\mathcal{T}}_i^{\text{cri}}(\lambda_i^{\bar{\zeta}_j})\}_{i=1}^{N^{\text{cri}}}, \{\mathcal{T}_i^{\text{tr}}\}_{i=1}^{N^{\text{tr}} - N^{\text{cri}}})],$$

$$= E_{\zeta \sim \mathcal{N}(0,I)}\left[\frac{1}{N^{\text{tr}}}[\sum_{i=1}^{N^{\text{cri}}} \nabla_{\phi_\lambda}\lambda_j \int_{(s_{jj'}, a_{jj'}) \in \mathcal{S} \times \mathcal{A}} \left[\bar{\rho}^{\pi_{\theta_i}, \lambda_j}(s_{jj'}, a_{jj'})\left(\nabla_{\theta_i s} \log \pi_{\theta_i}(\bar{a}_{jj'}|\bar{s}_{jj'})(s_j - s_{j'})\right) \cdot\right.\right.$$

$$\left.\left. \bar{A}_{jj'} + \bar{\rho}^{\pi_{\theta_i}, \lambda_j}(s_{jj'}, a_{jj'})\nabla_{\theta_i} \log \pi_{\theta_i}(\bar{a}_{jj'}|\bar{s}_{jj'})(A_i^{\pi_{\theta_i}}(s_j, a_j) - A_i^{\pi_{\theta_i}}(s_{j'}, a_{j'}))\right] da_{jj'} ds_{jj'}\right].$$

$\square$

## H PROOF OF THEOREM 1

This section first prove that the conditional mutual information $I(\theta; \{\bar{\mathcal{T}}_i^{\text{cri}}(\lambda_i \sim P_{\phi_\lambda}(\lambda))\}_{i=1}^{N^{\text{cri}}} | \{\mathcal{T}_i^{\text{cri}}\}_{i=1}^{N^{\text{cri}}})$ is $C_I$-Lipschitz continuous and $\bar{C}_I$-smooth where $C_I$ and $\bar{C}_I$ are positive constants in Claim 1, and then prove that Algorithm 1 converges at the rate of $O(1/\sqrt{K})$.

**Claim 1.** *The conditional mutual information is $C_I$-Lipschitz continuous and $\bar{C}_I$-smooth where $C_I$ and $\bar{C}_I$ are positive constants.*

*Proof.* From (8), we know that

$$
\begin{aligned}
&I(\theta; \{\bar{\mathcal{T}}_i^{\text{cri}}(\lambda_i \sim P_{\phi_\lambda}(\lambda))\}_{i=1}^{N^{\text{cri}}} | \{\mathcal{T}_i^{\text{cri}}\}_{i=1}^{N^{\text{cri}}}), \\
&= E_{\lambda_i}\left[\log \frac{1}{\sqrt{|(\sigma^*(\{\lambda_i^{\bar{\zeta}}\}_{i=1}^{N^{\text{cri}}}))^\top \sigma^*(\{\lambda_i^{\bar{\zeta}}\}_{i=1}^{N^{\text{cri}}})|}}\right] - \log E_{\lambda_i}\left[\frac{1}{\sqrt{|(\sigma^*(\{\lambda_i^{\bar{\zeta}}\}_{i=1}^{N^{\text{cri}}}))^\top \sigma^*(\{\lambda_i^{\bar{\zeta}}\}_{i=1}^{N^{\text{cri}}})|}}\right],
\end{aligned}
$$

where $\lambda_i^{\bar{\zeta}} = \mu_\lambda + \sigma_\lambda \bar{\zeta}_i$ and $\bar{\zeta}_i \sim \mathcal{N}(0,1)$. Therefore, we can get the gradient

$$
\begin{aligned}
&\nabla_{\phi_\lambda} I(\theta; \{\bar{\mathcal{T}}_i^{\text{cri}}(\lambda_i \sim P_{\phi_\lambda}(\lambda))\}_{i=1}^{N^{\text{cri}}} | \{\mathcal{T}_i^{\text{cri}}\}_{i=1}^{N^{\text{cri}}}), \\
&= \frac{E_{\bar{\zeta} \sim \mathcal{N}(0,1)}[\nabla_{\phi_\lambda} \sigma^*(\{\lambda_i^{\bar{\zeta}}\}_{i=1}^{N^{\text{cri}}})]}{||E_{\bar{\zeta} \sim \mathcal{N}(0,1)}[\sigma^*(\{\lambda_i^{\bar{\zeta}}\}_{i=1}^{N^{\text{cri}}})]||} - E_{\bar{\zeta} \sim \mathcal{N}(0,1)}\left[\frac{\nabla_{\phi_\lambda} \sigma^*(\{\lambda_i^{\bar{\zeta}}\}_{i=1}^{N^{\text{cri}}})}{||\sigma^*(\{\lambda_i^{\bar{\zeta}}\}_{i=1}^{N^{\text{cri}}})||}\right].
\end{aligned} \tag{10}
$$

Now, we consider the Hessian

$$
\begin{aligned}
&\nabla_{\phi_\lambda \phi_\lambda}^2 I(\theta; \{\bar{\mathcal{T}}_i^{\text{cri}}(\lambda_i \sim P_{\phi_\lambda}(\lambda))\}_{i=1}^{N^{\text{cri}}} | \{\mathcal{T}_i^{\text{cri}}\}_{i=1}^{N^{\text{cri}}}), \\
&= \nabla_{\phi_\lambda} \frac{E_{\bar{\zeta} \sim \mathcal{N}(0,1)}[\nabla_{\phi_\lambda} \sigma^*(\{\lambda_i^{\bar{\zeta}}\}_{i=1}^{N^{\text{cri}}})]}{||E_{\bar{\zeta} \sim \mathcal{N}(0,1)}[\sigma^*(\{\lambda_i^{\bar{\zeta}}\}_{i=1}^{N^{\text{cri}}})]||} - E_{\bar{\zeta} \sim \mathcal{N}(0,1)}\left[\nabla_{\phi_\lambda}\left[\frac{\nabla_{\phi_\lambda} \sigma^*(\{\lambda_i^{\bar{\zeta}}\}_{i=1}^{N^{\text{cri}}})}{||\sigma^*(\{\lambda_i^{\bar{\zeta}}\}_{i=1}^{N^{\text{cri}}})||}\right]\right],, \\
&= \frac{E_{\bar{\zeta} \sim \mathcal{N}(0,1)}[\nabla_{\phi_\lambda \phi_\lambda}^2 \sigma^*(\{\lambda_i^{\bar{\zeta}}\}_{i=1}^{N^{\text{cri}}})]}{||E_{\bar{\zeta} \sim \mathcal{N}(0,1)}[\sigma^*(\{\lambda_i^{\bar{\zeta}}\}_{i=1}^{N^{\text{cri}}})]||} \\
&\quad - \frac{E_{\bar{\zeta} \sim \mathcal{N}(0,1)}[\nabla_{\phi_\lambda} \sigma^*(\{\lambda_i^{\bar{\zeta}}\}_{i=1}^{N^{\text{cri}}})](E_{\bar{\zeta} \sim \mathcal{N}(0,1)}[\sigma^*(\{\lambda_i^{\bar{\zeta}}\}_{i=1}^{N^{\text{cri}}})])^\top E_{\bar{\zeta} \sim \mathcal{N}(0,1)}[\nabla_{\phi_\lambda} \sigma^*(\{\lambda_i^{\bar{\zeta}}\}_{i=1}^{N^{\text{cri}}})]}{||E_{\bar{\zeta} \sim \mathcal{N}(0,1)}[\sigma^*(\{\lambda_i^{\bar{\zeta}}\}_{i=1}^{N^{\text{cri}}})]||^3} \\
&\quad - E_{\bar{\zeta} \sim \mathcal{N}(0,1)}\left[\frac{\nabla_{\phi_\lambda \phi_\lambda}^2 \sigma^*(\{\lambda_i^{\bar{\zeta}}\}_{i=1}^{N^{\text{cri}}})}{||\sigma^*(\{\lambda_i^{\bar{\zeta}}\}_{i=1}^{N^{\text{cri}}})||} - \frac{\nabla_{\phi_\lambda} \sigma^*(\{\lambda_i^{\bar{\zeta}}\}_{i=1}^{N^{\text{cri}}})(\sigma^*(\{\lambda_i^{\bar{\zeta}}\}_{i=1}^{N^{\text{cri}}}))^\top \nabla_{\phi_\lambda} \sigma^*(\{\lambda_i^{\bar{\zeta}}\}_{i=1}^{N^{\text{cri}}})}{||\sigma^*(\{\lambda_i^{\bar{\zeta}}\}_{i=1}^{N^{\text{cri}}})||^3}\right].
\end{aligned} \tag{11}
$$

From (10), we know that if we can lower bound $||\sigma^*||$ and upper bound $||\nabla_{\phi_\lambda} \sigma^*||$, the norm of the gradient $\nabla_{\phi_\lambda} I(\theta; \{\bar{\mathcal{T}}_i^{\text{cri}}(\lambda_i \sim P_{\phi_\lambda}(\lambda))\}_{i=1}^{N^{\text{cri}}} | \{\mathcal{T}_i^{\text{cri}}\}_{i=1}^{N^{\text{cri}}})$ is bounded. From (11), we know that if we can lower bound $||\sigma^*||$ and upper bound $||\nabla_{\phi_\lambda} \sigma^*||$ and $||\nabla_{\phi_\lambda \phi_\lambda}^2 \sigma^*||$, the norm of the Hessian $||\nabla_{\phi_\lambda \phi_\lambda}^2 I(\theta; \{\bar{\mathcal{T}}_i^{\text{cri}}(\lambda_i \sim P_{\phi_\lambda}(\lambda))\}_{i=1}^{N^{\text{cri}}} | \{\mathcal{T}_i^{\text{cri}}\}_{i=1}^{N^{\text{cri}}})||$ is bounded. Note that $\lambda \in [0,1]$ is bounded within a compact set. Therefore, as long as we can prove that $\sigma^*$, $\nabla_{\phi_\lambda} \sigma^*$, and $\nabla_{\phi_\lambda \phi_\lambda}^2 \sigma^*$ are continuous in $\lambda$, their norms are both upper bounded and lower bounded. To show that $\sigma^*$, $\nabla_{\phi_\lambda} \sigma^*$, and $\nabla_{\phi_\lambda \phi_\lambda}^2 \sigma^*$ are continuous in $\lambda$, we can show that they are differentiable w.r.t. $\lambda$. Since $\phi_\lambda$ is differentiable w.r.t. $\lambda$, we only need to show that $\sigma^*$, $\nabla_{\phi_\lambda} \sigma^*$, and $\nabla_{\phi_\lambda \phi_\lambda}^2 \sigma^*$ are differentiable w.r.t. $\phi_\lambda$. This suffices to show that $\nabla_{\phi_\lambda} \sigma^*$, $\nabla_{\phi_\lambda \phi_\lambda}^2 \sigma^*$, and $\nabla_{\phi_\lambda \phi_\lambda \phi_\lambda}^3 \sigma^*$ exist.

From Lemma 1, we know that $\nabla_{\phi_\lambda} \sigma^*$ exists and

$$
\begin{aligned}
\nabla_{\phi_\lambda} \sigma^* = &-\left[\nabla_{\sigma\sigma}^2 E_{P_{\phi^*}(\theta)}[L(\theta, \{\bar{\mathcal{T}}_i^{\text{cri}}(\lambda_i^{\bar{\zeta}_j})\}_{i=1}^{N^{\text{cri}}}, \{\mathcal{T}_i^{\text{tr}}\}_{i=1}^{N^{\text{tr}}-N^{\text{cri}}})]\right]^{-1} \cdot \\
&\nabla_{\sigma\phi_\lambda}^2 E_{P_{\phi^*}(\theta)}[L(\theta, \{\bar{\mathcal{T}}_i^{\text{cri}}(\lambda_i^{\bar{\zeta}_j})\}_{i=1}^{N^{\text{cri}}}, \{\mathcal{T}_i^{\text{tr}}\}_{i=1}^{N^{\text{tr}}-N^{\text{cri}}})].
\end{aligned}
$$

Since $\log \pi_\theta$ is smooth in $\theta$ (Assumption 1), we can see that $L(\theta, \{\bar{\mathcal{T}}_i^{\mathrm{cri}}(\lambda_i^{\bar{\zeta}_j})\}_{i=1}^{N^{\mathrm{cri}}}, \{\mathcal{T}_i^{\mathrm{tr}}\}_{i=1}^{N^{\mathrm{tr}}-N^{\mathrm{cri}}})$ is also smooth in $\theta$. Since $\theta$ is smooth in $\sigma$, $L(\theta, \{\bar{\mathcal{T}}_i^{\mathrm{cri}}(\lambda_i^{\bar{\zeta}_j})\}_{i=1}^{N^{\mathrm{cri}}}, \{\mathcal{T}_i^{\mathrm{tr}}\}_{i=1}^{N^{\mathrm{tr}}-N^{\mathrm{cri}}})$ is also smooth in $\sigma$. Similarly, we can derive

$$\nabla_{\phi_\lambda \phi_\lambda}^2 \sigma^* = \left[\nabla_{\sigma\sigma}^2 E_{P_{\phi^*}(\theta)}[L(\theta, \{\bar{\mathcal{T}}_i^{\mathrm{cri}}(\lambda_i^{\bar{\zeta}_j})\}_{i=1}^{N^{\mathrm{cri}}}, \{\mathcal{T}_i^{\mathrm{tr}}\}_{i=1}^{N^{\mathrm{tr}}-N^{\mathrm{cri}}})]\right]^{-1}.$$
$$\nabla_{\sigma\sigma\phi_\lambda}^3 E_{P_{\phi^*}(\theta)}[L(\theta, \{\bar{\mathcal{T}}_i^{\mathrm{cri}}(\lambda_i^{\bar{\zeta}_j})\}_{i=1}^{N^{\mathrm{cri}}}, \{\mathcal{T}_i^{\mathrm{tr}}\}_{i=1}^{N^{\mathrm{tr}}-N^{\mathrm{cri}}})].$$
$$\left[\nabla_{\sigma\sigma}^2 E_{P_{\phi^*}(\theta)}[L(\theta, \{\bar{\mathcal{T}}_i^{\mathrm{cri}}(\lambda_i^{\bar{\zeta}_j})\}_{i=1}^{N^{\mathrm{cri}}}, \{\mathcal{T}_i^{\mathrm{tr}}\}_{i=1}^{N^{\mathrm{tr}}-N^{\mathrm{cri}}})]\right]^{-1}$$
$$- \left[\nabla_{\sigma\sigma}^2 E_{P_{\phi^*}(\theta)}[L(\theta, \{\bar{\mathcal{T}}_i^{\mathrm{cri}}(\lambda_i^{\bar{\zeta}_j})\}_{i=1}^{N^{\mathrm{cri}}}, \{\mathcal{T}_i^{\mathrm{tr}}\}_{i=1}^{N^{\mathrm{tr}}-N^{\mathrm{cri}}})]\right]^{-1}.$$
$$\nabla_{\sigma\phi_\lambda\phi_\lambda}^3 E_{P_{\phi^*}(\theta)}[L(\theta, \{\bar{\mathcal{T}}_i^{\mathrm{cri}}(\lambda_i^{\bar{\zeta}_j})\}_{i=1}^{N^{\mathrm{cri}}}, \{\mathcal{T}_i^{\mathrm{tr}}\}_{i=1}^{N^{\mathrm{tr}}-N^{\mathrm{cri}}})],$$

and similarly we can derive the expression of $\nabla_{\phi_\lambda\phi_\lambda\phi_\lambda}^3 \sigma^*$. Therefore, we can see that $||\sigma^*||$, $||\nabla_{\phi_\lambda} \sigma^*||$, and $||\nabla_{\phi_\lambda\phi_\lambda}^2 \sigma^*||$ are both lower bounded and upper bounded, and thus there exists positive constants $C_I$ and $\bar{C}_I$ such that $||\nabla_{\phi_\lambda} I(\theta; \{\bar{\mathcal{T}}_i^{\mathrm{cri}}(\lambda_i \sim P_{\phi_\lambda}(\lambda))\}_{i=1}^{N^{\mathrm{cri}}}|\{\mathcal{T}_i^{\mathrm{cri}}\}_{i=1}^{N^{\mathrm{cri}}})|| \leq C_I$ and $||\nabla_{\phi_\lambda\phi_\lambda}^2 I(\theta; \{\bar{\mathcal{T}}_i^{\mathrm{cri}}(\lambda_i \sim P_{\phi_\lambda}(\lambda))\}_{i=1}^{N^{\mathrm{cri}}}|\{\mathcal{T}_i^{\mathrm{cri}}\}_{i=1}^{N^{\mathrm{cri}}})|| \leq \bar{C}_I$. $\qquad \square$

For simplicity, we denote $f(\phi_{\lambda,k}) = I(\theta; \{\bar{\mathcal{T}}_i^{\mathrm{cri}}(\lambda_i \sim P_{\phi_\lambda,k}(\lambda))\}_{i=1}^{N^{\mathrm{cri}}}|\{\mathcal{T}_i^{\mathrm{cri}}\}_{i=1}^{N^{\mathrm{cri}}})$. Claim 1 shows that $f(\phi_{\lambda,k})$ is $\bar{C}_I$-smooth, therefore, we have that

$$f(\phi_{\lambda,k+1}) \geq f(\phi_{\lambda,k}) + \langle \nabla_{\phi_\lambda} f(\phi_{\lambda,k}), \phi_{\lambda,k+1} - \phi_{\lambda,k}\rangle - \frac{\bar{C}_I}{2}||\phi_{\lambda,k+1} - \phi_{\lambda,k}||^2,$$

$$\overset{(a)}{=} f(\phi_{\lambda,k}) + \beta||\nabla_{\phi_\lambda} f(\phi_{\lambda,k})||^2 - \frac{\bar{C}_I \beta^2}{2}||\nabla_{\phi_\lambda} f(\phi_{\lambda,k})||^2,$$

$$\overset{(b)}{\Rightarrow} \beta||\nabla_{\phi_\lambda} f(\phi_{\lambda,k})||^2 \leq f(\phi_{\lambda,k+1}) - f(\phi_{\lambda,k}) + \frac{\bar{C}_I C_I^2 \beta^2}{2}$$

$$\overset{(c)}{\Rightarrow} ||\nabla_{\phi_\lambda} f(\phi_{\lambda,k})||^2 \leq \frac{\bar{C}_I \sqrt{K}}{2}[f(\phi_{\lambda,k+1}) - f(\phi_{\lambda,k})] + \frac{C_I^2}{\sqrt{K}},$$

$$\Rightarrow \frac{1}{K} \sum_{k=0}^{K-1} ||\nabla_{\phi_\lambda} f(\phi_{\lambda,k})||^2 \leq \frac{\bar{C}_I}{2\sqrt{K}}[f(\phi_{\lambda,K}) - f(\phi_{\lambda,0})] + \frac{C_I^2}{\sqrt{K}},$$

where $(a)$ follows the fact that $\phi_{\lambda,k+1} = \phi_{\lambda,k} + \beta \nabla_{\phi_\lambda} f(\phi_{\lambda,k})$, $(b)$ follows the fact that $||\nabla_{\phi_\lambda} f(\phi_\lambda)|| \leq C_I$, and $(c)$ follows the fact that $\beta = \frac{2}{\bar{C}_I \sqrt{K}}$.

## I  PROOF OF THEOREM 2

This section proves Theorem 2 via two steps. Step (i): we prove that $I(\theta; \{\bar{\mathcal{T}}_i^{\mathrm{cri}}(\lambda_i \sim P_{\phi_\lambda,k}(\lambda))\}_{i=1}^{N^{\mathrm{cri}}}|\{\mathcal{T}_i^{\mathrm{cri}}\}_{i=1}^{N^{\mathrm{cri}}})$ is monotonically increasing in Claim 2. Step (ii): we provide that $I(\theta; \{\bar{\mathcal{T}}_i^{\mathrm{cri}}(\lambda_i \sim P_{\phi_\lambda,K}(\lambda))\}_{i=1}^{N^{\mathrm{cri}}}|\{\mathcal{T}_i^{\mathrm{cri}}\}_{i=1}^{N^{\mathrm{cri}}}) > 0$.

**Claim 2.** *If $\beta < \frac{2}{\bar{C}_I}$, the conditional mutual information is monotonically increasing, i.e., $I(\theta; \{\bar{\mathcal{T}}_i^{cri}(\lambda_i \sim P_{\phi_\lambda,k+1}(\lambda))\}_{i=1}^{N^{cri}}|\{\mathcal{T}_i^{cri}\}_{i=1}^{N^{cri}}) \geq I(\theta; \{\bar{\mathcal{T}}_i^{cri}(\lambda_i \sim P_{\phi_\lambda,k}(\lambda))\}_{i=1}^{N^{cri}}|\{\mathcal{T}_i^{cri}\}_{i=1}^{N^{cri}})$, and is strictly increasing if $||\nabla_{\phi_\lambda} I(\theta; \{\bar{\mathcal{T}}_i^{cri}(\lambda_i \sim P_{\phi_\lambda,k}(\lambda))\}_{i=1}^{N^{cri}}|\{\mathcal{T}_i^{cri}\}_{i=1}^{N^{cri}})|| > 0$.*

*Proof.* For simplicity, we denote $f(\phi_{\lambda,k}) = I(\theta; \{\bar{\mathcal{T}}_i^{\mathrm{cri}}(\lambda_i \sim P_{\phi_\lambda,k}(\lambda))\}_{i=1}^{N^{\mathrm{cri}}}|\{\mathcal{T}_i^{\mathrm{cri}}\}_{i=1}^{N^{\mathrm{cri}}})$. Therefore, we have that

$$f(\phi_{\lambda,k+1}) \overset{(a)}{\geq} f(\phi_{\lambda,k}) + \langle \nabla_{\phi_\lambda} f(\phi_{\lambda,k}), \phi_{\lambda,k+1} - \phi_{\lambda,k}\rangle - \frac{\bar{C}_I}{2}||\phi_{\lambda,k+1} - \phi_{\lambda,k}||^2,$$

$$\overset{(b)}{=} f(\phi_{\lambda,k}) + \beta||\nabla_{\phi_\lambda} f(\phi_{\lambda,k})||^2 - \frac{\bar{C}_I \beta^2}{2}||\nabla_{\phi_\lambda} f(\phi_{\lambda,k})||^2,$$

$$\Rightarrow f(\phi_{\lambda,k+1}) - f(\phi_{\lambda,k}) \geq \frac{2\beta - \bar{C}_I \beta^2}{2} ||\nabla_{\phi_\lambda} f(\phi_{\lambda,k})||^2 \geq 0 \tag{12}$$

where $(a)$ follows the fact that $f(\phi_\lambda)$ is $\bar{C}_I$-smooth (Claim 1), $(b)$ follows the fact that $\phi_{\lambda,k+1} = \phi_{\lambda,k} + \beta \nabla_{\phi_\lambda} f(\phi_{\lambda,k})$. from (12), we can see that $f(\phi_{\lambda,k+1}) \geq f(\phi_{\lambda,k})$. Moreover, $f(\phi_{\lambda,k+1}) > f(\phi_{\lambda,k})$ if $||\nabla_{\phi_\lambda} f(\phi_{\lambda,k})||^2 > 0$. $\square$

From Claim 2, we know that $I(\theta; \{\bar{\mathcal{T}}_i^{\text{cri}}(\lambda_i \sim P_{\phi_\lambda,K}(\lambda))\}_{i=1}^{N^{\text{cri}}} | \{\mathcal{T}_i^{\text{cri}}\}_{i=1}^{N^{\text{cri}}}) \geq I(\theta; \{\bar{\mathcal{T}}_i^{\text{cri}}(\lambda_i \sim P_{\phi_\lambda,0}(\lambda))\}_{i=1}^{N^{\text{cri}}} | \{\mathcal{T}_i^{\text{cri}}\}_{i=1}^{N^{\text{cri}}})$. The only situation where $I(\theta; \{\bar{\mathcal{T}}_i^{\text{cri}}(\lambda_i \sim P_{\phi_\lambda,K}(\lambda))\}_{i=1}^{N^{\text{cri}}} | \{\mathcal{T}_i^{\text{cri}}\}_{i=1}^{N^{\text{cri}}}) = I(\theta; \{\bar{\mathcal{T}}_i^{\text{cri}}(\lambda_i \sim P_{\phi_\lambda,0}(\lambda))\}_{i=1}^{N^{\text{cri}}} | \{\mathcal{T}_i^{\text{cri}}\}_{i=1}^{N^{\text{cri}}})$ is that $\nabla_{\phi_\lambda} I(\theta; \{\bar{\mathcal{T}}_i^{\text{cri}}(\lambda_i \sim P_{\phi_\lambda,0}(\lambda))\}_{i=1}^{N^{\text{cri}}} | \{\mathcal{T}_i^{\text{cri}}\}_{i=1}^{N^{\text{cri}}}) = 0$, i.e., the initialization is a stationary point, which is of zero probability. Therefore, we know that $I(\theta; \{\bar{\mathcal{T}}_i^{\text{cri}}(\lambda_i \sim P_{\phi_\lambda,K}(\lambda))\}_{i=1}^{N^{\text{cri}}} | \{\mathcal{T}_i^{\text{cri}}\}_{i=1}^{N^{\text{cri}}}) > I(\theta; \{\bar{\mathcal{T}}_i^{\text{cri}}(\lambda_i \sim P_{\phi_\lambda,0}(\lambda))\}_{i=1}^{N^{\text{cri}}} | \{\mathcal{T}_i^{\text{cri}}\}_{i=1}^{N^{\text{cri}}})$. Since conditional mutual information is always nonnegative (Wyner, 1978), we know that $I(\theta; \{\bar{\mathcal{T}}_i^{\text{cri}}(\lambda_i \sim P_{\phi_\lambda,K}(\lambda))\}_{i=1}^{N^{\text{cri}}} | \{\mathcal{T}_i^{\text{cri}}\}_{i=1}^{N^{\text{cri}}}) > I(\theta; \{\bar{\mathcal{T}}_i^{\text{cri}}(\lambda_i \sim P_{\phi_\lambda,0}(\lambda))\}_{i=1}^{N^{\text{cri}}} | \{\mathcal{T}_i^{\text{cri}}\}_{i=1}^{N^{\text{cri}}}) \geq 0$.

## J   THE TASK INFORMATION OF THE NON-CRITICAL TASKS STORED IN THE META-PARAMETER DOES NOT CHANGE AFTER THE TASK AUGMENTATION

This section shows that the task information of the non-critical tasks stored in the meta-parameter does not change after the task augmentation. In brief, we prove that the mutual information between the meta-parameter and the non-critical tasks remains unchanged even if the mutual information between the meta-parameter and the critical tasks increases after task augmentation (i.e., the conditional mutual information is positive).

Suppose we augment the critical tasks $\{\mathcal{T}_i^{\text{cri}}\}_{i=1}^{N^{\text{cri}}}$ to $\{\bar{\mathcal{T}}_i^{\text{cri}}\}_{i=1}^{N^{\text{cri}}}$. Note that the difference between $\{\mathcal{T}_i^{\text{cri}}\}_{i=1}^{N^{\text{cri}}}$ and $\{\bar{\mathcal{T}}_i^{\text{cri}}\}_{i=1}^{N^{\text{cri}}}$ is that they have different distributions, i.e., $P(\{\mathcal{T}_i^{\text{cri}}\}_{i=1}^{N^{\text{cri}}})$ and $P(\{\bar{\mathcal{T}}_i^{\text{cri}}\}_{i=1}^{N^{\text{cri}}})$. Therefore, we use $A$ to generally represent the critical tasks (either before augmentation or after augmentation), and use $P(A = \{\mathcal{T}_i^{\text{cri}}\}_{i=1}^{N^{\text{cri}}})$ and $P(A = \{\bar{\mathcal{T}}_i^{\text{cri}}\}_{i=1}^{N^{\text{cri}}})$ to respectively denote that $A$ follows the distribution of $\{\mathcal{T}_i^{\text{cri}}\}_{i=1}^{N^{\text{cri}}}$ and $A$ follows the distribution of $\{\bar{\mathcal{T}}_i^{\text{cri}}\}_{i=1}^{N^{\text{cri}}}$. We now quantify the change of the mutual information between the meta-parameter and the non-critical tasks $\{\mathcal{T}_i^{\text{tr}}\}_{i=1}^{N^{\text{tr}}-N^{\text{cri}}}$:

$$I(\theta; \{\mathcal{T}_i^{\text{tr}}\}_{i=1}^{N^{\text{tr}}-N^{\text{cri}}} | \{\bar{\mathcal{T}}_i^{\text{cri}}\}_{i=1}^{N^{\text{cri}}}) - I(\theta; \{\mathcal{T}_i^{\text{tr}}\}_{i=1}^{N^{\text{tr}}-N^{\text{cri}}} | \{\mathcal{T}_i^{\text{cri}}\}_{i=1}^{N^{\text{cri}}}),$$

$$\overset{(a)}{=} \int P(\theta, \{\mathcal{T}_i^{\text{tr}}\}_{i=1}^{N^{\text{tr}}-N^{\text{cri}}}, \{\bar{\mathcal{T}}_i^{\text{cri}}\}_{i=1}^{N^{\text{cri}}}) \cdot$$

$$\log \frac{P(\theta, \{\mathcal{T}_i^{\text{tr}}\}_{i=1}^{N^{\text{tr}}-N^{\text{cri}}} | \{\bar{\mathcal{T}}_i^{\text{cri}}\}_{i=1}^{N^{\text{cri}}})}{P(\theta | \{\bar{\mathcal{T}}_i^{\text{cri}}\}_{i=1}^{N^{\text{cri}}}) P(\{\mathcal{T}_i^{\text{tr}}\}_{i=1}^{N^{\text{tr}}-N^{\text{cri}}} | \{\bar{\mathcal{T}}_i^{\text{cri}}\}_{i=1}^{N^{\text{cri}}})} d\theta (d\{\mathcal{T}_i^{\text{tr}}\}_{i=1}^{N^{\text{tr}}-N^{\text{cri}}})(d\{\bar{\mathcal{T}}_i^{\text{cri}}\}_{i=1}^{N^{\text{cri}}})$$

$$- \int P(\theta, \{\mathcal{T}_i^{\text{tr}}\}_{i=1}^{N^{\text{tr}}-N^{\text{cri}}}, \{\mathcal{T}_i^{\text{cri}}\}_{i=1}^{N^{\text{cri}}}) \cdot$$

$$\log \frac{P(\theta, \{\mathcal{T}_i^{\text{tr}}\}_{i=1}^{N^{\text{tr}}-N^{\text{cri}}} | \{\mathcal{T}_i^{\text{cri}}\}_{i=1}^{N^{\text{cri}}})}{P(\theta | \{\mathcal{T}_i^{\text{cri}}\}_{i=1}^{N^{\text{cri}}}) P(\{\mathcal{T}_i^{\text{tr}}\}_{i=1}^{N^{\text{tr}}-N^{\text{cri}}} | \{\mathcal{T}_i^{\text{cri}}\}_{i=1}^{N^{\text{cri}}})} d\theta (d\{\mathcal{T}_i^{\text{tr}}\}_{i=1}^{N^{\text{tr}}-N^{\text{cri}}})(d\{\mathcal{T}_i^{\text{cri}}\}_{i=1}^{N^{\text{cri}}})$$

$$= \int P(\theta, \{\mathcal{T}_i^{\text{tr}}\}_{i=1}^{N^{\text{tr}}-N^{\text{cri}}}, \{\bar{\mathcal{T}}_i^{\text{cri}}\}_{i=1}^{N^{\text{cri}}}) \log \frac{P(\theta | \{\mathcal{T}_i^{\text{tr}}\}_{i=1}^{N^{\text{tr}}-N^{\text{cri}}}, \{\bar{\mathcal{T}}_i^{\text{cri}}\}_{i=1}^{N^{\text{cri}}})}{P(\theta | \{\bar{\mathcal{T}}_i^{\text{cri}}\}_{i=1}^{N^{\text{cri}}})} d\theta (d\{\mathcal{T}_i^{\text{tr}}\}_{i=1}^{N^{\text{tr}}-N^{\text{cri}}})(d\{\bar{\mathcal{T}}_i^{\text{cri}}\}_{i=1}^{N^{\text{cri}}})$$

$$- \int P(\theta, \{\mathcal{T}_i^{\text{tr}}\}_{i=1}^{N^{\text{tr}}-N^{\text{cri}}}, \{\mathcal{T}_i^{\text{cri}}\}_{i=1}^{N^{\text{cri}}}) \log \frac{P(\theta | \{\mathcal{T}_i^{\text{tr}}\}_{i=1}^{N^{\text{tr}}-N^{\text{cri}}}, \{\mathcal{T}_i^{\text{cri}}\}_{i=1}^{N^{\text{cri}}})}{P(\theta | \{\mathcal{T}_i^{\text{cri}}\}_{i=1}^{N^{\text{cri}}})} d\theta (d\{\mathcal{T}_i^{\text{tr}}\}_{i=1}^{N^{\text{tr}}-N^{\text{cri}}})(d\{\mathcal{T}_i^{\text{cri}}\}_{i=1}^{N^{\text{cri}}}),$$

$$\overset{(b)}{=} \int P(\theta | \{\mathcal{T}_i^{\text{tr}}\}_{i=1}^{N^{\text{tr}}-N^{\text{cri}}}, A) P(\{\mathcal{T}_i^{\text{tr}}\}_{i=1}^{N^{\text{tr}}-N^{\text{cri}}}) P(A = \{\bar{\mathcal{T}}_i^{\text{cri}}\}_{i=1}^{N^{\text{cri}}}) \cdot$$

$$\log \frac{P(\theta | \{\mathcal{T}_i^{\text{tr}}\}_{i=1}^{N^{\text{tr}}-N^{\text{cri}}}, A)}{P(\theta | A)} d\theta (d\{\mathcal{T}_i^{\text{tr}}\}_{i=1}^{N^{\text{tr}}-N^{\text{cri}}})(dA)$$

$$- \int P(\theta | \{\mathcal{T}_i^{\text{tr}}\}_{i=1}^{N^{\text{tr}}-N^{\text{cri}}}, A) P(\{\mathcal{T}_i^{\text{tr}}\}_{i=1}^{N^{\text{tr}}-N^{\text{cri}}}) P(A = \{\mathcal{T}_i^{\text{cri}}\}_{i=1}^{N^{\text{cri}}}) \cdot$$

$$\log \frac{P(\theta|\{\mathcal{T}_i^{\mathrm{tr}}\}_{i=1}^{N^{\mathrm{tr}}-N^{\mathrm{cri}}}, A)}{P(\theta|A)} d\theta (d\{\mathcal{T}_i^{\mathrm{tr}}\}_{i=1}^{N^{\mathrm{tr}}-N^{\mathrm{cri}}})(dA),$$

$$= \int P(\theta|\{\mathcal{T}_i^{\mathrm{tr}}\}_{i=1}^{N^{\mathrm{tr}}-N^{\mathrm{cri}}}, A) P(\{\mathcal{T}_i^{\mathrm{tr}}\}_{i=1}^{N^{\mathrm{tr}}-N^{\mathrm{cri}}}) \Big[ P(A = \{\bar{\mathcal{T}}_i^{\mathrm{cri}}\}_{i=1}^{N^{\mathrm{cri}}}) - P(A = \{\mathcal{T}_i^{\mathrm{cri}}\}_{i=1}^{N^{\mathrm{cri}}}) \Big] \cdot$$

$$\log \frac{P(\theta|\{\mathcal{T}_i^{\mathrm{tr}}\}_{i=1}^{N^{\mathrm{tr}}-N^{\mathrm{cri}}}, A)}{P(\theta|A)} d\theta (d\{\mathcal{T}_i^{\mathrm{tr}}\}_{i=1}^{N^{\mathrm{tr}}-N^{\mathrm{cri}}})(dA),$$

$$= \int P(\theta|\{\mathcal{T}_i^{\mathrm{tr}}\}_{i=1}^{N^{\mathrm{tr}}-N^{\mathrm{cri}}}, A) P(\{\mathcal{T}_i^{\mathrm{tr}}\}_{i=1}^{N^{\mathrm{tr}}-N^{\mathrm{cri}}}) \Big[ P(A = \{\bar{\mathcal{T}}_i^{\mathrm{cri}}\}_{i=1}^{N^{\mathrm{cri}}}) - P(A = \{\mathcal{T}_i^{\mathrm{cri}}\}_{i=1}^{N^{\mathrm{cri}}}) \Big] \cdot$$

$$\log \frac{P(\theta|\{\mathcal{T}_i^{\mathrm{tr}}\}_{i=1}^{N^{\mathrm{tr}}-N^{\mathrm{cri}}}, A)}{\int P(\theta|\{\mathcal{T}_i^{\mathrm{tr}}\}_{i=1}^{N^{\mathrm{tr}}-N^{\mathrm{cri}}}, A) P(\{\mathcal{T}_i^{\mathrm{tr}}\}_{i=1}^{N^{\mathrm{tr}}-N^{\mathrm{cri}}})(d\{\mathcal{T}_i^{\mathrm{tr}}\}_{i=1}^{N^{\mathrm{tr}}-N^{\mathrm{cri}}})} d\theta (d\{\mathcal{T}_i^{\mathrm{tr}}\}_{i=1}^{N^{\mathrm{tr}}-N^{\mathrm{cri}}})(dA),$$

$$\overset{(c)}{=} \int P(\theta|\{\mathcal{T}_i^{\mathrm{tr}}\}_{i=1}^{N^{\mathrm{tr}}-N^{\mathrm{cri}}}, A) P(\{\mathcal{T}_i^{\mathrm{tr}}\}_{i=1}^{N^{\mathrm{tr}}-N^{\mathrm{cri}}}) \Big[ P(A = \{\bar{\mathcal{T}}_i^{\mathrm{cri}}\}_{i=1}^{N^{\mathrm{cri}}}) - P(A = \{\mathcal{T}_i^{\mathrm{cri}}\}_{i=1}^{N^{\mathrm{cri}}}) \Big] \cdot$$

$$\log 1 d\theta (d\{\mathcal{T}_i^{\mathrm{tr}}\}_{i=1}^{N^{\mathrm{tr}}-N^{\mathrm{cri}}})(dA),$$

$$= 0, \tag{13}$$

where $(a)$ follows the definition of conditional mutual information (Wyner, 1978), $(b)$ follows the fact that the critical tasks and the non-critical tasks are independent (i.e., $P(\theta, \{\mathcal{T}_i^{\mathrm{tr}}\}_{i=1}^{N^{\mathrm{tr}}-N^{\mathrm{cri}}}, A) = P(\theta|\{\mathcal{T}_i^{\mathrm{tr}}\}_{i=1}^{N^{\mathrm{tr}}-N^{\mathrm{cri}}}, A) P(\{\mathcal{T}_i^{\mathrm{tr}}\}_{i=1}^{N^{\mathrm{tr}}-N^{\mathrm{cri}}}, A) = P(\theta|\{\mathcal{T}_i^{\mathrm{tr}}\}_{i=1}^{N^{\mathrm{tr}}-N^{\mathrm{cri}}}, A) P(\{\mathcal{T}_i^{\mathrm{tr}}\}_{i=1}^{N^{\mathrm{tr}}-N^{\mathrm{cri}}}) P(A))$, and $(c)$ follows the fact that the non-critical tasks $\{\mathcal{T}_i^{\mathrm{tr}}\}_{i=1}^{N^{\mathrm{tr}}-N^{\mathrm{cri}}}$ are given and thus $P(\{\mathcal{T}_i^{\mathrm{tr}}\}_{i=1}^{N^{\mathrm{tr}}-N^{\mathrm{cri}}}) = 1$.

From (13), we can see that $I(\theta; \{\mathcal{T}_i^{\mathrm{tr}}\}_{i=1}^{N^{\mathrm{tr}}-N^{\mathrm{cri}}}|\{\bar{\mathcal{T}}_i^{\mathrm{cri}}\}_{i=1}^{N^{\mathrm{cri}}}) - I(\theta; \{\mathcal{T}_i^{\mathrm{tr}}\}_{i=1}^{N^{\mathrm{tr}}-N^{\mathrm{cri}}}|\{\mathcal{T}_i^{\mathrm{cri}}\}_{i=1}^{N^{\mathrm{cri}}}) = 0$, and thus the information of the non-critical tasks stored in the meta-parameter does not change after the task augmentation.

## K  PROOF OF LEMMA 2

In this section, we prove that the learned augmentation $P_{\phi_{\lambda,K}}(\lambda)$ imposes a quadratic regularization on the original meta-objective. Let's first consider $\bar{J}_i^{\mathrm{cri}}(\pi_i^{\mathrm{cri}}(\theta), \lambda_i)$. We use $\phi_i$ to denote the parameter of the task-specific adaptation, i.e., $\pi_{\phi_i} = \pi_i^{\mathrm{cri}}(\theta)$. Since we use MAML to compute the task-specific adaptation, we know that $\phi_i = \theta - \alpha \nabla_\theta J_i^{\mathrm{cri}}(\pi_\theta)$. We use $\bar{s}(\lambda)$ to represent $\bar{s}$ to highlight the mixing coefficient $\lambda$. Note that the action $\bar{a}$ indirectly depends on $\lambda$ because $\lambda$ will affect $\bar{s}$ and $\bar{s}$ will affect the distribution of $\pi_{\phi_i}(\cdot|\bar{s})$. However, we do not need to directly reason about how $\lambda$ affects $\bar{a}$ because we can capture this relation by analyzing how $\bar{s}$ affects the distribution $\pi_\phi(\cdot|\bar{s})$. Therefore, we still use the notation $\bar{a}$ instead of $\bar{a}(\lambda)$. The RL objective of the augmented task $\bar{\mathcal{T}}_i^{\mathrm{cri}}(\lambda_i)$ is:

$$E_{\lambda_i \sim \mathcal{N}(\mu_{\lambda,K}, \sigma_{\lambda,K}^2)} \Big[ \bar{J}_i^{\mathrm{cri}}(\pi_i^{\mathrm{cri}}(\theta), \lambda_i) \Big],$$

$$= E_{(s,a),(s',a') \sim \rho^{\pi_{\phi_i}}, \lambda_i \sim \mathcal{N}(\mu_{\lambda,K}, \sigma_{\lambda,K}^2)} \Big[ \log \pi_{\phi_i}(\bar{a}|\bar{s}(\lambda_i)) A_i^{\pi_{\phi_i}}(\bar{s}(\lambda_i), \bar{a}) \Big].$$

Let $x_i = 1 - \lambda_i$ and $F_i(x_i) = \log \pi_{\phi_i}(\bar{a}|\bar{s}(\lambda_i)) A_i^{\pi_{\phi_i}}(\bar{s}(\lambda_i), \bar{a})$, therefore, the second-order approximation of $F_i(x_i)$ is:

$$F_i(x_i) \approx F_i(0) + F_i'(0)x_i + \frac{1}{2} F_i''(0)x_i^2. \tag{14}$$

We now derive the expression of $F_i'(0)$ and $F_i''(0)$.

$$F_i'(x_i) = \frac{\partial F_i(x_i)}{\partial \bar{s}(\lambda_i)} \frac{\partial \bar{s}(\lambda_i)}{\partial x_i},$$

$$= \Big[ \nabla_s \log \pi_{\phi_i}(\bar{a}|\bar{s}(\lambda_i)) \cdot A_i^{\pi_{\phi_i}}(\bar{s}(\lambda_i), \bar{a}) + \log \pi_{\phi_i}(\bar{a}|\bar{s}(\lambda_i)) \cdot \nabla_s A_i^{\pi_{\phi_i}}(\bar{s}(\lambda_i), \bar{a}) \Big]^\top (s' - s)$$

$$\Rightarrow F_i'(0) = \Big[ \nabla_s \log \pi_{\phi_i}(a|s) \cdot A_i^{\pi_{\phi_i}}(s, a) + \log \pi_{\phi_i}(a|s) \cdot \nabla_s A_i^{\pi_{\phi_i}}(s, a) \Big]^\top (s' - s). \tag{15}$$

We now reason about the second-order derivative:

$$F_i''(x_i) = \frac{\partial}{\partial x_i}\Big[\nabla_s \log \pi_{\phi_i}(\bar{a}|\bar{s}(\lambda_i)) \cdot A_i^{\pi_{\phi_i}}(\bar{s}(\lambda_i), \bar{a}) + \log \pi_{\phi_i}(\bar{a}|\bar{s}(\lambda_i)) \cdot \nabla_s A_i^{\pi_{\phi_i}}(\bar{s}(\lambda_i), \bar{a})\Big](s'-s)$$

$$= (s'-s)^\top \Big[\nabla_{ss}^2 \log \pi_{\phi_i}(\bar{a}|\bar{s}(\lambda_i)) \cdot A_i^{\pi_{\phi_i}}(\bar{s}(\lambda_i), \bar{a}) + 2(\nabla_s \log \pi_{\phi_i}(\bar{a}|\bar{s}(\lambda_i)))^\top \cdot \nabla_s A_i^{\pi_{\phi_i}}(\bar{s}(\lambda_i), \bar{a})$$

$$+ \log \pi_{\phi_i}(\bar{a}|\bar{s}(\lambda_i)) \cdot \nabla_{ss}^2 A_i^{\pi_{\phi_i}}(\bar{s}(\lambda_i), \bar{a})\Big](s'-s),$$

$$\Rightarrow F_i''(0) = (s'-s)^\top \Big[\nabla_{ss}^2 \log \pi_{\phi_i}(a|s) \cdot A_i^{\pi_{\phi_i}}(s, a) + 2(\nabla_s \log \pi_{\phi_i}(a|s))^\top \nabla_s A_i^{\pi_{\phi_i}}(s, a)$$

$$+ \log \pi_{\phi_i}(a|s)\nabla_{ss}^2 A_i^{\pi_{\phi_i}}(s, a)\Big](s'-s). \tag{16}$$

By plugging (15)-(16) into (14), we have that

$$F_i(x_i) \approx \log \pi_{\phi_i}(a|s) A_i^{\pi_{\phi_i}}(s, a)$$

$$+ \Big[\nabla_s \log \pi_{\phi_i}(a|s) \cdot A_i^{\pi_{\phi_i}}(s, a) + \log \pi_{\phi_i}(a|s) \cdot \nabla_s A_i^{\pi_{\phi_i}}(s, a)\Big]^\top (s'-s)(1-\lambda_i)$$

$$+ (s'-s)^\top \Big[\nabla_{ss}^2 \log \pi_{\phi_i}(a|s) \cdot A_i^{\pi_{\phi_i}}(s, a) + 2(\nabla_s \log \pi_{\phi_i}(a|s))^\top \nabla_s A_i^{\pi_{\phi_i}}(s, a)$$

$$+ \log \pi_{\phi_i}(a|s)\nabla_{ss}^2 A_i^{\pi_{\phi_i}}(s, a)\Big](s'-s)(1-\lambda_i)^2,$$

$$= \log \pi_{\phi_i}(a|s) A_i^{\pi_{\phi_i}}(s, a) + C_{\lambda_i}(s, a) + (s'-s)^\top A_i^{\pi_{\phi_i}}(s, a)\Big[\nabla_{ss}^2 \log \pi_{\phi_i}(a|s)\Big](s'-s)(1-\lambda_i)^2, \tag{17}$$

where $C_{\lambda_i}(s_j, a_j) = \Big[\nabla_s \log \pi_{\phi_i}(a|s) \cdot A_i^{\pi_{\phi_i}}(s, a) + \log \pi_{\phi_i}(a|s) \cdot \nabla_s A_i^{\pi_{\phi_i}}(s, a)\Big]^\top (s'-s)(1-\lambda_i) + (s'-s)^\top \Big[2(\nabla_s \log \pi_{\phi_i}(a|s))^\top \nabla_s A_i^{\pi_{\phi_i}}(s, a) + \log \pi_{\phi_i}(a|s)\nabla_{ss}^2 A_i^{\pi_{\phi_i}}(s, a)\Big](s'-s)(1-\lambda_i)^2$.

Now we take a look at the term $\nabla_{ss}^2 \log \pi_{\phi_i}(a_j|s_j)$. Recall that the softmax policy parameterization $\pi_{\phi_i}(a|s) = \frac{e^{\phi_i^\top f(s,a)}}{\sum_{a' \in \mathcal{A}} e^{\phi_i^\top f(s,a')}}$, therefore we have that

$$\nabla_{ss}^2 \log \pi_{\phi_i}(a|s) = \nabla_{ss}^2 \Big[\phi_i^\top f(s, a) - \log \sum_{a' \in \mathcal{A}} e^{\phi_i^\top f(s,a')}\Big],$$

$$= \phi_i^\top \nabla_{ss}^2 f(s, a) - \frac{\sum_{a' \in \mathcal{A}} \phi_i^\top \nabla_{ss}^2 f(s, a') e^{\phi_i^\top f(s,a')} + \phi_i^\top (\nabla_s f(s, a'))(\nabla_s f(s, a'))^\top e^{\phi_i^\top f(s,a')} \phi_i}{\sum_{a' \in \mathcal{A}} e^{\phi_i^\top f(s,a')}}$$

$$+ \frac{(\sum_{a' \in \mathcal{A}} \phi_i^\top \nabla_s f(s, a') e^{\phi_i^\top f(s,a')})^2}{(\sum_{a' \in \mathcal{A}} e^{\phi_i^\top f(s,a')})^2},$$

$$= \phi_i^\top \nabla_{ss}^2 f(s, a) - \frac{\sum_{a' \in \mathcal{A}} \phi_i^\top \nabla_{ss}^2 f(s, a') e^{\phi_i^\top f(s,a')}}{\sum_{a' \in \mathcal{A}} e^{\phi_i^\top f(s,a')}}$$

$$- \phi_i^\top \Big[\frac{[\sum_{a' \in \mathcal{A}}(\nabla_s f(s, a'))(\nabla_s f(s, a'))^\top e^{\phi_i^\top f(s,a')}](\sum_{a' \in \mathcal{A}} e^{\phi_i^\top f(s,a')}) - (\sum_{a' \in \mathcal{A}} \nabla_s f(s, a') e^{\phi_i^\top f(s,a')})^2}{(\sum_{a' \in \mathcal{A}} e^{\phi_i^\top f(s,a')})^2}\Big]\phi_i,$$

$$= \phi_i^\top \nabla_{ss}^2 f(s, a) - \frac{\sum_{a' \in \mathcal{A}} \phi_i^\top \nabla_{ss}^2 f(s, a') e^{\phi_i^\top f(s,a')}}{\sum_{a' \in \mathcal{A}} e^{\phi_i^\top f(s,a')}} - \phi_i^\top H(s, a)\phi_i, \tag{18}$$

where $H(s, a) = \frac{[\sum_{a' \in \mathcal{A}}(\nabla_s f(s, a'))(\nabla_s f(s, a'))^\top e^{\phi_i^\top f(s,a')}](\sum_{a' \in \mathcal{A}} e^{\phi_i^\top f(s,a')}) - (\sum_{a' \in \mathcal{A}} \nabla_s f(s, a') e^{\phi_i^\top f(s,a')})^2}{(\sum_{a' \in \mathcal{A}} e^{\phi_i^\top f(s,a')})^2} \succ 0$ by Cauchy-Schwartz inequality. By plugging (18) in to (17), we have that

$$F_i(x_i) \approx \log \pi_{\phi_i}(a|s) A_i^{\pi_{\phi_i}}(s, a) + C_{\lambda_i}(s, a) + A_i^{\pi_{\phi_i}}(s, a)(s'-s)^\top \cdot$$

$$\Big[\phi_i^\top \nabla_{ss}^2 f(s, a) - \frac{\sum_{a' \in \mathcal{A}} \phi_i^\top \nabla_{ss}^2 f(s, a') e^{\phi_i^\top f(s,a')}}{\sum_{a' \in \mathcal{A}} e^{\phi_i^\top f(s,a')}} - \phi_i^\top H(s, a)\phi_i\Big](s'-s)x_i^2,$$

$$= \log \pi_{\phi_i}(a|s) A_i^{\pi_{\phi_i}}(s, a) + \bar{C}_{\lambda_i}(s, a) - \phi_i^\top \bar{H}_{\lambda_i}^{\text{cri}}(s, a)\phi_i,$$

$$
\stackrel{(d)}{=} \log \pi_{\phi_i}(a|s) A_i^{\pi_{\phi_i}}(s,a) + \bar{C}_{\lambda_i}(s,a) - (\theta - \alpha \nabla_\theta J_i^{\mathrm{cri}}(\pi_\theta))^\top \bar{H}_{\lambda_i}^{\mathrm{cri}}(s,a)(\theta - \alpha \nabla_\theta J_i^{\mathrm{cri}}(\pi_\theta)),
$$
$$
= \log \pi_{\phi_i}(a|s) A_i^{\pi_{\phi_i}}(s,a) + \tilde{C}_{\lambda_i}(s,a) - \theta^\top \bar{H}_{\lambda_i}^{\mathrm{cri}}(s,a)\theta, \tag{19}
$$

where $(d)$ follows the fact that $\phi_i = \theta - \alpha \nabla_\theta J_i^{\mathrm{cri}}(\pi_\theta)$, $\bar{C}_{\lambda_i}(s,a) = C_{\lambda_i}(s,a) + A_i^{\pi_{\phi_i}}(s,a)(s'-s)^\top \left[ \phi_i^\top \nabla_{ss}^2 f(s,a) - \frac{\sum_{a'\in\mathcal{A}} \phi_i^\top \nabla_{ss}^2 f(s,a') e^{\phi_i^\top f(s,a')}}{\sum_{a'\in\mathcal{A}} e^{\phi_i^\top f(s,a')}} \right](s'-s)x_i^2$, $\bar{H}_{\lambda_i}^{\mathrm{cri}}(s,a) = A_i^{\pi_{\phi_i}}(s,a)H(s,a)(s'-s)x_i^2 \succ 0$ given that $H(s,a) \succ 0$, and $\tilde{C}_{\lambda_i}(s,a) = \bar{C}_{\lambda_i}(s,a) - \alpha^2(\nabla_\theta J_i^{\mathrm{cri}}(\pi_\theta))^\top \bar{H}_{\lambda_i}^{\mathrm{cri}}(s,a)(\nabla_\theta J_i^{\mathrm{cri}}(\pi_\theta))$.

Therefore, we have that

$$
\bar{J}_i^{\mathrm{cri}}(\pi_i^{\mathrm{cri}}(\theta), \lambda_i) = E_{(s,a),(s',a')\sim\rho^{\pi_{\phi_i}}}[F_i(x_i)]
$$
$$
\stackrel{(e)}{=} E_{(s,a)\sim\rho^{\pi_{\phi_i}}} \left[ \log \pi_{\phi_i}(a|s)A_i^{\pi_{\phi_i}}(s,a) + \tilde{C}_{\lambda_i}(s,a) - \theta^\top \bar{H}_{\lambda_i}^{\mathrm{cri}}(s,a)\theta \right],
$$
$$
= J_i^{\mathrm{cri}}(\pi_i^{\mathrm{cri}}(\theta)) + \tilde{C}_{\lambda_i} - \theta^\top \bar{H}_{\lambda_i}^{\mathrm{cri}}\theta,
$$

where $(e)$ follows (19), $\tilde{C}_{\lambda_i} = E_{(s,a)\sim\rho^{\pi_{\phi_i}}}[\tilde{C}_{\lambda_i}(s,a)]$, and $\bar{H}_{\lambda_i}^{\mathrm{cri}} = E_{(s,a)\sim\rho^{\pi_{\phi_i}}}[\bar{H}_{\lambda_i}^{\mathrm{cri}}(s,a)] \succ 0$ given that $\bar{H}_{\lambda_i}^{\mathrm{cri}}(s,a) \succ 0$. If we only consider the second-order term, we can see that $\bar{J}_i^{\mathrm{cri}}(\pi_i^{\mathrm{cri}}(\theta), \lambda_i) \approx J_i^{\mathrm{cri}}(\pi_i^{\mathrm{cri}}(\theta)) - \theta^\top \bar{H}_{\lambda_i}^{\mathrm{cri}}\theta$. Therefore, we have that $L(\theta, \{\bar{\mathcal{T}}_i^{\mathrm{cri}}(\lambda_i)\}_{i=1}^{N^{\mathrm{cri}}}, \{\mathcal{T}_i^{\mathrm{tr}}\}_{i=1}^{N^{\mathrm{tr}}-N^{\mathrm{cri}}}) \approx L(\theta, \{\mathcal{T}_i^{\mathrm{tr}}\}_{i=1}^{N^{\mathrm{tr}}}) - \theta^\top(\sum_{i=1}^{N^{\mathrm{cri}}} \bar{H}_{\lambda_i}^{\mathrm{cri}})\theta$ where $(\sum_{i=1}^{N^{\mathrm{cri}}} \bar{H}_{\lambda_i}^{\mathrm{cri}}) \succ 0$ given that $\bar{H}_{\lambda_i}^{\mathrm{cri}} \succ 0$. Thus we have that $E_{\lambda_i\sim P_{\phi_{\lambda,K}}(\lambda)}[L(\theta, \{\bar{\mathcal{T}}_i^{\mathrm{cri}}(\lambda_i)\}_{i=1}^{N^{\mathrm{cri}}}, \{\mathcal{T}_i^{\mathrm{tr}}\}_{i=1}^{N^{\mathrm{tr}}-N^{\mathrm{cri}}})] \approx L(\theta, \{\mathcal{T}_i^{\mathrm{tr}}\}_{i=1}^{N^{\mathrm{tr}}}) - \theta^\top(\sum_{i=1}^{N^{\mathrm{cri}}} \bar{H}_i^{\mathrm{cri}})\theta$ where $\bar{H}_i^{\mathrm{cri}} = E_{\lambda_i\sim P_{\phi_{\lambda,K}}(\lambda)}[\bar{H}_{\lambda_i}^{\mathrm{cri}}] \succ 0$ given that $\bar{H}_{\lambda_i}^{\mathrm{cri}} \succ 0$.

## L  PROOF OF THEOREM 3

We start with standard uniform deviation bound based on Rademacher complexity (Bartlett & Mendelson, 2002).

**Claim 3** ((Bartlett & Mendelson, 2002)). *Let the sample $\{z_1, \cdots, z_N\}$ be drawn i.i.d. from a distribution $P$ over $\mathcal{Z}$ and let $\mathcal{F}$ be a function class on $\mathcal{Z}$ mapping from $\mathcal{Z}$ to a bounded set. Then for $\delta > 0$, with probability at least $1 - \delta$, it holds that $\sup_{f\sim\mathcal{F}} ||\frac{1}{N}\sum_{i=1}^N f(z_i) - E_{z\sim P}[f(z)]|| \leq 2R(\mathcal{F}, z_1, \cdots, z_n) + \sqrt{\frac{\log(1/\delta)}{N}}$, where $R(\mathcal{F}, z_i, \cdots, z_N)$ is the Rademacher complexity of the function class $\mathcal{F}$.*

From Claim 3, we know that the generalization gap $|\mathcal{G}(\mathcal{F}_\gamma)| \leq R(\bar{\mathcal{F}}_{\bar\gamma}, \mathcal{T}_1^{\mathrm{tr}}, \cdots, \mathcal{T}_{N^{\mathrm{tr}}}^{\mathrm{tr}}) + \sqrt{\frac{\log(1/\delta)}{N^{\mathrm{tr}}}}$, where $\bar{F}_{\bar\gamma} \triangleq \{J_i(\pi_\theta) : \pi_\theta \in \mathcal{F}_{\bar\gamma}\}$. Therefore, we can compute the Rademacher complexity:

$$
R(\bar{\mathcal{F}}_{\bar\gamma}, \mathcal{T}_1^{\mathrm{tr}}, \cdots, \mathcal{T}_{N^{\mathrm{tr}}}^{\mathrm{tr}}) = E_{\sigma_i}\left[ \sup_{J\sim\bar{F}_{\bar\gamma}} \frac{1}{N^{\mathrm{tr}}} \sum_{i=1}^{N^{\mathrm{tr}}} \sigma_i J_i^{\mathrm{tr}}(\pi_i(\theta)) \right],
$$
$$
\leq \sup_{\pi_\theta\sim\mathcal{F}_{\bar\gamma}, i\sim P(\mathcal{T})} J_i(\pi_i(\theta)),
$$
$$
= \sup_{\pi_\theta\sim\mathcal{F}_{\bar\gamma}, i\sim P(\mathcal{T})} E_{(s,a)\sim\rho^{\pi_{\phi_i}}}^{\pi_{\phi_i}} [\log \pi_{\phi_i}(a|s) A_i^{\pi_{\phi_i}}(s,a)],
$$
$$
= \sup_{\pi_\theta\sim\mathcal{F}_{\bar\gamma}, i\sim P(\mathcal{T})} E_{(s,a)\sim\rho^{\pi_{\phi_i}}}^{\pi_{\phi_i}} [(\phi_i^\top f(s,a) - \log(\sum_{a'\in\mathcal{A}} e^{\phi_i^\top f(s,a)})) A_i^{\pi_{\phi_i}}(s,a)],
$$

where $\sigma_i$ is a random variable with equal probability of choose 1 and $-1$. Recall that $\phi_i = \theta - \alpha \nabla_\theta J_i(\pi_\theta)$ and $||\nabla_\theta J_i(\pi_\theta)||$ is bounded. Moreover, $A_i^{\pi_{\phi_i}}(s,a)$ is also bounded given that the reward value is bounded, and the chosen feature vector $f(s,a)$ is also bounded. Therefore, there exists a constant $C_1$ such that $R(\bar{\mathcal{F}}_{\bar\gamma}, \mathcal{T}_1^{\mathrm{tr}}, \cdots, \mathcal{T}_{N^{\mathrm{tr}}}^{\mathrm{tr}}) \leq \frac{C_1}{\sqrt{N^{\mathrm{tr}}}} \sup_{\pi_\theta\sim\mathcal{F}_{\bar\gamma}, i\sim P(\mathcal{T})} E_{(s,a)\sim\rho^{\pi_{\phi_i}}}^{\pi_{\phi_i}} [\theta^\top \bar{h}_i]$ where $\bar{h}_i^\top \bar{h}_i = E_{i\sim P(\mathcal{T})}[\bar{H}_i]$. Therefore, we have that $R(\bar{\mathcal{F}}_{\bar\gamma}, \mathcal{T}_1^{\mathrm{tr}}, \cdots, \mathcal{T}_{N^{\mathrm{tr}}}^{\mathrm{tr}}) \leq C_2\sqrt{\frac{\gamma}{N^{\mathrm{tr}}}}$ where $C_2$ is a positive constant. Therefore, we have that $|\mathcal{G}(\mathcal{F}_\gamma)| \leq 2C_2\sqrt{\frac{\gamma}{N^{\mathrm{tr}}}} + \sqrt{\frac{\log(1/\delta)}{N^{\mathrm{tr}}}} = O(\sqrt{\frac{\gamma}{N^{\mathrm{tr}}}} + \sqrt{\frac{\log(1/\delta)}{N^{\mathrm{tr}}}})$.

# M    EXPERIMENT DETAILS

To update the meta-parameter, we need to sample 20 trajectories for each task and we use data mixup augmentation to generate another 20 trajectories for the critical tasks. For a fair comparison, the baselines use 40 sampled trajectories for the critical tasks to update the meta-parameter.

## M.1    PRACTICAL CONSIDERATION OF ALGORITHM 1

In practice, we use warm start to accelerate Algorithm 1. At each iteration $k$, the upper-level update produces a new mixing-coefficient distribution $P_{\phi_{\lambda,k}}(\lambda)$. The lower level samples a set $\{\lambda_{i,k}\}_{i=1}^{N^{\mathrm{cri}}}$ of mixing coefficients to augment the critical tasks and computes the posterior $P^*(\theta|\{\bar{\mathcal{T}}_i^{\mathrm{cri}}(\lambda_{i,k})\})$ by solving the optimization problem $\phi^*(\{\lambda_{i,k}\}_{i=1}^{N^{\mathrm{cri}}}) = \arg\max_\phi E_{p_\phi(\theta)}[L(\theta, \{\bar{\mathcal{T}}_i^{\mathrm{cri}}(\lambda_{i,k})\}_{i=1}^{N^{\mathrm{cri}}}, \{\mathcal{T}_i^{\mathrm{tr}}\}_{i=1}^{N^{\mathrm{tr}}-N^{\mathrm{cri}}})]$ in (5) where $P_{\phi^*(\{\lambda_{i,k}\}_{i=1}^{N^{\mathrm{cri}}})}(\theta) = P^*(\theta|\{\bar{\mathcal{T}}_i^{\mathrm{cri}}(\lambda_{i,k})\})$. The most straightforward way to solve the optimization problem in (5) at each iteration $k$ is that, we randomly initialize $\phi$ and solve the optimization problem until convergence to get $\phi^*(\{\lambda_{i,k}\}_{i=1}^{N^{\mathrm{cri}}})$. However, it requires many gradient-ascent steps which can be computationally expensive. To address this issue, we use warm start. Specifically, instead of randomly initializing $\phi$, we initialize $\phi$ in iteration $k$ with the parameter $\phi(\{\lambda_{i,k-1}\}_{i=1}^{N^{\mathrm{cri}}})$ learned from the previous iteration $k-1$ and use one-step gradient ascent to obtain the new parameter $\phi(\{\lambda_{i,k}\}_{i=1}^{N^{\mathrm{cri}}})$. The warm start can significantly reduce the number of gradient-ascent steps because it provides a good initialization. Note that the mixing coefficient parameters in two consecutive iterations $k-1$ and $k$ are close because they are only different in one-step gradient $\phi_{\lambda,k} - \phi_{\lambda,k-1} = \beta g_{\lambda,k-1}$ where $\beta$ is a small learning rate and $g_{\lambda,k-1}$ is the gradient. Therefore, it is expected that their corresponding lower-level optimal parameters $\phi^*(\{\lambda_{i,k}\}_{i=1}^{N^{\mathrm{cri}}})$ and $\phi^*(\{\lambda_{i,k-1}\}_{i=1}^{N^{\mathrm{cri}}})$ are also close. Note that we only use one-step gradient ascent to update $\phi$ and thus $\phi(\{\lambda_{i,k-1}\}_{i=1}^{N^{\mathrm{cri}}})$ and $\phi^*(\{\lambda_{i,k-1}\}_{i=1}^{N^{\mathrm{cri}}})$ are different, however, $\phi(\{\lambda_{i,k-1}\}_{i=1}^{N^{\mathrm{cri}}})$ is updated towards $\phi^*(\{\lambda_{i,k-1}\}_{i=1}^{N^{\mathrm{cri}}})$ and $\phi(\{\lambda_{i,k}\}_{i=1}^{N^{\mathrm{cri}}})$ is initialized from $\phi(\{\lambda_{i,k-1}\}_{i=1}^{N^{\mathrm{cri}}})$ and updated towards $\phi^*(\{\lambda_{i,k}\}_{i=1}^{N^{\mathrm{cri}}})$. Intuitively, it is expected that $\phi(\{\lambda_{i,k}\}_{i=1}^{N^{\mathrm{cri}}})$ will approach $\phi^*(\{\lambda_{i,k}\}_{i=1}^{N^{\mathrm{cri}}})$ and finally become $\phi^*(\{\lambda_{i,k}\}_{i=1}^{N^{\mathrm{cri}}})$ when $k$ increases. In fact, it has been theoretically guaranteed that $\phi^*(\{\lambda_{i,k}\}_{i=1}^{N^{\mathrm{cri}}})$ will finally reach $\phi^*(\{\lambda_{i,k}\}_{i=1}^{N^{\mathrm{cri}}})$ when $k$ increases (Hong et al., 2023).

## M.2    FEASIBILITY OF THE AUGMENTED STATE

Recall that the augmented state $\bar{s} = \lambda_i s + (1 - \lambda_i)s'$ is a convex combination of the two states $s$ and $s'$. These two states $(s, s')$ are both valid states because they are visited by the policy. Therefore, their convex combination $\bar{s}$ will always be a feasible state in the state space if the state space is convex. In our experiments, the state spaces are all convex and thus the augmented state $\bar{s}$ is always a feasible state.

## M.3    HOW TO FIND POORLY ADAPTED TASKS

The poorly adapted tasks are neither the training tasks nor the testing tasks, but a third kind of tasks: validation tasks. Specifically, we have three kind of tasks: training tasks, validation tasks, and testing tasks. The original meta-policy $\pi_0$ is trained on the training tasks, and we find poorly adapted validation tasks. We then identify the critical training tasks to explain these poorly adapted tasks, augment the critical tasks, and retrain a meta-policy on the augmented critical tasks and the other training tasks. The generalization is evaluated on the testing tasks. The poorly adapted tasks are automatically selected, and we discuss how to select the poorly adapted tasks in Appendix M.4, M.5, and M.6. Take the MuJoCo experiment as an example, we randomly sample 50 validation tasks and compute the cumulative reward of the meta-policy $\pi_0$ after adaptation. We pick the top 20% tasks with the smallest cumulative reward as the poorly adapted tasks.

### M.4 DRONE NAVIGATION WITH OBSTACLES

We cannot run the meta-learning algorithm directly on the physical AR.Drone 2.0 because during training, the drone needs to interact with the environment and can be damaged due to collision with the obstacle and the wall. Therefore, we build a simulator in Gazebo (Figure 5) that mimics the physical environment with the scale of $1 : 1$. We run the meta-learning algorithm on the simulated drone in the simulator. By using the simulator, we can avoid damage of the physical drone. Once we obtain a learned policy that has good performance in the simulator, we implement the policy on the physical drone and count the successful rate.

**Discussion of the sim-to-real problem**. In some cases, the policy that has good performance in the simulator may not have good performance in the real world due to the discrepancy between the simulator and the real world. However, in our case, the sim-to-real issue is not significant because of two reasons: (i) the simulated drone is built according to the dynamics of a real Ar. Drone 2.0 (Huang & Sturm, 2014); (ii) the states and actions in our case are just the coordinates of the location and the heading direction of the drone instead of some low-level control such as the motor's velocity, rotation direction, etc. Given that Vicon can monitor precise pose of the physical drone and the simulator is built on the $1 : 1$ scale, if a learned trajectory can succeed in the simulator, it can succeed in the real world as long as that the low-level control of both the simulated and physical drones can strictly follow the actions.

In this experiment, the state of the drone is its 3-D coordinate $(x, y, z)$ and the action of the drone is also a 3-D coordinate $(dx, dy, dz)$ which captures the heading direction of the drone. We fix the length of each step as $0.1$ and thus the next state is $(x + \frac{dx}{10\sqrt{(dx)^2+(dy)^2+(dz)^2}}, y + \frac{dy}{10\sqrt{(dx)^2+(dy)^2+(dz)^2}}, z + \frac{dz}{10\sqrt{(dx)^2+(dy)^2+(dz)^2}})$. In this experiment, we do not need the drone to change its height so that we usually fix the value of $z$ and set $dz = 0$. The goal is an $1 \times 1$ square. Denote the coordinate of the center of the goal as $(x_{\text{goal}}, y_{\text{goal}})$, then for all the different tasks, $x_{\text{goal}} \in (0.5, 7.5)$ and $y_{\text{goal}} \in (8, 11)$. The obstacle is a $3 \times 1$ square. Denote the coordinate of the lower left corner of the obstacle as $(x_{\text{obstacle}}, y_{\text{obstacle}})$, then for different tasks, $x_{\text{obstacle}} \in (0, 4)$ and $y_{\text{obstacle}} \in (4, 5)$.

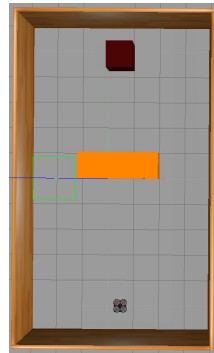

Figure 5: Simulator

### M.5 STOCK MARKET

AI for finance (Ke et al., 2025) has been receiving increasing attention. We use the real-world data of 30 constitute stocks in Dow Jones Industrial Average from 2021-01-01 to 2022-01-01. The 30 stocks are respectively: 'AXP', 'AMGN', 'AAPL', 'BA', 'CAT', 'CSCO', 'CVX', 'GS', 'HD', 'HON', 'IBM', 'INTC', 'JNJ', 'KO', 'JPM', 'MCD', 'MMM', 'MRK', 'MSFT', 'NKE', 'PG', 'TRV', 'UNH', 'CRM', 'VZ', 'V', 'WBA', 'WMT', 'DIS', 'DOW'.

The state of the stock market MDP is the perception of the stock market, including the open/close price of each stock, the current asset, and some technical indices (Liu et al., 2021). The action has the same dimension as the number of stocks where each dimension represents the amount of buying/selling the corresponding stock. The detailed formulation of the MDP can be found in FinRL (Liu et al., 2021). In addition to the applications to finance, AI has been applied to multiple domains, including graph (Ouyang et al., 2024), job shop scheduling (Wang et al., 2025), natural language (Lan et al., 2025; Kang et al., 2025; Guo et al., 2024), security Guo et al. (2025), and robotics (Qiao et al., 2023).

The RL agent trades stocks on every stock market opening day in order to maximize profit as well as avoid taking risks. The reward function is defined as $p_1 - p_2$ where $p_1$ is the profit which is the money earned from trading stocks subtracting the transaction cost, and $p_2$ models the preference of whether willing to take risks. In specific, $p_2$ is positive if the investor buys stocks whose turbulence indices are larger than a certain turbulence threshold, and zero otherwise. The value of $p_2$ depends on the type and amount of the trading stocks. The turbulence index measures the risk of buying a stock (Liu et al., 2021), and a lower turbulence threshold means that the RL agent is less willing to take risks. The turbulence thresholds for different RL tasks are different. The turbulence index is a technical index of stock market and is included as a dimension of the state (Liu et al., 2021). The turbulence

index measures the price fluctuation of a stock. If the turbulence index is high, the corresponding stock has a high fluctuating price and thus is risky to buy. Therefore, an investor unwilling to take risks has a relatively low turbulence threshold. The function $p_2$ is defined as the amount of buying the stocks whose turbulence index is larger than the turbulence threshold. Therefore, the more the target investor buys the stocks whose turbulence index is larger than the turbulence threshold, the larger $p_2$ will be and thus the smaller reward the target investor will receive. For different tasks, we randomly sample the turbulence threshold between 45 and 50.

We first sample 50 training tasks to learn a meta-policy. We then randomly sample 10 tasks and find the top 3 tasks where the meta-policy adapts with the worst performance. These 3 tasks are the poorly adapted tasks. We run our algorithm on the 50 training tasks. To evaluate the generalization performance, we randomly sample 100 test tasks.

## M.6 MuJoCo

The target velocity problem of MuJoCo is a standard problem for MRL (Finn et al., 2017; Fallah et al., 2021). We follow the standard setting in (Finn et al., 2017) to study the target velocity problem of two MuJoCo robots: HalfCheetah and Ant, where the episode length is 200. In the original HalfCheetah environment, the reward function is *forward_reward-ctrl_cost* where the forward reward is the velocity of the robot. In the target velocity task, we do not change *ctrl_cost* and we change *forward_reward* as $-|v - v_{\text{target}}|$ where $v$ is the current robot velocity and $v_{\text{target}}$ is the target velocity. In the original Ant environment, the reward function is *healthy_reward+forward_reward-ctrl_cost-contact_cost*, and we only change *forward_reward* as $-|v - v_{\text{target}}|$. Following (Finn et al., 2017), we choose the target velocity range of HalfCheetah as $[0, 2]$ and the target velocity range of Ant as $[0, 3]$. For both HalfCheetah and Ant scenarios, we sample 100 training tasks. We find 10 poorly adapted tasks and run our algorithm on the 100 training tasks to find critical tasks and improve generalization. To evaluate the generalization performance, we randomly sample 200 test tasks.

## M.7 Ablation study on the augmentation method

In this section, we include an ablation study to show the effectiveness of our augmentation method. In specific, the previous data mixup augmentation methods (Yao et al., 2021; Wang et al., 2020; Zhang et al., 2018) use a predefined Beta distribution $P(\lambda) = \text{Beta}(\alpha, \alpha)$ where $\alpha = 1$ to sample $\lambda_i$. In contrast, we propose to optimize the distribution $P(\lambda)$ by solving the problem (6). To show the effectiveness of our method, we compare to a method that uses the predefined distribution $P(\lambda) = \text{Beta}(1, 1)$ to augment the critical tasks, we refer to this method as "predefined augmentation". We also choose different number of critical tasks $N^{\text{cri}}$ to find the case where the predefined augmentation method performs the best. We find that the optimal number of critical tasks of the predefined augmentation method is same as the one of our method.

Table 3: Comparison of augmentation methods.

| Experiment | Drone | Stock market | HalfCheetah | Ant |
|---|---|---|---|---|
| Ours | $0.97 \pm 0.01$ | $421.13 \pm 12.11$ | $-44.67 \pm 4.35$ | $119.15 \pm 4.02$ |
| Predefined augmentation | $0.93 \pm 0.02$ | $394.16 \pm 16.85$ | $-58.53 \pm 4.82$ | $109.62 \pm 5.47$ |

Table 3 shows that our method significantly outperforms the method that uses the predefined distribution $P(\lambda) = \text{Beta}(1, 1)$ to augment the critical tasks.

## M.8 Evaluation of the explanation

This section evaluates the fidelity and usefulness of the explanation.

**Evaluation of fidelity**. Fidelity means the correctness of the explanation. Recall that the explanation (i.e., the critical tasks) aims to identify the most important training tasks to achieve high cumulative reward on the poorly adapted tasks. To evaluate the fidelity, we train a meta-policy on the critical tasks and evaluate the performance of the meta-policy on the poorly adapted tasks. We introduce

two baselines for comparison. The first baseline is the "original meta-policy" that trains on all the training tasks. We refer to this baseline as "original". The second baseline is that we randomly pick $N^{\text{cri}}$ training tasks and train a meta-policy over the $N^{\text{cri}}$ training tasks. We refer to this baseline as "random". Note that we chose $N^{\text{cri}}$ as the optimal number of critical tasks (shown in Figure 3), i.e., 10 for Drone, 15 for Stock Market, and 10 for HalfCheetah and Ant. We compare the performance on the poorly adapted tasks with these two baselines.

Table 4: Fidelity comparison.

| Experiment | Drone | Stock market | HalfCheetah | Ant |
|------------|-------|--------------|-------------|-----|
| Ours | $0.97 \pm 0.02$ | $442.29 \pm 12.79$ | $-37.14 \pm 5.15$ | $132.62 \pm 5.15$ |
| Original | $0.68 \pm 0.16$ | $296.27 \pm 35.16$ | $-104.79 \pm 12.72$ | $62.47 \pm 11.03$ |
| Random | $0.71 \pm 0.08$ | $284.97 \pm 29.85$ | $-96.78 \pm 9.24$ | $65.25 \pm 3.10$ |

Table 4 shows that our explanation has high fidelity because the meta-policy trained on our explanation significantly outperforms the two baselines on the poorly adapted tasks.

**Evaluation of usefulness**. Usefulness means whether the explanation can indeed help improve generalization. Table 1 already shows that our method (XMRL) can significantly improve MAML. However, this might be the effect of the task augmentation method. To evaluate whether the critical tasks help improve generalization. We randomly pick $N^{\text{cri}}$ training tasks and use the same algorithm (Algorithm 1) to augment these critical tasks. The choice of $N^{\text{cri}}$ is same as the one we use to evaluate the fidelity. We refer to this method as random, and we compare the generalization of our method with this random method.

Table 5: Usefulness comparison.

| Experiment | Drone | Stock market | HalfCheetah | Ant |
|------------|-------|--------------|-------------|-----|
| MAML | $0.87 \pm 0.01$ | $359.13 \pm 18.63$ | $-68.89 \pm 4.36$ | $100.64 \pm 3.63$ |
| Ours | $0.97 \pm 0.01$ | $421.13 \pm 12.11$ | $-44.67 \pm 4.35$ | $119.15 \pm 4.02$ |
| Random | $0.89 \pm 0.02$ | $365.16 \pm 11.07$ | $-71.12 \pm 5.09$ | $104.98 \pm 3.65$ |

Table 5 shows that our explanation has high usefulness because randomly pick $N^{\text{cri}}$ training tasks and augment can only slightly improve the generalization, while our method can significantly improve generalization.

## M.9 META-WORLD EXPERIMENT

In this section, we conduct an experiment on ML10 of Meta-World. We first validate our observation that "$\pi_0$ adapts well to some tasks but poorly to others". In particular, we first use MAML to train a meta-policy $\pi_0$ and evaluate $\pi_0$ on the test tasks of ML10. We report the results below:

Table 6: MAML generalization on ML10 test tasks (success rate).

| door close | drawer open | level pull | shelf place | sweep into | average |
|------------|-------------|------------|-------------|------------|---------|
| 0.86 | 0.35 | 0.26 | 0.00 | 0.00 | 0.29 |

Table 6 validates that the imbalanced generalization indeed exists where $\pi_0$ adapt well to the task of "door close" but adapts poorly to the tasks of "shelf place" and "sweep into". We next evaluate our method MAML+XMRL and the baselines (MAML+TW, MAML+MA, and MAML+MR) on ML10. In addition, we compare with two state-of-the-art MRL algorithms: SDVT (Lee et al., 2023) and ECET (Shala et al., 2025).

Table 7: Performance to ML10 test tasks (success rate).

|  | door close | drawer open | level pull | shelf place | sweep into | average |
|---|---|---|---|---|---|---|
| MAML | 0.86 | 0.35 | 0.26 | 0.00 | 0.00 | 0.29 |
| MAML+XMRL | 0.85 | 0.38 | 0.25 | 0.09 | 0.26 | 0.37 |
| MAML+TW | 0.72 | 0.29 | 0.28 | 0.00 | 0.24 | 0.31 |
| MAML+MA | 0.82 | 0.38 | 0.22 | 0.00 | 0.02 | 0.29 |
| MAML+MR | 0.87 | 0.33 | 0.26 | 0.04 | 0.00 | 0.30 |
| SDVT | 0.08 | 0.65 | 0.01 | 0.00 | 0.90 | 0.33 |
| ECET | 0.58 | 0.26 | 0.24 | 0.04 | 0.46 | 0.32 |

The results in Table 7 demonstrate that our method can significantly outperform MAML by more than $20\%$ (in terms of average success rate), and outperform the other baselines. Note that while MAML+XMRL outperforms MAML in terms of average success rate, its success rates on "door close" and "level pull" are slightly lower than MAML. This is reasonable because it is not expected that MAML+XMRL can outperform MAML on every task according to no free lunch theorem.

**Statistical significance test**. We provide p-values from paired t-tests below where we choose the significance value $\alpha = 0.05$.

Table 8: P-values from paired t-tests comparing each method to MAML on ML-10 ($\alpha = 0.05$).

|  | door close | drawer open | level pull | shelf place | sweep into | average |
|---|---|---|---|---|---|---|
| MAML+XMRL | 0.0209 | 0.0095 | 0.1192 | 0.0065 | 0.0000 | 0.0081 |
| MAML+TW | 0.0001 | 0.0039 | 0.1945 | 0.2589 | 0.0000 | 0.2551 |
| MAML+MA | 0.0041 | 0.0183 | 0.0340 | 0.4784 | 0.2850 | 0.5765 |
| MAML+MR | 0.0490 | 0.3747 | 0.0159 | 0.0010 | 0.4469 | 0.9293 |
| SDVT | 0.0000 | 0.0000 | 0.0000 | 0.7373 | 0.0000 | 0.1087 |
| ECET | 0.0000 | 0.0007 | 0.0021 | 0.0097 | 0.0000 | 0.0586 |

The results in Table 8 demonstrate that our method MAML+XMRL significantly outperforms MAML in terms of average success rate as its p-value is under the significance threshold. Among the other five baselines, only the p-value of ECET is close to but still above the significance threshold.

**Ablation study on the number of critical tasks** $N^{\mathbf{cri}}$. We vary the number of critical tasks and record the corresponding success rates of MAML+XMRL below. Similarly as in Appendix **??**, we report the performance against the percentage of critical tasks. Please refer to Appendix **??** for design details.

Table 9: Average success rates against percentages of critical tasks on ML10

| Percentage of critical tasks | 0% | 10% | 20% | 30% | 40% | 50% |
|---|---|---|---|---|---|---|
| Success rate | 0.29 | 0.33 | 0.37 | 0.32 | 0.28 | 0.20 |

The results in Table 9 demonstrate that our method achieves the highest performance when the critical task are 20% of the training tasks.

**Ablation study on the augmentation method**. To demonstrate the effectiveness of our learned augmentation method, we compare our method to a predefined augmentation distribution $P(\lambda) = \text{Beta}(1, 1)$. Please refer to Appendix M.7 for design details.

Table 10: Comparison of augmentation methods on ML10.

|  | door close | drawer open | level pull | shelf place | sweep into | average |
|---|---|---|---|---|---|---|
| Our method | 0.85 | 0.38 | 0.25 | 0.09 | 0.26 | 0.37 |
| Predefined augmentation | 0.83 | 0.37 | 0.24 | 0.00 | 0.09 | 0.31 |

The results in Table 10 validate the effectiveness of our learned augmentation as it outperforms a predefined augmentation distribution.

**Evaluation of the explanation**. We follow the setup in Appendix M.8 and evaluate the fidelity and usefulness of our method below. Please refer to Appendix M.8 for definitions of fidelity, usefulness, and the comparison baselines.

Table 11: Fidelity and usefulness comparison on ML10.

| Fidelity | | | Usefulness | | |
|---|---|---|---|---|---|
| Ours | Original | Random | MAML | Ours | Random |
| 0.24 | 0.00 | 0.02 | 0.29 | 0.37 | 0.31 |

The results in Table 11 demonstrate that our explanation has high fidelity and high usefulness.

# N    LIMITATIONS

Despite the benefits of our proposed algorithms, the limitation of our method is that we require to interact with the environment to collect augmented data, which makes it infeasible for offline RL cases. We will explore how to extend our method to the offline RL case in future works.

# O    THE USE OF LLMS

We use LLMs to help polish paragraphs in the introduction.

