# OpenReview forum: "Leveraging Explanation to Improve Generalization of Meta Reinforcement Learning"
_ICLR.cc/2026/Conference — ICLR 2026 Poster_

### Official Review · Reviewer_kbyg · 2025-10-27

**Soundness:** 2
**Presentation:** 2
**Contribution:** 2
**Rating:** 2
**Confidence:** 4

**Summary:**

This paper primarily aims to improve overall generalization performance by identifying critical tasks and further strengthening them through data augmentation.

Regarding the identification of critical tasks, the authors define the algorithm as finding the optimal weights of critical task pairs. I checked the proofs in the appendix, and it seems to be correct.

As for further improving generalization performance by augmenting critical tasks, the authors propose optimizing the previously predefined distribution P(λ) to maximally store information. This is also formulated as a bi-level optimization problem, for which the authors provide a tighter generalization bound.

Experiments are conducted both in MuJoCo simulations and on real robotic systems.

**Strengths:**

1. I appreciate the authors can state the empirical observation on performance degradation, which I find is significant to make the paper complete.
2. I also appreciate the authors could hold an ablation study to compare with the policy obtained through (2) to further validate the role of task augmentation.
3. The experiments are conducted even on real-robot, which is very impressive for me.

**Weaknesses:**

1. After reviewing the theoretical derivation, I feel it might be problematic.

In Appendix E DERIVATION OF THE CONDITIONAL MUTUAL INFORMATION: line 819, where is the distribution P(\overline{\mathcal{T}}). Line 827, why the integral concerning P(\overline{\mathcal{T}}) is removed.

In Appendix F line 859, why extending the distribution of P(\overline{\mathcal{T}}) can be transformed to P(\lambda_i). In line 870, I do think the fact that P(\overline{\mathcal{T}_{i=1:N}} | \mathcal{T}_{i=1:N})  = P(\lambda_i) is wrong. The left-hand side is a joint distribution over N augmented tasks (given the original tasks), while the right-hand side is merely P(\lambda_i), which is not only ambiguous in terms of the index I, but also represents only a single-variable distribution, not a joint distribution over N variables. Furthermore, even if we only consider one-task condition, I do think P(\overline{\mathcal{T_i}} | \mathcal{T_i}) not equals to P(\lambda_i). Mathematically, it should be \int P(\overline{\mathcal{T_i}} |  \mathcal{T_i}, \lambda_i) P(\lambda_i) d\lambda_i

I feel that these issues have seriously compromised the theoretical rigor of the paper, so I did not proceed to check the subsequent theoretical content further.

2. The authors propose to augment the state by convex combination, but I do think in many scenarios, the convex combination of state would not be acceptable. Also, I think this is a strong assumption for applying the algorithm. Therefore, the authors at least should make this clarified.

3. In meta RL, while most online methods emerged around 2021-2022, several offline methods also aim to improve generalization ability, including those also based on information theory [1]. I believe discussing these works properly would be beneficial for providing a more complete literature review.

4. Could you please give me an intuition about why only minor degradation on very few tasks happens when focusing more on critical tasks? Maybe this is a combinatorial phenomena stems from  the number of critical tasks and the optimized distribution P(\lambda) ?

5. Solving the proposed algorithm needs to optimize two bi-level problems iteratively. It seems that there may exist some instability. Do the authors use some tricks to stabilize the training process?

[1] Towards an information theoretic framework of context-based offline meta-reinforcement learning

**Questions:**

See weakness above

---

> ### Author Response · Authors · 2025-11-19
> **Response to Reviewer kbyg**
>
> Thank you for your insightful and constructive reviews. We believe that our discussions can lead to a stronger paper. We have uploaded a revised version with revisions highlighted in blue. Because the line numbers in the revised version differ from those in the original, we will use "(original)" and "(revised)" to indicate whether a referenced line number corresponds to the original or the revised version. We address your comments in a point-by-point manner. For weakness 1, we first address the comment on Appendix F and then the comment on Appendix E.
>
> **Weakness 1(a)**: In Appendix F line 859(original), why extending the distribution of $P(\bar{\mathcal{T}}\_i^{\text{cri}})$ can be transformed to $P(\lambda\_i)$. In line 870(original), I do think the fact that $P(\\{\bar{\mathcal{T}}\_i^{\text{cri}}(\lambda\_i)\\}\_{i=1}^{N^{\text{cri}}} | \\{\mathcal{T}\_i^{\text{cri}}\\}\_{i=1}^{N^{\text{cri}}}) = P(\lambda\_i)$ is wrong. The left-hand side is a joint distribution over N augmented tasks (given the original tasks), while the right-hand side is merely $P(\lambda\_i)$, which is not only ambiguous in terms of the index I, but also represents only a single-variable distribution, not a joint distribution over N variables. Furthermore, even if we only consider one-task condition, I do think $P(\bar{\mathcal{T}}\_i^{\text{cri}}(\lambda\_i)|\mathcal{T}\_i^{\text{cri}})$ not equals to $P(\lambda\_i)$. Mathematically, it should be $\int P(\bar{\mathcal{T}}\_i^{\text{cri}} |\mathcal{T}\_i^{\text{cri}},\lambda\_i) P(\lambda\_i) d\lambda\_i$.
> **Answer**: Thank you for mentioning the derivation in Appendix F. We really appreciate your reading our proof and we are very happy to discuss it with you. We would like to clarify the derivation in Appendix F to address the confusion.
> For simplicity, we first consider discrete probabilities and one task, e.g., the first critical task $\mathcal{T}\_1^{\text{cri}}$. We use $\Lambda\_1$ to denote a random variable that follows the distribution $P(\lambda)$ and use $\lambda\_1$ to denote a specific value sampled from $P(\lambda)$, e.g., $\lambda\_1=0.5$. Therefore, the augmented task $\bar{\mathcal{T}}\_1^{\text{cri}}(\Lambda\_1)$ is also a random variable because the augmented task $\bar{\mathcal{T}}\_1^{\text{cri}}(\Lambda\_1)$ is parameterized by the variable $\Lambda\_1$ and $\Lambda\_1$ is a random variable. Once the value of $\Lambda\_1$ is set to a specific value, the augmented task $\bar{\mathcal{T}}\_1^{\text{cri}}(\Lambda\_1)$ **deterministically** becomes a specific task. Therefore, $\bar{\mathcal{T}}\_1^{\text{cri}}(\Lambda\_1=\lambda\_1=0.5)$ is a specific task because $\lambda\_1$ is a specific value. Note that the critical task $\mathcal{T}\_1^{\text{cri}}$ is a training task that is already given, therefore, the probability of the (random) augmented task $\bar{\mathcal{T}}\_1^{\text{cri}}(\Lambda\_1)$ being the specific task $\bar{\mathcal{T}}\_1^{\text{cri}}(\Lambda\_1=\lambda\_1=0.5)$ is the probability of the specific value $\lambda\_1=0.5$ being sampled from $P(\lambda)$. Therefore, $P(\bar{\mathcal{T}}\_1^{\text{cri}}(\Lambda\_1=\lambda\_1=0.5)|\mathcal{T}\_1^{\text{cri}})=P(\Lambda\_1=\lambda\_1=0.5)$.
> We have noted that you want to use marginalization to get the probability: $P(\bar{\mathcal{T}}\_1^{\text{cri}}(\Lambda\_1=\lambda\_1=0.5)|\mathcal{T}\_1^{\text{cri}})=\sum\_{\lambda\_1'} P(\bar{\mathcal{T}}\_1^{\text{cri}}(\Lambda\_1=\lambda\_1=0.5)|\mathcal{T}\_1^{\text{cri}},\lambda\_1')P(\Lambda\_1=\lambda\_1')$. In fact, this marginalization is equivalent to our derivation.
>
>
> \begin{align*}
>   &P(\bar{\mathcal{T}}\_1^{\text{cri}}(\Lambda\_1=\lambda\_1=0.5)|\mathcal{T}\_1^{\text{cri}})\\\\
>   &=\sum\_{\lambda\_1'} P(\bar{\mathcal{T}}\_1^{\text{cri}}(\Lambda\_1=\lambda\_1=0.5)|\mathcal{T}\_1^{\text{cri}},\Lambda\_1=\lambda\_1')P(\Lambda\_1=\lambda\_1'|\mathcal{T}\_1^{\text{cri}}) \\\\
>   &\overset{(i)}{=}\sum\_{\lambda\_1'} P(\bar{\mathcal{T}}\_1^{\text{cri}}(\Lambda\_1=\lambda\_1=0.5)|\mathcal{T}\_1^{\text{cri}},\Lambda\_1=\lambda\_1')P(\Lambda\_1=\lambda\_1')  \\\\
>   &\overset{(ii)}{=} \sum\_{\lambda\_1'} \mathbf{1} \\{\lambda\_1'=0.5\\}P(\Lambda\_1=\lambda\_1') \\\\
>   & = P(\Lambda\_1=\lambda\_1'=0.5),
> \end{align*}
> where $(i)$ follows the fact that $P(\Lambda\_1=\lambda\_1')=P(\Lambda\_1=\lambda\_1'|\mathcal{T}\_1^{\text{cri}})$ because $\Lambda\_1$ is independent of $\mathcal{T}\_1^{\text{cri}}$, $(ii)$ follows the fact that $P(\bar{\mathcal{T}}\_1^{\text{cri}}(\Lambda\_1=\lambda\_1=0.5)|\mathcal{T}\_1^{\text{cri}},\Lambda\_1=\lambda\_1')=1$ if $\lambda\_1'=0.5$ and $P(\bar{\mathcal{T}}\_1^{\text{cri}}(\Lambda\_1=\lambda\_1=0.5)|\mathcal{T}\_1^{\text{cri}},\Lambda\_1=\lambda\_1')=0$ if $\lambda\_1'\neq 0.5$, and $\mathbf{1}\\{\cdot\\}$ is the indicator function.

---

> ### Author Response · Authors · 2025-11-19
>
> In the proof, we use the notation $P(\mathcal{T}\_i^{\text{cri}}(\lambda\_i))$ to represent $P(\mathcal{T}\_i^{\text{cri}}(\Lambda\_i=\lambda\_i))$ for a simple notation, however, we find that this might be a cause of confusion. Therefore, we have revised this misleading notation in the revised version.
> In line 870(original), $P(\\{\bar{\mathcal{T}}\_i^{\text{cri}}(\lambda\_i)\\}\_{i=1}^{N^{\text{cri}}} | \\{\mathcal{T}\_i^{\text{cri}}\\}\_{i=1}^{N^{\text{cri}}}) = P(\lambda\_i)$ is a typo, the correct version should be $P(\\{\bar{\mathcal{T}}\_i^{\text{cri}}(\Lambda\_i=\lambda\_i)\\}\_{i=1}^{N^{\text{cri}}} | \\{\mathcal{T}\_i^{\text{cri}}\\}\_{i=1}^{N^{\text{cri}}}) = P(\\{\Lambda\_i=\lambda\_i\\}\_{i=1}^{N^{\text{cri}}})$, which we have corrected in the revised version.
> If we consider a continuous version, this relation still holds when we consider probability densities: $p\_{\bar{\mathcal{T}}\_1^{\text{cri}}|\mathcal{T}\_1^{\text{cri}}}(\bar{\mathcal{T}}\_1^{\text{cri}}(\Lambda\_1=\lambda\_1=0.5)|\mathcal{T}\_1^{\text{cri}})=p\_{\Lambda\_1}(\Lambda\_1=\lambda\_1=0.5)$.
>
> $$
> p\_{\bar{\mathcal{T}}\_1^{\text{cri}}|\mathcal{T}\_1^{\text{cri}}}
> (\bar{\mathcal{T}}\_1^{\text{cri}}(\Lambda\_1=\lambda\_1=0.5) | \mathcal{T}\_1^{\text{cri}}) =
> \int p\_{\bar{\mathcal{T}}\_1^{\text{cri}},\Lambda\_1|\mathcal{T}\_1^{\text{cri}}}
> (\bar{\mathcal{T}}\_1^{\text{cri}}(\Lambda\_1=\lambda\_1=0.5),\Lambda\_1=\lambda\_1' | \mathcal{T}\_1^{\text{cri}})
>  d\lambda\_1'.
> $$
>
> Because the augmented task $\bar{\mathcal{T}}\_1^{\text{cri}}(\Lambda\_1)$ is a deterministic
> function of $(\mathcal{T}\_1^{\text{cri}},\Lambda\_1)$, the conditional joint density
> factorizes as
>
> $$
> p\_{\bar{\mathcal{T}}\_1^{\text{cri}},\Lambda\_1|\mathcal{T}\_1^{\text{cri}}}(\bar{\mathcal{T}}\_1^{\text{cri}}(\Lambda\_1=\lambda\_1=0.5),\Lambda\_1=\lambda\_1' | \mathcal{T}\_1^{\text{cri}}) =
> p\_{\Lambda\_1|\mathcal{T}\_1^{\text{cri}}}(\lambda\_1' | \mathcal{T}\_1^{\text{cri}})
> \delta(\bar{\mathcal{T}}\_1^{\text{cri}}(\Lambda\_1=\lambda\_1=0.5)-\bar{\mathcal{T}}\_1^{\text{cri}}(\Lambda\_1=\lambda\_1')),
> $$
>
> where $\delta(\cdot)$ is the Dirac delta. Substituting this expression into the marginalization, we obtain
>
> $$
> p\_{\bar{\mathcal{T}}\_1^{\text{cri}}|\mathcal{T}\_1^{\text{cri}}}(\bar{\mathcal{T}}\_1^{\text{cri}}(\Lambda\_1=\lambda\_1=0.5) | \mathcal{T}\_1^{\text{cri}}) =\int p\_{\Lambda\_1|\mathcal{T}\_1^{\text{cri}}}(\lambda\_1' | \mathcal{T}\_1^{\text{cri}})\delta(
> \bar{\mathcal{T}}\_1^{\text{cri}}(\Lambda\_1=\lambda\_1=0.5)-\bar{\mathcal{T}}\_1^{\text{cri}}(\Lambda\_1=\lambda\_1'))d\lambda\_1'.
> $$
>
> Since $\bar{\mathcal{T}}\_1^{\text{cri}}(\Lambda\_1=\lambda\_1=0.5)=\bar{\mathcal{T}}\_1^{\text{cri}}(\Lambda\_1=\lambda\_1')$ when $\lambda\_1'=0.5$, then the Dirac delta collapses the integral to
>
> \begin{align*}
> p\_{\bar{\mathcal{T}}\_1^{\text{cri}}|\mathcal{T}\_1^{\text{cri}}}
> (\bar{\mathcal{T}}\_1^{\text{cri}}(\Lambda\_1=\lambda\_1=0.5) |\mathcal{T}\_1^{\text{cri}})=p\_{\Lambda\_1|\mathcal{T}\_1^{\text{cri}}}(\lambda\_1'=0.5 \mid \mathcal{T}\_1^{\text{cri}}).
> \end{align*}
>
> Finally, since $\Lambda\_1$ is independent of $\mathcal{T}\_1^{\text{cri}}$, we have
> $$p\_{\Lambda\_1|\mathcal{T}\_1^{\text{cri}}}(\lambda\_1'=0.5 \mid \mathcal{T}\_1^{\text{cri}})=p\_{\Lambda\_1}(\lambda\_1=0.5),
> $$
>
> and therefore
>
> \begin{align*}
> p\_{\bar{\mathcal{T}}\_1^{\text{cri}}|\mathcal{T}\_1^{\text{cri}}}(\bar{\mathcal{T}}\_1^{\text{cri}}(\Lambda\_1=\lambda\_1=0.5) | \mathcal{T}\_1^{\text{cri}})=p\_{\Lambda\_1}(\Lambda\_1=\lambda\_1=0.5).
> \end{align*}
>
> For simplicity, in the following answers, we only include scenarios for discrete probabilities.

---

> > ### Author Response · Authors · 2025-11-19
> >
> > **Weakness 1(b)**: In Appendix E DERIVATION OF THE CONDITIONAL MUTUAL INFORMATION: line 819(original), where is the distribution $P(\overline{\mathcal{T}})$. Line 827(original), why the integral concerning $P(\overline{\mathcal{T}})$ is removed.
> > **Answer**: Thank you for mentioning the derivation in Appendix E. The derivation omits some intermediate steps, and that is why it looks confusing. Here we would like to include all the intermediate steps to avoid confusion. We also include the full derivation in Appendix E in the revised version. The fundamental reason that $P(\\{\bar{\mathcal{T}}\_i^{\text{cri}}(\Lambda\_i=\lambda\_i)\\}\_{i=1}^{N^{\text{cri}}})$ can be removed is that $P(\\{\bar{\mathcal{T}}\_i^{\text{cri}}(\Lambda\_i=\lambda\_i)\\}\_{i=1}^{N^{\text{cri}}})$ is decomposed by $P(\\{\mathcal{T}\_i^{\text{cri}}\\}\_{i=1}^{N^{\text{cri}}})P(\\{\Lambda\_i=\lambda\_i\\}\_{i=1}^{N^{\text{cri}}})$.
> >
> > \begin{align*}
> >     &I(\theta;\\{\bar{\mathcal{T}}\_i^{\text{cri}}(\Lambda\_i\sim P(\lambda))\\}\_{i=1}^{N^{\text{cri}}}|\\{\mathcal{T}\_i^{\text{cri}}\\}\_{i=1}^{N^{\text{cri}}}),\\\\
> >     & \overset{(a)}{=} \int P(\theta,\\{\bar{\mathcal{T}}\_i^{\text{cri}}(\Lambda\_i=\lambda\_i)\\}\_{i=1}^{N^{\text{cri}}},\\{\mathcal{T}\_i^{\text{cri}}\\}\_{i=1}^{N^{\text{cri}}})\cdot\\\\
> >     &\log\frac{P(\theta,\\{\bar{\mathcal{T}}\_i^{\text{cri}}(\Lambda\_i=\lambda\_i)\\}\_{i=1}^{N^{\text{cri}}}|\\{\mathcal{T}\_i^{\text{cri}}\\}\_{i=1}^{N^{\text{cri}}})}{P(\theta|\\{\mathcal{T}\_i^{\text{cri}}\\}\_{i=1}^{N^{\text{cri}}})P(\\{\bar{\mathcal{T}}\_i^{\text{cri}}(\Lambda\_i=\lambda\_i)\\}\_{i=1}^{N^{\text{cri}}}|\\{\mathcal{T}\_i^{\text{cri}}\\}\_{i=1}^{N^{\text{cri}}})}(d\theta)(d\\{\bar{\mathcal{T}}\_i^{\text{cri}}(\Lambda\_i=\lambda\_i)\\}\_{i=1}^{N^{\text{cri}}})(d\\{\mathcal{T}\_i^{\text{cri}}\\}\_{i=1}^{N^{\text{cri}}}),\\\\
> >     &\overset{(b)}{=}  \int P(\theta|\\{\bar{\mathcal{T}}\_i^{\text{cri}}(\Lambda\_i=\lambda\_i)\\}\_{i=1}^{N^{\text{cri}}},\\{\mathcal{T}\_i^{\text{cri}}\\}\_{i=1}^{N^{\text{cri}}})\cdot P(\\{\bar{\mathcal{T}}\_i^{\text{cri}}(\Lambda\_i=\lambda\_i)\\}\_{i=1}^{N^{\text{cri}}}|\\{\mathcal{T}\_i^{\text{cri}}\\}\_{i=1}^{N^{\text{cri}}})\cdot P(\\{\mathcal{T}\_i^{\text{cri}}\\}\_{i=1}^{N^{\text{cri}}})\cdot\\\\
> >     &\log\frac{P(\theta,\\{\bar{\mathcal{T}}\_i^{\text{cri}}(\Lambda\_i=\lambda\_i)\\}\_{i=1}^{N^{\text{cri}}}|\\{\mathcal{T}\_i^{\text{cri}}\\}\_{i=1}^{N^{\text{cri}}})}{P(\theta|\\{\mathcal{T}\_i^{\text{cri}}\\}\_{i=1}^{N^{\text{cri}}})P(\\{\bar{\mathcal{T}}\_i^{\text{cri}}(\Lambda\_i=\lambda\_i)\\}\_{i=1}^{N^{\text{cri}}}|\\{\mathcal{T}\_i^{\text{cri}}\\}\_{i=1}^{N^{\text{cri}}})}(d\theta)(d\\{\bar{\mathcal{T}}\_i^{\text{cri}}(\Lambda\_i=\lambda\_i)\\}\_{i=1}^{N^{\text{cri}}})(d\\{\mathcal{T}\_i^{\text{cri}}\\}\_{i=1}^{N^{\text{cri}}}),\\\\
> >     &\overset{(c)}{=}\int P(\theta|\\{\bar{\mathcal{T}}\_i^{\text{cri}}(\Lambda\_i=\lambda\_i)\\}\_{i=1}^{N^{\text{cri}}},\\{\mathcal{T}\_i^{\text{cri}}\\}\_{i=1}^{N^{\text{cri}}})\cdot P(\\{\Lambda\_i=\lambda\_i\\}\_{i=1}^{N^{\text{cri}}})\cdot P(\\{\mathcal{T}\_i^{\text{cri}}\\}\_{i=1}^{N^{\text{cri}}})\cdot\\\\
> >     &\log\frac{P(\theta,\\{\bar{\mathcal{T}}\_i^{\text{cri}}(\Lambda\_i=\lambda\_i)\\}\_{i=1}^{N^{\text{cri}}}|\\{\mathcal{T}\_i^{\text{cri}}\\}\_{i=1}^{N^{\text{cri}}})}{P(\theta|\\{\mathcal{T}\_i^{\text{cri}}\\}\_{i=1}^{N^{\text{cri}}})P(\\{\bar{\mathcal{T}}\_i^{\text{cri}}(\Lambda\_i=\lambda\_i)\\}\_{i=1}^{N^{\text{cri}}}|\\{\mathcal{T}\_i^{\text{cri}}\\}\_{i=1}^{N^{\text{cri}}})}(d\theta)(d\\{\bar{\mathcal{T}}\_i^{\text{cri}}(\Lambda\_i=\lambda\_i)\\}\_{i=1}^{N^{\text{cri}}})(d\\{\mathcal{T}\_i^{\text{cri}}\\}\_{i=1}^{N^{\text{cri}}}),\\\\
> >     &\overset{(d)}{=}\int P(\theta|\\{\bar{\mathcal{T}}\_i^{\text{cri}}(\Lambda\_i=\lambda\_i)\\}\_{i=1}^{N^{\text{cri}}},\\{\mathcal{T}\_i^{\text{cri}}\\}\_{i=1}^{N^{\text{cri}}})\cdot P(\\{\Lambda\_i=\lambda\_i\\}\_{i=1}^{N^{\text{cri}}})\cdot P(\\{\mathcal{T}\_i^{\text{cri}}\\}\_{i=1}^{N^{\text{cri}}})\cdot\\\\
> >     &\log\frac{P(\theta,\\{\bar{\mathcal{T}}\_i^{\text{cri}}(\Lambda\_i=\lambda\_i)\\}\_{i=1}^{N^{\text{cri}}}|\\{\mathcal{T}\_i^{\text{cri}}\\}\_{i=1}^{N^{\text{cri}}})}{P(\theta|\\{\mathcal{T}\_i^{\text{cri}}\\}\_{i=1}^{N^{\text{cri}}})P(\\{\bar{\mathcal{T}}\_i^{\text{cri}}(\Lambda\_i=\lambda\_i)\\}\_{i=1}^{N^{\text{cri}}}|\\{\mathcal{T}\_i^{\text{cri}}\\}\_{i=1}^{N^{\text{cri}}})}(d\theta)(d\\{\Lambda\_i=\lambda\_i\\}\_{i=1}^{N^{\text{cri}}})(d\\{\mathcal{T}\_i^{\text{cri}}\\}\_{i=1}^{N^{\text{cri}}}),\\\\
> > \end{align*}

---

> > > ### Author Response · Authors · 2025-11-19
> > >
> > > \begin{align*}
> > >     & = \int P(\theta|\\{\bar{\mathcal{T}}\_i^{\text{cri}}(\Lambda\_i=\lambda\_i)\\}\_{i=1}^{N^{\text{cri}}},\\{\mathcal{T}\_i^{\text{cri}}\\}\_{i=1}^{N^{\text{cri}}})P(\\{\Lambda\_i=\lambda\_i\\}\_{i=1}^{N^{\text{cri}}})P(\\{\mathcal{T}\_i^{\text{cri}}\\}\_{i=1}^{N^{\text{cri}}})\cdot\\\\
> > >     &\log\frac{P(\theta|\\{\bar{\mathcal{T}}\_i^{\text{cri}}(\Lambda\_i=\lambda\_i)\\}\_{i=1}^{N^{\text{cri}}},\\{\mathcal{T}\_i^{\text{cri}}\\}\_{i=1}^{N^{\text{cri}}})}{P(\theta|\\{\mathcal{T}\_i^{\text{cri}}\\}\_{i=1}^{N^{\text{cri}}})}(d\theta)(d\\{\Lambda\_i=\lambda\_i\\}\_{i=1}^{N^{\text{cri}}})(d\\{\mathcal{T}\_i^{\text{cri}}\\}\_{i=1}^{N^{\text{cri}}}),\\\\
> > >     &\overset{(e)}{=}\int P(\theta|\\{\bar{\mathcal{T}}\_i^{\text{cri}}(\Lambda\_i=\lambda\_i)\\}\_{i=1}^{N^{\text{cri}}})P(\\{\Lambda\_i=\lambda\_i\\}\_{i=1}^{N^{\text{cri}}})P(\\{\mathcal{T}\_i^{\text{cri}}\\}\_{i=1}^{N^{\text{cri}}})\cdot\\\\
> > >     &\log\frac{P(\theta|\\{\bar{\mathcal{T}}\_i^{\text{cri}}(\Lambda\_i=\lambda\_i)\\}\_{i=1}^{N^{\text{cri}}})}{P(\theta|\\{\mathcal{T}\_i^{\text{cri}}\\}\_{i=1}^{N^{\text{cri}}})}(d\theta)(d\\{\Lambda\_i=\lambda\_i\\}\_{i=1}^{N^{\text{cri}}})(d\\{\mathcal{T}\_i^{\text{cri}}\\}\_{i=1}^{N^{\text{cri}}}),\\\\
> > >     &=E\_{\Lambda\_i\in[0,1],\Lambda\_i\sim P
> > >     (\lambda),\theta\sim P(\cdot|\\{\bar{\mathcal{T}}\_i^{\text{cri}}(\Lambda\_i)\\}\_{i=1}^{N^{\text{cri}}})}\Big[\log\frac{P(\theta|\\{\bar{\mathcal{T}}\_i^{\text{cri}}(\Lambda\_i)\\}\_{i=1}^{N^{\text{cri}}})}{P(\theta|\\{\mathcal{T}\_i^{\text{cri}}\\}\_{i=1}^{N^{\text{cri}}})}\Big],
> > > \end{align*}
> > > where $(a)$ follows the definition of conditional mutual information, $(b)$ follows the standard chain rule of probability $P(A,B,C)=P(A|B,C)P(B|C)P(C)$, $(c)$ follows the fact that $P(\\{\bar{\mathcal{T}}\_i^{\text{cri}}(\Lambda\_i=\lambda\_i)\\}\_{i=1}^{N^{\text{cri}}}|\\{\mathcal{T}\_i^{\text{cri}}\\}\_{i=1}^{N^{\text{cri}}})=P(\\{\Lambda\_i=\lambda\_i\\}\_{i=1}^{N^{\text{cri}}})$ (already explained), $(d)$ follows the fact that $\int P(\\{\bar{\mathcal{T}\_i}(\Lambda\_i=\lambda\_i)\\}\_{i=1}^{N^{\text{cri}}})d(\\{\bar{\mathcal{T}\_i}(\Lambda\_i=\lambda\_i)\\}\_{i=1}^{N^{\text{cri}}})=\int P(\\{\bar{\mathcal{T}\_i}(\Lambda\_i=\lambda\_i)\\}\_{i=1}^{N^{\text{cri}}})d(\\{\Lambda\_i=\lambda\_i\\}\_{i=1}^{N^{\text{cri}}})d(\\{\mathcal{T}\_i\\}\_{i=1}^{N^{\text{cri}}})$ because $\bar{\mathcal{T}}\_i^{\text{cri}}(\Lambda\_i=\lambda\_i)$ is determinisitically determined by $\mathcal{T}\_i^{\text{cri}}$ and $\lambda\_i$, and $(e)$ follows the fact that $P(\theta|\\{\bar{\mathcal{T}}\_i^{\text{cri}}(\Lambda\_i=\lambda\_i)\\}\_{i=1}^{N^{\text{cri}}},\\{\mathcal{T}\_i^{\text{cri}}\\}\_{i=1}^{N^{\text{cri}}})=P(\theta|\\{\bar{\mathcal{T}}\_i^{\text{cri}}(\Lambda\_i=\lambda\_i)\\}\_{i=1}^{N^{\text{cri}}})$ because the meta-parameter is trained on the augmented critical tasks $\\{\bar{\mathcal{T}}\_i^{\text{cri}}(\Lambda\_i=\lambda\_i)\\}\_{i=1}^{N^{\text{cri}}}$.
> > >
> > > **Weakness 2**: The authors propose to augment the state by convex combination, but I do think in many scenarios, the convex combination of state would not be acceptable. Also, I think this is a strong assumption for applying the algorithm. Therefore, the authors at least should make this clarified.
> > > **Answer**: Thank you for mentioning the convex combination. We agree that this convex combination assumption is fundamental to our algorithm and we have clarified in line 182(original) (or line 186(revised)) and Appendix M.1(original) (or Appendix N.2(revised)) that the states augmented from the convex combination are always feasible when state space is convex, and the experiments in this paper all satisfy this condition. In fact, this convex combination operation is standard in data mixup for RL [D1]. Following your suggestion, we have explicitly mentioned that this convex combination assumption is fundamental to our approach in line 185(revised).
> > >
> > > **Weakness 3**: In meta RL, while most online methods emerged around 2021-2022, several offline methods also aim to improve generalization ability, including those also based on information theory [1]. I believe discussing these works properly would be beneficial for providing a more complete literature review.
> > > **Answer**: Thank you for providing this valuable reference. We have included discussions on this reference and other offline meta-RL works in lines 749-761(revised).

---

> > > > ### Author Response · Authors · 2025-11-19
> > > >
> > > > **Weakness 4**: Could you please give me an intuition about why only minor degradation on very few tasks happens when focusing more on critical tasks? Maybe this is a combinatorial phenomena stems from the number of critical tasks and the optimized distribution P($\lambda$)?
> > > > **Answer**: It is very insightful of you to note the minor degradation on very few tasks. The fundamental reason for this phenomenon is that our approach does not significantly bias the meta-policy. We would like to elaborate on this phenomenon from two complementary aspects: the number of critical tasks and the way the meta-policy is optimized.
> > > > First, as shown in the ablation study in Appendix M.8(original) (or lines 418-428(revised)), the best performance occurs when the number of critical tasks is only a small portion (approximately 10–30\%) of all training tasks. This means that augmentation is applied to only a limited subset of training tasks, and thus its influence is naturally constrained. Since these critical tasks constitute a small subset of the entire training set, they do not dominate the learning signal. In practice, we vary the number of critical tasks and choose the best performance.
> > > > Second, in the lower-level problem (5) the meta-policy is learned by optimizing the average return of all the training tasks (including augmented critical tasks and unaugmented non-critical tasks). We agree that if the critical tasks are assigned with higher weights, it is easy for the meta-policy to be biased and overfit on the critical tasks. However, in problem (5), the augmented critical tasks still have the same weights as the unaugmented non-critical tasks. This design is very important to avoid bias because it prevents the meta-policy from overfitting to or over-emphasizing the critical tasks. While task augmentation enriches the critical tasks by embedding more information into them, it does not increase their relative importance in the meta-objective.
> > > >
> > > > **Weakness 5**: Solving the proposed algorithm needs to optimize two bi-level problems iteratively. It seems that there may exist some instability. Do the authors use some tricks to stabilize the training process?
> > > > **Answer**: The fundamental idea to stabilize the training is to restrict the deviation of the learned meta-policy from the original meta-policy $\pi\_0$. Recall that the meta-policy $\pi\_0$ is learned by solving the vanilla meta-learning problem (1) and then we find critical tasks corresponding to $\pi\_0$ and improve $\pi\_0$. We use two designs for this purpose.
> > > > First, we use warm start initialization: every optimization run begins from the original meta-policy $\pi\_0$ instead of random initialization. Since $\pi\_0$ is already a reasonably good meta-policy, initializing from it provides two benefits: (1) it significantly reduces the number of iterations required for convergence, (2) it also serves as a soft regularization, ensuring that the meta-policy explores around $\pi\_0$. In contrast, the learned meta-policy can drift toward unstable regions of the parameter space if initialized randomly.
> > > > Second, we adopt a TRPO-style policy update, which explicitly constrains the KL divergence between the current meta-policy and $\pi\_0$. This ensures that each update step remains within a trust region, preventing abrupt policy shifts. This KL-control mechanism is particularly effective in stabilizing the meta-objective optimization, where abrupt movements in the meta-parameter distribution could otherwise propagate instability.
> > > >
> > > > [D1] Kaixin Wang, Bingyi Kang, Jie Shao, and Jiashi Feng. "Improving generalization in reinforcement learning with mixture regularization", NeurIPS, 2020.

---

> ### Comment · Reviewer_kbyg · 2025-11-20
> **Response by Reviewer kbyg**
>
> I appreciate the authors' detailed response.
>
> Concerning 1: I will check the details ASAP.
> Concerning 2: I still have some concern about why the state space can always be assumed to be convex?
> Concerning 5: How do you adjust the hyper-parameters in TRPO-based update. As you know, TRPO is very sensitive to the hyper-parameters.

---

> > ### Author Response · Authors · 2025-11-20
> >
> > Thank you for your feedback. We address your concerns in a point-by-point manner.
> >
> > **Answer to Concern 2**: We do not assume that the state space is always convex. Our discussion in Appendix M.1(original) (or Appendix N.2(revised)) clarifies a conditional statement: the augmented states are always feasible **if** the underlying state space is convex. In our four experimental environments (Drone, MuJoCo, Stock Market, and Meta-World), the state spaces are all convex, so the augmented states are all feasible. However, if the state space is not convex, some augmented states may not be feasible. In conclusion, we do not assume that the state space is always convex. In order to evaluate our algorithms, we choose benchmarks with convex spaces so that feasibility is preserved.
> >
> > **Answer to Concern 5**: We agree that TRPO is sensitive to its hyper-parameters. The key hyper-parameter we tune is the KL-divergence trust-region radius $\delta$, which limits how far the updated meta-policy is allowed to deviate from the original meta-policy $\pi_0$. Intuitively, a smaller $\delta$ stabilizes training because it constrains the meta-policy within a small neighborhood of $\pi_0$, however, it may also restrict exploration to a better solution. On the other hand, a larger $\delta$ provides more room for exploration but can be risky because the policy can deviate too much from $\pi_0$ and explore some unstable regions. In practice, we start from $\delta=0.01$ which is a standard and default value in Stable-Baselines3 and we gradually increase $\delta$ until the performance does not improve or the training becomes unstable. This procedure provides a principled balance between stability and policy improvement. Empirically, the values for our experiments are: 0.05(Drone), 0.06(HalfCheetah), 0.04(Ant), 0.02(Stock Market), 0.03(Meta-World).

---

> > > ### Comment · Reviewer_kbyg · 2025-11-21
> > > **Response by reviewer kbyg**
> > >
> > > Thanks for the author's response.
> > >
> > > Concerning 2: According to the statement of what I said in weakness "The authors propose to augment the state by convex combination, but I do think in many scenarios, the convex combination of state would not be acceptable. Also, I think this is a strong assumption for applying the algorithm.". I feel concerned that the method can not be applied into many scenarios as there is an assumption about the convex coverage of the state space. Could the authors provide a detailed explanation of the scenarios in which a non-convex state-space may arise, and evaluate the performance degradation of the algorithm under such conditions?
> > >
> > > Concerning 5: Thanks for the clarification

---

> > > > ### Author Response · Authors · 2025-11-22
> > > >
> > > > Thank you for mentioning applying our algorithm to scenarios where the state space is non-convex. In the following context, we discuss (1) when the state space can be non-convex; (2) how to apply our algorithm when the state space is non-convex; (3) empirical evaluation of our algorithm on a non-convex state space.
> > > > **When the state space can be non-convex**. Non-convex state space appears when the environment has forbidden regions or hard constraints. For example, the state space in the drone navigation task is a rectangle room. If there is a forbidden region like a hole in the room, the state space becomes non-convex because it is now a rectangle with a hole inside. Another example is that in robotic manipulation tasks, if the angle of a joint cannot be within -30 degree to 30 degree, the state space also becomes non-convex because there is an interval gap within allowable angular space.
> > > > **Apply our algorithm to non-convex state space**. Given two states $s$ and $s'$, the corresponding augmented state is $\bar{s}=\lambda s+(1-\lambda)s'$. When the state space is non-convex, we can still apply our algorithm by discarding invalid augmented states and only keeping tuples $(\bar{s},\bar{a},\bar{r},\bar{s}\_{\text{next}})$ where $\bar{s}$ is a valid state in the state space. Since each policy update collects a large number of $(s,a,r,s\_{\text{next}})$ tuples and randomly samples many pairs of tuples to generate augmented tuples, a substantial number of feasible augmented tuples typically remain even after invalid states are removed. In the worst (and practically unlikely) case where all augmented states are infeasible, the augmentation step contributes no additional tuples and our algorithm reduces to standard meta-RL without harming performance of standard meta-RL.
> > > > **Empirical evaluation**. Since we need to evaluate how non-convex state space degrades our algorithm, we would like to modify the environments of the current experiments (Drone, MuJoCo, Stock Market, and Meta-World) so that the state space becomes non-convex. By doing so, we can directly compare the algorithm performance under the non-convex state space to the one under the original convex state space in the paper. Since MuJoCo, Stock Market, and Meta-World limit the users' access to fully modify the environments, we modify the original Drone environment. The drone experiment is a physical experiment that involves real drones. For the time being, we only conduct the drone experiment for non-convex state space in simulation.
> > > > The original drone environment is a $7\times 12$ rectangle room with a convex state space. For each task, the goal is a $1\times1$ square and its center's position $(x\_{\text{goal}},y\_{\text{goal}})$ varies within $x\_{\text{goal}}\in(0.5,7.5)$ and $y\_{\text{goal}}\in(8,11)$. The obstacle is a $3\times1$ rectangle whose lower left corner $(x\_{\text{obstacle}},y\_{\text{obstacle}})$ varies within $x\_{\text{obstacle}}\in(0,4)$ and $y\_{\text{obstacle}}\in(4,5)$. To make the state space non-convex, we remove a rectangle forbidden region $\mathcal{C}=\\{(x,y)|x\in[2,5],y\in[6,7]\\}$, which creates a “hole” in the middle of the rectangle room. This makes the state space become non-convex. We use the same training tasks and test tasks from the paper and report the results below:
> > > >
> > > > |                                | MAML          | MAML+XMRL     |
> > > > |--------------------------------|---------------|---------------|
> > > > | Drone (convex state space)     | 0.87 ± 0.01   | 0.97 ± 0.01   |
> > > > | Drone (non-convex state space) | 0.85 ± 0.02   | 0.93 ± 0.01   |
> > > >
> > > > The results demonstrate that (1) Our method still improves vanilla meta-RL when the state space becomes non-convex as many feasible augmented tuples continue to exist. (2) Both MAML and MAML+XMRL experience mild degradation, which is expected because navigating in a non-convex environment is inherently more difficult. (3) Compared to the case where the state space is convex, the improvement of our algorithm over MAML is slightly reduced. Specifically, the improvement is $0.93-0.85=0.08$ for non-convex state space and $0.97-0.87=0.1$ for convex state space. In conclusion, our method can improve meta-RL in both convex and non-convex state spaces, and the improvement is larger for convex state spaces because in this case, all the augmented tuples are valid.

---

> > > > > ### Comment · Reviewer_kbyg · 2025-11-23
> > > > > **Response by reviewer kbyg**
> > > > >
> > > > > Thanks for the authors' detailed clarification. My question concerning the convex-state space is generally resolved. I think adding this would significantly improve the completeness of this paper.
> > > > >
> > > > > I will adjust my score after checking the mathematical details.

---

> > > > > > ### Author Response · Authors · 2025-11-23
> > > > > >
> > > > > > Thank you for confirming that we have addressed the concern. We have included the discussion on convex state space in Appendix N.2 in the revised version.

---

> > > > > > > ### Comment · Reviewer_kbyg · 2025-11-24
> > > > > > > **Response by reviewer kbyg**
> > > > > > >
> > > > > > > After checking the mathematical details of the rebuttal part, I decide to adjust my score from 2 -> 6. I believe the paper holds some interesting information.
> > > > > > >
> > > > > > > Nevertheless, due to the time limit and the so-called "confusion" existing in the original submission, I can not carefully check the mathematical details in the remaining part now. I will lower my confidence. I strongly suggest the authors could invest more time into polishing the derivation details and mitigating the potential confusions.

---

> > > > > > > > ### Author Response · Authors · 2025-11-24
> > > > > > > >
> > > > > > > > We sincerely appreciate your time and effort in reviewing our paper. Thank you for recognizing the contributions of our work and for finding the paper interesting.
> > > > > > > >
> > > > > > > > Following your suggestion, we have included additional explanation for the proof to avoid confusion and included the discussion on the convex state space. We will proofread the rest of the proof to avoid any confusion.
> > > > > > > >
> > > > > > > > Thank you again for acknowledging our efforts in addressing your concerns.

---

### Official Review · Reviewer_HExQ · 2025-10-30

**Soundness:** 3
**Presentation:** 3
**Contribution:** 3
**Rating:** 6
**Confidence:** 3

**Summary:**

This paper addresses the problem of imbalanced generalization in Meta-Reinforcement Learning (MRL), where the meta-policy θ exhibits performance disparities when adapting to new tasks. To tackle this issue, the paper proposes a post-hoc improvement method. First, it identifies "critical tasks" that are most beneficial for improving performance on "poorly-performing tasks" through a bilevel optimization problem. Subsequently, the paper formulates another bilevel optimization framework to learn the optimal data augmentation for these critical tasks. The upper-level objective maximizes the Conditional Mutual Information (CMI) to ensure that the augmentation provides maximum additional information. The lower-level objective updates the meta-policy distribution based on the current augmentation strategy. Through this process, the algorithm iteratively optimizes the data augmentation strategy (specifically, the sampling distribution for the mixup coefficient λ). The authors theoretically prove the convergence and generalization improvement of their algorithm and validate its superior performance through experiments on MuJoCo, Meta-World, and real-world tasks.

**Strengths:**

The research direction of this paper holds significant industrial value and practical relevance. In industrial applications, there is a strong emphasis on synchronous and balanced convergence across various tasks, with a particular focus on the performance on long-tail or difficult tasks. Identifying critical training tasks is indeed key to enhancing performance on downstream poor-performing tasks from my point of view.

The method of identifying critical tasks by finding an optimal weight vector w is clever and intuitive, with a well-formulated optimization objective that aligns with cognitive reasoning.

The introduction of the CMI concept to construct the optimization objective for learning the sampling distribution of the mixup coefficient λ is reasonable and promising. Furthermore, the use of an augmented dataset {\hat{T}_{cri}} to estimate the posterior P(θ|{T_{cri}}) via expectation (which can be viewed as an application of the total probability formula) is an approximation method.

The mathematical derivations in the paper are robust, with a solid formulation of the problem and a well-established convergence proof.

**Weaknesses:**

As mentioned in the paper, this data augmentation method requires online MDP tuples from interaction with the environment, which seems to be unavailable in offline RL settings for now.

The overall method appears complex. It first requires finding the weights w and then proceeds to optimize the mixup sampling distribution parameters for λ, leading to high computational complexity.

The approach of using poorly-performing new tasks to supplement the training data (via data augmentation) might be viewed as "hacking" the test set. It does not seem to enhance the real generalization capability of MRL from a more universal perspective.

**Questions:**

see weakness

---

> ### Author Response · Authors · 2025-11-19
> **Response to Reviewer HExQ**
>
> Thank you for your constructive reviews. We appreciate that you find that our work holds significant industrial value and practical relevance, and we believe that our discussion can lead to a better paper. We have uploaded a revised version. Whenever we refer to an appendix, we will explicitly indicate whether the appendix index corresponds to the original version or the revised version. We address your comments in a point-to-point manner.
>
> **Weakness 1**: As mentioned in the paper, this data augmentation method requires online MDP tuples from interaction with the environment, which seems to be unavailable in offline RL settings for now.
> **Answer**: Thank you for raising this point. We agree that our current method requires online interaction to obtain the reward of the augmented tuple $(\bar{s},\bar{a})$, and extending the approach to offline RL is an important future direction. We discuss this in Appendix N(original) (or Appendix O(revised)), and here we outline how our method can be adapted to the purely offline setting.
> In offline RL setting, we can no longer obtain the augmented reward $\bar{r}$ from the environment and thus we have to estimate this augmented reward. A straightforward way [C1] is to use convex combinations of existing samples. Given two samples $(s,a,r)$ and $(s',a',r')$, [C1] constructs the augmented state as $\bar{s}=\lambda s +(1-\lambda)s'$, the augmented action as $\bar{a}=a$ if $\lambda\geq 0.5$ and $\bar{a}=a'$ otherwise. The augmented reward is constructed as a convex combination: $\bar{r}=\lambda r+(1-\lambda)r'$. This procedure does not require online interaction and therefore works in offline RL.
> However, this convex-combination reward estimate implicitly assumes that the reward varies linearly along the interpolation path between two states, which may not hold in many RL tasks. To address this issue, our planned extension is to learn a reward model from the offline dataset and use it to evaluate augmented tuples. Suppose that the offline dataset $\\{(s\_i,a\_i,r\_i,s\_i')\\}\_{i=1}^N$ includes $N$ tuples and we can learn a reward function $r\_\theta$ to fit this dataset by solving the regression problem $\min\_\theta \frac{1}{N}\sum\_{i=1}^N ||r\_\theta(s\_i,a\_i)-r\_i||^2$. After generating the augmented state $\bar{s}$, we use the current policy $\pi$ to select an action $\bar{a}$ and obtain the augmented reward via $\bar{r}=r\_\theta(\bar{s},\bar{a})$. This reward-model–based augmentation avoids assuming linearity of the reward function and is compatible with standard offline RL pipelines.
>
> **Weakness 2**: The overall method appears complex. It first requires finding the weights $\omega$ and then proceeds to optimize the mixup sampling distribution parameters for $\lambda$, leading to high computational complexity.
> **Answer**: Thank you for mentioning the computational complexity. Although our method conceptually involves two steps, i.e., solving the weighting problem to identify critical tasks and then optimizing the mixup parameter distribution, we intentionally designed both components to be computationally efficient.
> To address the complexity of solving the bi-level optimization problem (2) to find the weight $\omega$, we adopt a **single-loop** algorithm rather than the classic **double-loop** algorithm. A double-loop algorithm has two nested loops: inner loop and outer loop. At each outer iteration, the inner loop first solves the lower-level problem by multi-step gradient ascent to obtain the weighted meta-policy and then the weighted meta-policy is used to update the weight vector in the outer loop. Double-loop algorithms are usually computationally expensive because at each outer iteration, it requires to completely solve the lower-level optimization problem in the inner loop. In contrast, single-loop algorithms only have one loop and at each iteration, we only partially solve both the upper-level and lower-level problems via one-step gradient ascent. The single-loop algorithm is more efficient as it does not require an inner loop to fully solve the lower-level problem and it has been demonstrated that the single-loop algorithm can reach the same and even better results than double-loop algorithms [C2].
> In our method, we adopt this efficient single-loop strategy to solve both the weighting problem and mixup parameter optimization problem. We report detailed runtime comparisons in Appendix M.6(original) (or Appendix N.7(revised)), where our method demonstrates comparable computation time to the baselines.

---

> > ### Author Response · Authors · 2025-11-19
> >
> > **Weakness 3**: The approach of using poorly-performing new tasks to supplement the training data (via data augmentation) might be viewed as "hacking" the test set. It does not seem to enhance the real generalization capability of MRL from a more universal perspective.
> > **Answer**: Thank you for raising this important concern. We would like to clarify that our method does not use any test tasks when identifying poorly adapted tasks or performing data augmentation. In our setting, tasks are strictly divided into three disjoint sets: training tasks, validation tasks, and test tasks.
> > The original meta-policy $\pi\_0$ is trained only on the training tasks. To diagnose its weaknesses, we adapt $\pi\_0$ to each validation task and identify the validation tasks on which the adapted policy performs worst; these are the “poorly adapted” tasks. Importantly, these poorly adapted tasks are not part of the test set.
> > We then use these validation tasks solely to identify which training tasks are most critical to improve performance on the poorly adapted tasks. Afterwards, we augment only the critical training tasks and retrain a meta-policy on the entire training set. The final evaluation is conducted exclusively on the test tasks, which remain completely unseen during both the explanation and augmentation processes.
> > Because the poorly adapted tasks do not serve as training tasks and no information from test tasks is used when selecting or augmenting tasks, our approach does not constitute test-set leakage or “hacking.”
> >
> >
> > [C1] Kaixin Wang, Bingyi Kang, Jie Shao, and Jiashi Feng. "Improving generalization in reinforcement learning with mixture regularization", NeurIPS, 2020.
> >
> > [C2] Mingyi Hong et al., "A two-timescale stochastic algorithm framework for bilevel optimization: Complexity analysis and application to actor-critic", SIAM Journal on Optimization, 2023.

---

> > > ### Comment · Reviewer_HExQ · 2025-11-27
> > > **Response**
> > >
> > > Thanks for your response.  I will keep my score.

---

### Official Review · Reviewer_4uvj · 2025-10-31

**Soundness:** 3
**Presentation:** 2
**Contribution:** 2
**Rating:** 6
**Confidence:** 4

**Summary:**

This paper addresses generalization in meta-reinforcement learning (meta-RL) by introducing a novel two-stage approach that leverages example-based explanation and information-theoretic task augmentation.Stage 1 identifies critical training tasks most relevant to poorly adapted tasks through a bilevel optimization that assigns task-specific importance weights. Stage 2 improves generalization by maximizing conditional mutual information (CMI) between the meta-policy and augmented critical tasks. The authors use a learnable task mixup augmentation strategy to achieve this, rather than fixed data-mixing rules. Theoretical analysis guarantees O(1/\sqrt{K}) convergence and improved generalization bounds. Experiments on real-world (drone, stock trading) and simulation benchmarks (MuJoCo, Meta-World) show consistent gains over MAML and recent meta-RL improvement baselines (task weighting, meta-augmentation, meta-regularization).

**Strengths:**

1. New conceptual link between explainability and meta-RL: the example-based explanation mechanism to identify “critical” tasks is creative and intuitively appealing.
2. Information-theoretic formulation: Using conditional mutual information to formalize “attention” toward critical tasks provides a principled grounding and unifies ideas from explanation, data augmentation, and meta-learning.
3. Theoretical guarantees: The paper presents convergence and generalization proofs, which, while incremental, give some rigor to the framework.
4. Empirical validation: Comprehensive experiments across four environments show clear improvements over baselines, with meaningful ablations (number of critical tasks, learned vs. fixed mixup distribution).

**Weaknesses:**

1. Meta-RL setup realism:
The method inherits the common limitation of meta-RL—assuming access to an oracle task distribution P(T) from which both training and test tasks are sampled. In real applications, such a distribution rarely exists, and constructing it requires a near-perfect world model. Therefore, while the algorithm is technically solid, its practical significance is limited.

2. Limited novelty in theory: 1) Theorem 1 (convergence) essentially restates standard SGD-style results for bilevel optimization [1]; 2)Theorem 3 (generalization bound) reduces to the observation that increasing the number of training tasks improves generalization, which is unsurprising and  closely parallels Theorem 1 in (Yao et al., 2021) [2].

3. Conceptual gap between optimization levels: In Eq. (2), the upper-level objective optimizes the weights \omega to maximize returns on poorly adapted tasks, whereas the final meta-RL evaluation concerns adapted tasks. The link between these two objectives could be clarified, as there may exist a performance gap.

4. Scalability concerns: The approach requires solving multiple nested bilevel problems (for both task weighting and CMI maximization). When N^{tr} is large, this may cause optimization difficulties or high computational cost.

5. Mutual information interpretation: While MI provides a neat information-theoretic lens, it is not fully clear how optimizing MI translates into concrete improvements beyond encouraging task diversity. The relation between the learned MI term and mixup augmentation could be elaborated further.



[1] Jun Shu et al., Meta-Weight-Net, NeurIPS 2019.

[2] Improving Generalization in Meta-learning via Task Augmentation, ICML 2021.

**Questions:**

1. In Eq. (2), when N^{tr} becomes large, does the bilevel optimization become unstable or computationally infeasible? How is this mitigated in practice?

---

> ### Author Response · Authors · 2025-11-19
> **Response to Reviewer 4uvj**
>
> Thank you for your constructive reviews. We appreciate that you find our work creative, and we believe that our discussion can lead to a better paper. We have uploaded a revised version with revisions highlighted in blue. Because the line numbers in the revised version differ from those in the original, we will use "(original)" and "(revised)" to indicate whether a referenced line number corresponds to the original or the revised version. We address your comments in a point-to-point manner.
>
> **Weakness 1**: Meta-RL setup realism: The method inherits the common limitation of meta-RL—assuming access to an oracle task distribution P(T) from which both training and test tasks are sampled. In real applications, such a distribution rarely exists, and constructing it requires a near-perfect world model. Therefore, while the algorithm is technically solid, its practical significance is limited.
> **Answer**: Thank you for mentioning the common limitation of meta-RL. Our work indeed follows the standard meta-RL setting, and therefore inherits the widely acknowledged assumption that tasks are sampled from some underlying task distribution $P(\mathcal{T})$.
> However, we would like to emphasize that in practice $P(\mathcal{T})$ does not need to be an explicit or analytically specified distribution, nor does our method require constructing a near-perfect world model. Instead, meta-RL methods (including ours) only require a collection of training tasks, which implicitly corresponds to samples from an underlying (and possibly unknown) task distribution.
>
> **Weakness 2**: Limited novelty in theory: 1) Theorem 1 (convergence) essentially restates standard SGD-style results for bilevel optimization [1]; 2)Theorem 3 (generalization bound) reduces to the observation that increasing the number of training tasks improves generalization, which is unsurprising and closely parallels Theorem 1 in (Yao et al., 2021) [2]
> **Answer**: Thank you for mentioning the novelty in theory and providing the two valuable references [B1,B2]. We would like to clarify our theoretical distinctions from these two references.
> **Distinctions in convergence guarantee**. Although our result is SGD-style, the setting differs significantly from [B1], leading to a nontrivial analysis. There are three key distinctions from [B1]. First, our lower-level optimization problem is more challenging than that of [B1] and this leads to a more complicated hyper-gradient (i.e., the gradient of the upper-level problem). In [B1], the lower level is one-step gradient descent, so the hyper-gradient can be computed using a single chain rule step.
> In contrast, our lower-level problem requires fully solving a distributional optimization to obtain the optimal distribution parameter. As a result, the hyper-gradient cannot be expressed in closed form, and we must rely on the implicit function theorem to derive a tractable approximation. Second, we require significantly weaker assumptions. Theorem 1 in [B1] requires the upper-level objective to be Lipschitz continuous, and assumes the lower-level objective is twice differentiable and has bounded Hessian.
> In contrast, our convergence guarantee avoids assumptions directly on the objectives. We instead rely only on standard Lipschitz and smoothness assumptions of the parameterized policy, which is strictly weaker and more aligned with standard RL theory. This requires us to explicitly prove Lipschitz continuity and smoothness properties in Appendix I(original) (or Appendix J(revised)), rather than assuming them as in [B1]. Third, our upper-level optimization problem is more challenging. The upper-level optimization in [B1] is a conventional single-vector optimization. In our case, the upper level optimizes a distribution over mixing coefficients, which significantly complicates the analysis. We must both parameterize this distribution and prove gradient boundedness with respect to the distribution parameters. Note that these conditions are directly assumed in [B1] but need to be established in our framework.

---

> > ### Author Response · Authors · 2025-11-19
> >
> > **Distinctions in generalization improvement**. The essence of the generalization improvement is not to increase the number of training tasks but that task augmentation imposes an implicit regularization. Although our generalization proof follows the high-level idea of [B2] (prove the implicit regularization), the technical details and settings differ substantially. There are two key distinctions from [B2]. First, [B2] uses a Beta distribution with conjugacy but our setting does not. In [B2], the distribution over mixing coefficients is a Beta distribution, and the proof leverages its conjugacy properties to obtain closed-form simplifications.
> > In our setting, the Beta distribution is difficult to optimize, so we use a Gaussian parameterization. Without conjugacy, we must instead work with a second-order approximation of the meta-objective to derive the implicit regularization effect. Second, [B2] is supervised learning but our setting is RL that requires substantial reformulation. [B2] analyzes losses of the form $l(\theta,(x,y))=h(f_\theta(x))-yf_\theta(x)$ where $x$ is the data, $y$ is the label, $f_\theta$ is the prediction function, and $h$ is some function. However, this form does not apply in RL. In contrast, our RL meta-objective involves a stochastic policy and environment transitions, and does not naturally admit such a decomposition. To transfer the idea of implicit regularization to RL, we must reformulate the RL objective into a softmax-based surrogate form and derive regularization on this surrogate.
> >
> > **Weakness 3**: Conceptual gap between optimization levels: In Eq. (2), the upper-level objective optimizes the weights $\omega$ to maximize returns on poorly adapted tasks, whereas the final meta-RL evaluation concerns adapted tasks. The link between these two objectives could be clarified, as there may exist a performance gap.
> > **Answer**: Thank you for pointing out the potential objective gap between problem (2) and the final meta-RL evaluation. We would like to clarify that the meta-policy obtained from problem (2) is not used for updating the original meta-policy $\pi_0$. The purpose of problem (2) is solely to identify critical tasks, not to produce a new meta-policy for training. We fully agree that the meta-policy obtained by solving (2) could be biased toward the poorly adapted tasks and therefore unsuitable for final evaluation.
> > This issue is precisely why our method does not reweight tasks or train the meta-policy using the weights $\omega$ (discussed in lines 161–166(original) or lines 162-168(revised)). Instead, once the critical tasks are identified, we augment these critical tasks and then train the meta-policy over all training tasks, i.e., both the augmented critical tasks and the unaugmented non-critical tasks. In this way, the final meta-policy is learned from the entire task distribution, preventing bias toward poorly adapted tasks.
> > Because task augmentation enriches the information contained in the critical tasks without changing the training distribution, the final meta-policy remains unbiased and achieves strong performance on the final meta-RL evaluation.

---

> > > ### Author Response · Authors · 2025-11-19
> > >
> > > **Weakness 4**: Scalability concerns: The approach requires solving multiple nested bilevel problems (for both task weighting and CMI maximization). When $N^{tr}$ is large, this may cause optimization difficulties or high computational cost.
> > > **Answer**: Thank you for mentioning the computational cost for bilevel optimization when $N^{\text{cri}}$ is large.
> > > To reduce computational cost, we include how to efficiently solve the bi-level optimization problems in Appendix N.1 in the revised version. Here, we would like to explain in detail. To start with, we would like to first introduce two major classes of algorithms to solve bi-level optimization problems: double-loop algorithm and single-loop algorithm. Take solving the task weighting problem (2) as an example, double-loop algorithms have two nested loops: inner loop and outer loop. At each outer iteration, the inner loop first solves the lower-level problem by multi-step gradient ascent to obtain the weighted meta-policy and then the weighted meta-policy is used to update the weight vector in the outer loop. We agree that double-loop algorithms are usually computationally expensive when $N^{\text{tr}}$ is large because at each outer iteration, it requires to completely solve the lower-level optimization problem in the inner loop. To address this issue, single-loop algorithms are proposed where there is only one loop and at each iteration, we start from the parameters in the previous iteration and only partially solve both the upper-level and lower-level problems via one-step gradient ascent. The single-loop algorithm is more efficient as it does not require an inner loop to fully solve the lower-level problem and it has been demonstrated that the single-loop algorithm can reach the same and even better results than double-loop algorithms [B3]. Note that the computational cost of single-loop algorithms is dominated by computing per-task gradients which scales linearly with $N^{\text{tr}}$. In our work, we use single-loop algorithms where we only partially solve the lower-level problem via one-step gradient ascent for efficiency, and the computational cost scales linearly with $N^{\text{tr}}$.
> > >
> > > **Weakness 5**: Mutual information interpretation: While MI provides a neat information-theoretic lens, it is not fully clear how optimizing MI translates into concrete improvements beyond encouraging task diversity. The relation between the learned MI term and mixup augmentation could be elaborated further.
> > > **Answer**: It is very insightful of you to question the concrete improvements beyond encouraging task diversity and the relation between the MI term and mixup augmentation. These two aspects are closely connected. While mixup augmentation always increases data diversity by generating additional augmented data, not all mixup augmentations are equally informative. Different mixing coefficients $\lambda$ produce different augmented state–action distributions and therefore introduce different amounts of additional task information.
> > > Our key idea is to use mutual information to quantitatively measure how much additional information an augmentation injects into the meta-policy beyond what is already contained in the original critical tasks. In other words, MI allows us to formally evaluate how much a particular augmentation increases the task-relevant information stored in the parameter $\theta$. We then **learn** the mixup distribution $P(\lambda)$ so that the augmentation maximally increases this stored information. In summary, (1) Mixup alone always introduces diversity, but the amount of useful information depends heavily on the choice of $\lambda$; (2) Our MI objective explicitly selects augmentations that yield maximally informative diversity, i.e., those that most increase the task information stored in the meta-policy; (3) This results in concrete generalization improvements that go beyond generic task augmentation.

---

> ### Author Response · Authors · 2025-11-19
>
> **Question 1**: In Eq. (2), when $N^{\text{tr}}$ becomes large, does the bilevel optimization become unstable or computationally infeasible? How is this mitigated in practice?
> **Answer**: Please refer to our answer to Weakness 4. In the answer to Weakness 4, we elaborate how to reduce computational cost. Here we would like to elaborate how to stabilize training in practice. The fundamental idea to stabilize the training is to restrict the deviation of the learned meta-policy from the original meta-policy $\pi_0$. Recall that the meta-policy $\pi_0$ is learned by solving the vanilla meta-learning problem (1) and then we find critical tasks corresponding to $\pi_0$ and improve $\pi_0$. We use two designs for this purpose.
> First, we use warm start initialization: during optimization, the initial meta-policy is the original meta-policy $\pi_0$ instead of random initialization. Since $\pi_0$ is already a reasonably good meta-policy, initializing from it provides two benefits: (1) it significantly reduces the number of iterations required for convergence, (2) it also serves as a soft regularization, ensuring that the meta-policy explores around $\pi_0$. In contrast, the learned meta-policy can drift toward unstable regions of the parameter space if initialized randomly.
> Second, we adopt a TRPO-style policy update, which explicitly constrains the KL divergence between the current meta-policy and $\pi_0$. This ensures that each update step remains within a trust region, preventing abrupt policy shifts. This KL-control mechanism is particularly effective in stabilizing the meta-objective optimization, where abrupt movements in the meta-parameter distribution could otherwise propagate instability.
>
> [B1] Jun Shu et al., "Meta-Weight-Net: Learning an Explicit Mapping for Sample Weighting", NeurIPS 2019.
>
> [B2] Huaxiu Yao et al., "Improving Generalization in Meta-learning via Task Augmentation", ICML, 2021.
>
> [B3] Mingyi Hong et al., "A two-timescale stochastic algorithm framework for bilevel optimization: Complexity analysis and application to actor-critic", SIAM Journal on Optimization, 2023.

---

> > ### Comment · Reviewer_4uvj · 2025-11-26
> >
> > Thank you for the detailed explanation, which addresses my concerns. I will be keeping my score.

---

### Official Review · Reviewer_u1Lj · 2025-11-03

**Soundness:** 3
**Presentation:** 2
**Contribution:** 3
**Rating:** 4
**Confidence:** 3

**Summary:**

This paper presents a method mining, reweighting and augmenting training tasks in meta-RL in order to improve on difficult validation tasks.  Poor-performance tasks are first mined from a large evaluation pool.  Rather than re-incorporate these into the training set directly, this method then weights original training tasks based on their ability to improve performance on the mined set.  Finally, the MRL is fine-tuned jointly with augmentation mixing parameters $\lambda$ on augmented versions of the critical tasks.

The augmentation here is a mixup-like linear combination of states drawn from the existing task setup, so that the new augmented states can fill out the convex hull of the original state space.  The state mixing coefficients are optimized to increase mutual information of the distribution over the meta-learned inits theta_0 between augmented and non-augmented tasks, modeled as Gaussians --- that is, it optimizes the mixing distributions to try to get resultant thetas (post-RL-inner-loop) that are different from what what they would have been training on just the original tasks, making the augmentation actually provide different datapoints for the outer MRL step.

Theoretical convergence results indicate soundness of the approach.  The method is evaluated on multiple MRL benchmarks and settings, showing substantial improvement in poorly performing tasks with little regression in those that already perform well.

**Strengths:**

The approach of optimizing augmentation parameters to explicitly result in different theta points (not just different mixing states) is quite interesting, and evaluations compared to predefined mixing demonstrate its effectiveness over a reasonable uniform baseline.  The task mining step is also formulated well and makes a lot of sense.

**Weaknesses:**

While there are some good results compared to appropriate baselines and good ideas (especially the mutual information for augmentations), I had a lot of trouble reading and understanding the paper and method.  (Note, I'm not an expert in RL, but do have familiarity with the subject).

Many of the objective formulations are presented at a high mathematical level while relegating crucial explanations to the appendices.  This is especially true of the core equations 4, 5 and 6, along with Alg 1, which offer definitions of objectives but aren't very clear on the actual steps performed by the algorithm in updating them.  There is also a lot to keep track of here, and simpler explanations and/or diagrams would help.  As I think the ideas are good and have promising results, I'm confident the explanations can be simplified more.

Appendix M is another case in point:  These are all good experiments and explanations illustrating the behavior of the method and its results quite well, and also provide concrete examples of the tasks and how the MRL state interpolations fit together.  Putting more of this in the main text would help ground the explanations and highlight key results like performance improvements on the poor tasks used in the mining step.  (Though I recognize this may be more my opinion, pushing these aside in favor of the convergence results in the main text is exactly backwards --- while the convergence results are good to see and summarize, I think much of Appendix M offers more explanatory value of the core ideas in the method).

**Questions:**

* It might also be interesting contrast the selection of critical tasks to coreset selection

* Alg. 1:  This doesn't say clearly when (or whether or how much) to run the MRL learning/tuning to update $\theta$ and how many times this should be run to estimate the P(\theta) distribs, all of this action seems to be subsumed in the last sentence "Estimate P*."

* l.145:  "We include the algorithm to solve the problem (2) in Appendix C." --- Eq 2 looks like it would requires rerunning the the entire meta-learning loops to solve the inner argmax_theta:  this argmax is exactly a weighted form of eq (1).  however the algorithm in Appendix C doesn't seem to do that, what is actually done here is more of an alternating coordinate descent.  What is actually done should be explained more up-front in the main text.

* line 326 says P(theta|{T^cri}) (non-augmented; no bars) is a marginalization of P(theta|{Tbar^cri}) (augmented with bars) over all sampled mixture coeffs lambda.   How is this possible,
given that the non-augmented tasks are the source of the states forming the convex hull of hte augmented states?  Shouldn't they be evaluated at lambda=0 or 1 to get the T^cri samples?  Or does this distribution mean something else?

* It looks likely that critical task weights will tend to be lower for well-represented tasks and higher for underrepresented (though this is not the only factor).  In a limiting case, if there are duplicate tasks in the meta-training set, the weights can be spread among them.  So if tasks that are near-dups are the ones important for a poor performing task, the weights may be lower and they could be missed in the mining selection.  Does this happen or is it not an issue in practice?

---

> ### Author Response · Authors · 2025-11-19
> **Response to Reviewer u1Lj**
>
> Thank you for your constructive reviews. We appreciate that you find our work interesting and sound, and we believe that our discussion can lead to a better paper. We have uploaded a revised version with revisions highlighted in blue. Because the line numbers in the revised version differ from those in the original, we will use "(original)" and "(revised)" to indicate whether a referenced line number corresponds to the original or the revised version. We address your comments in a point-by-point manner.
>
> **Weakness 1**: While there are some good results compared to appropriate baselines and good ideas (especially the mutual information for augmentations), I had a lot of trouble reading and understanding the paper and method. (Note, I'm not an expert in RL, but do have familiarity with the subject). Many of the objective formulations are presented at a high mathematical level while relegating crucial explanations to the appendices. This is especially true of the core equations 4, 5 and 6, along with Alg 1, which offer definitions of objectives but aren't very clear on the actual steps performed by the algorithm in updating them. There is also a lot to keep track of here, and simpler explanations and/or diagrams would help. As I think the ideas are good and have promising results, I'm confident the explanations can be simplified more. Appendix M is another case in point: These are all good experiments and explanations illustrating the behavior of the method and its results quite well, and also provide concrete examples of the tasks and how the MRL state interpolations fit together. Putting more of this in the main text would help ground the explanations and highlight key results like performance improvements on the poor tasks used in the mining step. (Though I recognize this may be more my opinion, pushing these aside in favor of the convergence results in the main text is exactly backwards --- while the convergence results are good to see and summarize, I think much of Appendix M offers more explanatory value of the core ideas in the method).
> **Answer**: Thank you for mentioning the involved math and moving more explanations from appendix to the main text. Following your suggestion, we have made the following revisions in the revised version to reduce the heavy math and provide more intuitive explanation and empirical evaluation in the main text:
> First, we add explanations and insights to help readers better understand the equations (4), (5), (6), in lines 251-257(revised), lines 267-273(revised), and lines 276-286(revised), respectively.
> Second, we completely rewrite the algorithm section (Section 4.2) to reduce the mathematical description of Algorithm 1 and provide simpler and more intuitive explanation and we explicitly explain which steps in Algorithm 1 solve which problems.
> Third, we simplify the theoretical analysis section where we move the intermediate theoretical statements to Appendix I (of the revised version) and only provide the final theoretical statement, i.e., generalization improvement, as the theoretical summarization in the main text.
> Fourth, we move the additional empirical evaluations in Appendix M (of the original version) to the main text to help explain the core idea of the method, including visualization of the critical tasks (lines 386-393(revised)), performance improvement on poorly adapted tasks and performance degradation on non-poorly adapted tasks (lines 429-458(revised)), and ablation on number of critical tasks (lines 407-428(revised)).

---

> > ### Author Response · Authors · 2025-11-19
> >
> > **Question 1**: It might also be interesting contrast the selection of critical tasks to coreset selection
> > **Answer**: Thank you for mentioning coreset selection. Note that the purposes of the selection of critical tasks and coreset selection are different. Coreset selection aims to select a small and representative subset of training tasks that can approximate training on the full training set. In contrast, the critical tasks are the most important tasks to the poorly adapted tasks and are not representative of the full training set. We include an additional experiment that compares the performance of training on all training tasks, training on critical tasks, training on coreset selection [A1], and our method (XMRL) on both poorly adaped tasks and non-poorly adapted tasks.
> >
> > | Environment    | Task Type                | Training on All Tasks | XMRL (Ours) | Training on Critical Tasks | Training on Coreset Selection |
> > |----------------|---------------------------|------------------------|-------------|-----------------------------|-------------------------------|
> > | **Drone**      | poorly adapted tasks      | 0.55                  | 0.93        | 0.81                        | 0.56                          |
> > |                | non-poorly adapted tasks  | 0.95                  | 0.98        | 0.48                        | 0.94                          |
> > | **Stock market** | poorly adapted tasks    | 71.05                 | 381.33      | 315.01                      | 69.27                         |
> > |                | non-poorly adapted tasks  | 431.15                | 431.08      | 62.72                       | 428.16                        |
> > | **HalfCheetah** | poorly adapted tasks     | -162.09               | -55.00      | -78.17                      | -158.29                       |
> > |                | non-poorly adapted tasks  | -45.59                | -42.10      | -182.22                     | -48.82                        |
> > | **Ant**        | poorly adapted tasks      | 39.68                 | 99.67       | 67.79                       | 37.29                         |
> > |                | non-poorly adapted tasks  | 115.88                | 124.02      | 37.72                       | 110.28                        |
> >
> > The above results demonstrate that (1) the performance of coreset selection aligns with training on all training tasks on both poorly adapted tasks and non-poorly adapted tasks; (2) training only on the critical tasks can significantly improve performance on the poorly adapted tasks but it is also biased such that the performance on the non-poorly adapted tasks is significantly degraded; (3) our method (XMRL) augments the critical tasks and trains on both augmented critical tasks and unaugmented non-critical tasks, and thus can improve performance on poorly adapte tasks while avoiding significant degradation on non-poorly adapted tasks.

---

> ### Author Response · Authors · 2025-11-19
>
> **Question 2**: Alg. 1: This doesn't say clearly when (or whether or how much) to run the MRL learning/tuning to update $\theta$ and how many times this should be run to estimate the $P(\theta)$ distribs, all of this action seems to be subsumed in the last sentence "Estimate P*."
> **Answer**: Thank you for raising this question about how many lower-level MRL updates are required to estimate the posterior distribution $P(\theta)$. We elaborate how to efficiently learn the distribution in Appendix N.1 in the revised version.
> At each iteration $k$, the upper-level update produces a new mixing-coefficient distribution $P\_{\phi\_{\lambda,k}}(\lambda)$. The lower level samples a set $\\{\lambda\_{i,k}\\}\_{i=1}^{N^{\text{cri}}}$ of mixing coefficients to augment the critical tasks and computes the posterior $P^{\ast}(\theta|\\{\bar{\mathcal{T}}\_i^{\text{cri}}(\lambda\_{i,k})\\})$ by solving the optimization problem $ \phi^{\ast}(\\{\lambda\_{i,k}\\}\_{i=1}^{N^{\text{cri}}})= \mathop{\arg\max}\_{\phi}E\_{p\_{\phi}(\theta)}[L(\theta,\\{\bar{\mathcal{T}}\_i^{\text{cri}}(\lambda\_{i,k})\\}\_{i=1}^{N^{\text{cri}}},\\{\mathcal{T}\_i^{\text{tr}}\\}\_{i=1}^{N^{\text{tr}}-N^{\text{cri}}})]$ in (5) where $P\_{\phi^{\ast}(\\{\lambda\_{i,k}\\}\_{i=1}^{N^{\text{cri}}})}(\theta)=P^{\ast}(\theta|\\{\bar{\mathcal{T}}\_i^{\text{cri}}(\lambda\_{i,k})\\})$. The most straightforward way to solve the optimization problem in (5) at each iteration $k$ is that, we randomly initialize $\phi$ and solve the optimization problem until convergence to get $\phi^{\ast}(\\{\lambda\_{i,k}\\}\_{i=1}^{N^{\text{cri}}})$. However, it requires many gradient-ascent steps which can be computationally expensive. To address this issue, we use warm start. Specifically, instead of randomly initializing $\phi$, we initialize $\phi$ in iteration $k$ with the parameter $\phi(\\{\lambda\_{i,k-1}\\}\_{i=1}^{N^{\text{cri}}})$ learned from the previous iteration $k-1$ and use one-step gradient ascent to obtain the new parameter $\phi(\\{\lambda\_{i,k}\\}\_{i=1}^{N^{\text{cri}}})$. The warm start can significantly reduce the number of gradient-ascent steps because it provides a good initialization. Note that the mixing coefficient parameters in two consecutive iterations $k-1$ and $k$ are close because they are only different in one-step gradient $\phi\_{\lambda,k}-\phi\_{\lambda,k-1}=\beta g\_{\lambda,k-1}$ where $\beta$ is a small learning rate and $g\_{\lambda,k-1}$ is the gradient. Therefore, it is expected that their corresponding lower-level optimal parameters $\phi^{\ast}(\\{\lambda\_{i,k}\\}\_{i=1}^{N^{\text{cri}}})$ and $\phi^{\ast}(\\{\lambda\_{i,k-1}\\}\_{i=1}^{N^{\text{cri}}})$ are also close. Note that we only use one-step gradient ascent to update $\phi$ and thus $\phi(\\{\lambda\_{i,k-1}\\}\_{i=1}^{N^{\text{cri}}})$ and $\phi^{\ast}(\\{\lambda\_{i,k-1}\\}\_{i=1}^{N^{\text{cri}}})$ are different, however, $\phi(\\{\lambda\_{i,k-1}\\}\_{i=1}^{N^{\text{cri}}})$ is updated towards $\phi^{\ast}(\\{\lambda\_{i,k-1}\\}\_{i=1}^{N^{\text{cri}}})$ and $\phi(\\{\lambda\_{i,k}\\}\_{i=1}^{N^{\text{cri}}})$ is initialized from $\phi(\\{\lambda\_{i,k-1}\\}\_{i=1}^{N^{\text{cri}}})$ and updated towards $\phi^{\ast}(\\{\lambda\_{i,k}\\}\_{i=1}^{N^{\text{cri}}})$. Intuitively, it is expected that $\phi(\\{\lambda\_{i,k}\\}\_{i=1}^{N^{\text{cri}}})$ will approach $\phi^{\ast}(\\{\lambda\_{i,k}\\}\_{i=1}^{N^{\text{cri}}})$ and finally become $\phi^{\ast}(\\{\lambda\_{i,k}\\}\_{i=1}^{N^{\text{cri}}})$ when $k$ increases. In fact, it has been theoretically guaranteed that $\phi(\\{\lambda\_{i,k}\\}\_{i=1}^{N^{\text{cri}}})$ will finally reach $\phi^{\ast}(\\{\lambda\_{i,k}\\}\_{i=1}^{N^{\text{cri}}})$ when $k$ increases [A2].

---

> ### Author Response · Authors · 2025-11-19
>
> **Question 3**: l.145(original): "We include the algorithm to solve the problem (2) in Appendix C." --- Eq 2 looks like it would requires rerunning the the entire meta-learning loops to solve the inner argmax theta: this argmax is exactly a weighted form of eq (1). however the algorithm in Appendix C doesn't seem to do that, what is actually done here is more of an alternating coordinate descent. What is actually done should be explained more up-front in the main text.
> **Answer**: Thank you for mentioning Algorithm 2. Before we explain Algorithm 2, we would like to first discuss two major classes of algorithms to solve bi-level optimization problems: double-loop algorithm and single-loop algorithm. Double-loop algorithms have two nested loops: inner loop and outer loop. At each outer iteration, the inner loop first solves the lower-level problem by multi-step gradient ascent to obtain the weighted meta-policy and then the weighted meta-policy is used to update the weight vector in the outer loop. What you describe is actually a kind of double-loop algorithm that requires rerunning the entire meta-learning loop to solve the lower-level problem. However, double-loop algorithms are usually computationally expensive because at each outer iteration, it requires to completely solve the lower-level optimization problem in the inner loop. To address this issue, single-loop algorithms are proposed where there is only one loop and at each iteration, we start from the parameters in the previous iteration and only partially solve both the upper-level and lower-level problems via one-step gradient ascent (which aligns with the "alternating coordinate descent" behavior). The single-loop algorithm is more efficient as it does not require an inner loop to fully solve the lower-level problem and it has been demonstrated that the single-loop algorithm can reach the same and even better results than double-loop algorithms [A2]. Algorithm 2 is a single-loop algorithm where we only partially solve the lower-level problem via one-step gradient ascent for efficiency.
> We have included clarification in the main text in lines 145-147 in the revised version.
>
> **Question 4**: line 326(original) says $P(\theta|\\{\mathcal{T}\_i^{\text{cri}}\\}\_{i=1}^{N^{\text{cri}}})$ (non-augmented; no bars) is a marginalization of $P(\theta|{\\{\bar{\mathcal{T}}\_i^{\text{cri}}(\lambda\_i)\\}\_{i=1}^{N^{\text{cri}}}})$ (augmented with bars) over all sampled mixture coeffs lambda. How is this possible, given that the non-augmented tasks are the source of the states forming the convex hull of the augmented states? Shouldn't they be evaluated at lambda=0 or 1 to get the $\\{\mathcal{T}\_i^{\text{cri}}\\}\_{i=1}^{N^{\text{cri}}}$
> **Answer**: It is very insightful of you to question why the posterior $P(\theta|\\{\mathcal{T}\_i^{\text{cri}}\\}\_{i=1}^{N^{\text{cri}}})$ cannot be evaluated simply by setting $\lambda=0$ or $\lambda=1$. We clarify this point in detail in Appendix F with an accompanying block diagram, and we elaborate on the reasoning here. The key issue is how the parameter $\theta$ is computed.
> In the vanilla meta-learning (problem (1)), the meta-parameter $\theta$ is trained directly on the unaugmented tasks. In this case, the critical tasks are not augmented and thus the posterior $P(\theta|\\{\mathcal{T}\_i^{\text{cri}}\\}\_{i=1}^{N^{\text{cri}}})$ does not involve any marginalization over $\\{\lambda\_i\\}\_{i=1}^{N^{\text{cri}}}$ as augmentation does not play a role at all. Setting $\lambda=0$ or $\lambda=1$ is equivalent to this scenario: there is no augmentation, so posterior evaluation is directly on the critical tasks. However, in our setting, the critical tasks are not used directly to train $\theta$. Instead, there is a pipeline: unaugmented_critical_tasks $\Rightarrow$ augmented_critical_tasks $\Rightarrow$ $\theta$. That is, the critical tasks are first augmented and then $\theta$ is trained on the augmented critical tasks. Therefore, when we only know the unaugmented critical tasks, we cannot directly get the distribution of $\theta$ because $\theta$ will be trained on the augmented critical tasks but we do not know the augmented critical tasks yet. In this case, in order to get the posterior distribution of $\theta$, we need to consider all possible augmentations by marginalizing over $\\{\lambda\_i\\}\_{i=1}^{N^{\text{cri}}}$.

---

> ### Author Response · Authors · 2025-11-19
>
> **Question 5**: It looks likely that critical task weights will tend to be lower for well-represented tasks and higher for underrepresented (though this is not the only factor). In a limiting case, if there are duplicate tasks in the meta-training set, the weights can be spread among them. So if tasks that are near-dups are the ones important for a poor performing task, the weights may be lower and they could be missed in the mining selection. Does this happen or is it not an issue in practice?
> **Answer**: Thank you for mentioning this "weight dilution" concern. In practice, this does not cause missing critical tasks in mining selection. First, our selection is rank-based: we always pick the top $N^{\text{cri}}$ tasks according to their learned weights, rather than using an absolute threshold. Therefore, even if duplicates slightly reduce the individual weights of some useful tasks, they will still be selected as long as their weights remain higher than those of truly non-helpful tasks. To verify this, we have conducted an ablation study on the number $N^{\text{cri}}$ in Appendix M.8(original) (or lines 407-428(revised)) where we vary the selection ratio from top 10% to top 50%. The best performance consistently occurs when selecting the top 10–20% tasks, and performance degrades when more than the top 30% are included. This indicates that the important tasks are reliably ranked near the top and are not lost due to dilution. More importantly, we visualize the critical tasks in Appendix M.3(original) (or lines 386-393(revised)), We can see that critical tasks are all selected even if they are close to each other.
>
> [A1] Donglin Zhan and James Anderson, "Data-Efficient and Robust Task Selection for Meta-Learning", CVPR, 2024
>
> [A2] Mingyi Hong, Hoi-To Wai, Zhaoran Wang, and Zhuoran Yang, "A two-timescale stochastic algorithm framework for bilevel optimization: Complexity analysis and application to actor-critic", SIAM Journal on Optimization, 2023.

---

### Author Response · Authors · 2025-11-30
**Summary of Reviewer-author discussion**

Dear AC,

Thank you for your time to consider our paper and our discussions with the reviewers. Here we would like to summarize our discussions with the reviewers.

We got an initial rating of 6 (reviewer 4uvj), 6 (reviewer HExQ), 2 (reviewer kbyg), 4 (reviewer u1Lj). During discussion

* Reviewer 4uvj and Reviewer HExQ acknowledge that we have addressed their concerns and decide to maintain their positive scores.
* Reviewer kbyg also states that we have addressed the concerns and decides to **raise the rating from 2 to 6**. The concerns of Reviewer kbyg mainly arise from the confusion about our proof and thus we provide detailed proof to clarify the confusion.
* Reviewer u1Lj has not yet responded in the discussion, but the major concern of Reviewer u1Lj is not technical. The major concern is the dense math. To address this concern, we have significantly revised the paper (in the revised version) to reduce the heavy math and include more explanations and empirical evaluations. The reviewer also asks for a new baseline for comparison and we have included this new baseline for comparison.

In summary, we believe that we have addressed the concerns of all reviewers. Three reviewers (4uvj, HExQ, and kbyg) explicitly acknowledge that their concerns are resolved and either raise the rating or maintain their positive ratings. Although the reviewer u1Lj has not replied yet, the concerns of the reviewer u1Lj are mainly about presentation and a simple baseline. We have significantly revised the presentation accordingly and include the comparison to baseline.

We hope that this summary is helpful when you consider the decision.

Best,
Authors

---

### Meta-Review · Area_Chair_F52C · 2026-01-09

**Summary:**

This paper addresses generalization in meta-reinforcement learning (meta-RL) by introducing a novel two-stage approach that leverages example-based explanation and information-theoretic task augmentation.Stage 1 identifies critical training tasks most relevant to poorly adapted tasks through a bilevel optimization that assigns task-specific importance weights. Stage 2 improves generalization by maximizing conditional mutual information (CMI) between the meta-policy and augmented critical tasks. The authors use a learnable task mixup augmentation strategy to achieve this, rather than fixed data-mixing rules. Theoretical analysis guarantees O(1/\sqrt{K}) convergence and improved generalization bounds. Experiments on real-world (drone, stock trading) and simulation benchmarks (MuJoCo, Meta-World) show consistent gains over MAML and recent meta-RL improvement baselines (task weighting, meta-augmentation, meta-regularization).

The reviewers generally appreciated the technical depth and strength of this work. The major concerns were:

1. Density of exposition.
2. Novelty in theory.
3. Meta-learning assumption realism and extension to offline settings.
4. Scalability concerns.
5. May be hacking the test set.

**Reviewer Concerns:**

Overall the rebuttal has addressed essentially all of the reviewer concerns. 3 of the reviewers alreaady raised their rating and for the last reviewer, the authors made all the requested changes. This paper should be an accept.

**Reviewer Scores:**

2 reviewers gave and kept a 6, one reviewer went from a 2 to a 6. And one reviewer u1Lj is a 4, but largely for misunderstanding/presentation reasons. This would likely have been updated.

---

### Decision · Program_Chairs · 2026-01-26

Accept (Poster)